# TDP-43 represses cryptic exon inclusion in the FTD–ALS gene *UNC13A*

X. Rosa Ma[1,16], Mercedes Prudencio[2,3,16], Yuka Koike[2,3,16], Sarat C. Vatsavayai[4,5], Garam Kim[1,6], Fred Harbinski[7], Adam Briner[1,8], Caitlin M. Rodriguez[1], Caiwei Guo[1], Tetsuya Akiyama[1], H. Broder Schmidt[9], Beryl B. Cummings[7], David W. Wyatt[7], Katherine Kurylo[7], Georgiana Miller[7], Shila Mekhoubad[7], Nathan Sallee[7], Gemechu Mekonnen[10,11], Laura Ganser[12], Jack D. Rubien[13], Karen Jansen-West[2], Casey N. Cook[2,3], Sarah Pickles[2,3], Björn Oskarsson[14], Neill R. Graff-Radford[14], Bradley F. Boeve[15], David S. Knopman[15], Ronald C. Petersen[15], Dennis W. Dickson[2,3], James Shorter[13], Sua Myong[10,11,12], Eric M. Green[7], William W. Seeley[4,5], Leonard Petrucelli[2,3 ✉] & Aaron D. Gitler[1 ✉]

A hallmark pathological feature of the neurodegenerative diseases amyotrophic lateral sclerosis (ALS) and frontotemporal dementia (FTD) is the depletion of RNA-binding protein TDP-43 from the nucleus of neurons in the brain and spinal cord[1]. A major function of TDP-43 is as a repressor of cryptic exon inclusion during RNA splicing[2–4]. Single nucleotide polymorphisms in *UNC13A* are among the strongest hits associated with FTD and ALS in human genome-wide association studies[5,6], but how those variants increase risk for disease is unknown. Here we show that TDP-43 represses a cryptic exon-splicing event in *UNC13A*. Loss of TDP-43 from the nucleus in human brain, neuronal cell lines and motor neurons derived from induced pluripotent stem cells resulted in the inclusion of a cryptic exon in *UNC13A* mRNA and reduced UNC13A protein expression. The top variants associated with FTD or ALS risk in humans are located in the intron harbouring the cryptic exon, and we show that they increase *UNC13A* cryptic exon splicing in the face of TDP-43 dysfunction. Together, our data provide a direct functional link between one of the strongest genetic risk factors for FTD and ALS (*UNC13A* genetic variants), and loss of TDP-43 function.

TDP-43, encoded by the *TARDBP* gene, is an abundant, ubiquitously expressed RNA-binding protein that normally localizes to the nucleus. It has a role in fundamental RNA-processing activities, including RNA transcription, alternative splicing and RNA transport[7]. A major splicing regulatory function of TDP-43 is to repress the inclusion of cryptic exons during splicing[2,8–10]. Unlike normal conserved exons, these cryptic exons occur in introns and are normally excluded from mature mRNAs. When TDP-43 is depleted from cells, these cryptic exons get spliced into messenger RNAs, often introducing frame shifts and premature termination, or even reduced RNA stability. However, the key cryptic splicing events that are integral to disease pathogenesis remain unknown.

*STMN2*—which encodes stathmin 2, a regulator of microtubule stability—is the gene whose expression is most significantly reduced when TDP-43 is depleted from neurons[3,4]. *STMN2* harbours a cryptic exon (exon 2a) that is normally excluded from the mature *STMN2* mRNA.

The first intron of *STMN2* contains a TDP-43 binding site. When TDP-43 is lost or its function is impaired, exon 2a gets incorporated into the mature mRNA. Exon 2a harbours a stop codon and a polyadenylation signal—this results in truncated *STMN2* mRNA and eightfold reduction[3] of stathmin 2. Aberrant splicing and reduced stathmin 2 levels seem to be a major feature of sporadic and familial cases of ALS (except those with *SOD1* mutations)[3,4] and in frontotemporal lobar degeneration (FTLD) due to TDP-43 proteinopathy[11] (FTLD-TDP). The discovery of *STMN2* cryptic exon splicing in ALS and FTLD-TDP highlights a key mRNA target—we aimed to identify other possible mRNA targets.

To discover cryptic splicing targets regulated by TDP-43 that may also have a role in disease pathogenesis, we used a recently generated RNA sequencing (RNA-seq) dataset[12]. To generate this dataset, fluorescence-activated cell sorting (FACS) was used to enrich neuronal nuclei with and without TDP-43 from postmortem brain tissue from patients with FTD and ALS (FTD–ALS); RNA-seq was performed

[1]Department of Genetics, Stanford University School of Medicine, Stanford, CA, USA. [2]Department of Neuroscience, Mayo Clinic, Jacksonville, FL, USA. [3]Neuroscience Graduate Program, Mayo Clinic Graduate School of Biomedical Sciences, Jacksonville, FL, USA. [4]Department of Neurology, University of California San Francisco, San Francisco, CA, USA. [5]Department of Pathology, University of California San Francisco, San Francisco, CA, USA. [6]Neurosciences Interdepartmental Program, Stanford University School of Medicine, Stanford, CA, USA. [7]Maze Therapeutics, South San Francisco, CA, USA. [8]Clem Jones Centre for Ageing Dementia Research (CJCADR), Queensland Brain Institute (QBI), The University of Queensland, Brisbane, Queensland, Australia. [9]Department of Biochemistry, Stanford University School of Medicine, Stanford, CA, USA. [10]Program in Cell, Molecular, Developmental Biology, and Biophysics, Johns Hopkins University, Baltimore, MD, USA. [11]Department of Biology, Johns Hopkins University, Baltimore, MD, USA. [12]Department of Biophysics, Johns Hopkins University, Baltimore, MD, USA. [13]Department of Biochemistry and Biophysics, Perelman School of Medicine, University of Pennsylvania, Philadelphia, PA, USA. [14]Department of Neurology, Mayo Clinic, Jacksonville, FL, USA. [15]Department of Neurology, Mayo Clinic, Rochester, MN, USA. [16]These authors contributed equally: X. Rosa Ma, Mercedes Prudencio, Yuka Koike. ✉e-mail: petrucelli.leonard@mayo.edu; agitler@stanford.edu

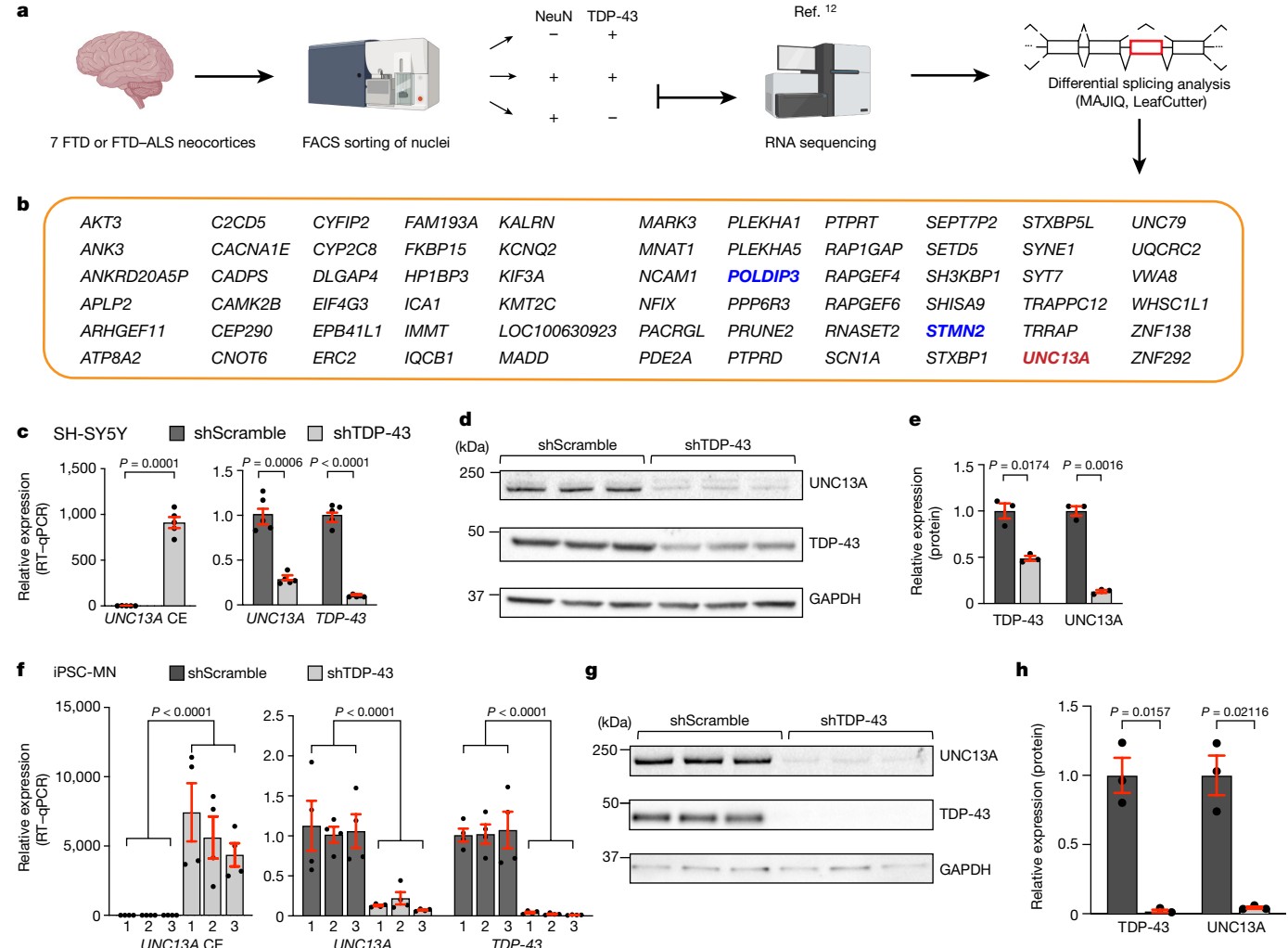

**Fig. 1 | Nuclear depletion of TDP-43 causes CE inclusion in *UNC13A* RNA and reduced expression of UNC13A protein. a**, Splicing analyses were performed on RNA-sequencing results from TDP-43-positive and TDP-43-negative neuronal nuclei isolated from frontal cortices of seven patients with FTD or FTD–ALS. Some illustrations were created with BioRender.com. **b**, Sixty-six alternatively spliced genes identified by both MAJIQ ($P(\Delta\Psi > 0.1) > 0.95$) and LeafCutter ($P < 0.05$). Genes in blue are previously validated TDP-43 splicing targets[3,4,11,15]. **c**, **f**, RT–qPCR confirmed inclusion of CE in *UNC13A* mRNA upon TDP-43 depletion in SH-SY5Y cells ($n = 5$ cell culture experiments for each condition; two sided-Welch two-sample $t$-test; mean ± s.e.m.) (**c**) and in 3

independent lines of iPSC-MNs ($n = 2$ independent cell culture experiments, each with 2 technical replicates for each iPSC-MN) (**f**). *RPLPO* was used to normalize RT–qPCR. Three-way ANOVA; mean ± s.e.m. *TDP-43* is also known as *TARDBP*. **d**, **e**, **g**, **h**, Immunoblotting for UNC13A and TDP-43 protein levels in SH-SY5Y cells (**d**; quantified in **e**) and iPSC-MNs (**g**; quantified in **h**) treated with scramble (shScramble) or *TDP-43* shRNA ($n = 3$ independent cell culture experiments for each condition). *GAPDH* served as a loading control. Two-sided Welch two-sample $t$-test, mean ± s.e.m. Gel source data are shown in Supplementary Fig. 1a, b.

to compare the transcriptomic profiles of TDP-43-positive and TDP-43-negative neuronal nuclei. This identified several differentially expressed genes[12]. We re-analysed the data to identify novel alternative splicing events affected by the loss of nuclear TDP-43. We performed splicing analyses using two pipelines, MAJIQ[13] and LeafCutter[14], designed to detect novel splicing events (Fig. 1a). We identified 266 alternative splicing events ($P(\Delta\Psi > 0.1) > 0.95$; where $\Delta\Psi$ signifies changes of local splicing variations between two conditions) with MAJIQ and 152 with LeafCutter ($P < 0.05$). There were 66 alternatively spliced genes in common between the two analyses (Fig. 1b), probably because each tool uses different definitions for transcript variations and different criteria to control for false positives (Supplementary Note 1). These genes have at least one region that both tools identified to be alternatively spliced (Supplementary Table 1). Among the alternatively spliced genes identified by both tools were *STMN2* and *POLDIP3*, both of which have been extensively validated as bona fide targets of splicing by TDP-43[3,4,11,15].

*UNC13A* was one of the genes with the most significant levels of alternative splicing (MAJIQ $\Delta\Psi = 0.779$; LeafCutter $\Delta\Psi = 0.8360$; $P < 0.0001$) in neurons with nuclear TDP-43 depletion (Fig. 1b). Depletion of TDP-43 resulted in the inclusion of a 128-bp or a 178-bp cryptic exon between the canonical exons 20 and 21 (hg38; chr19:17642414–17642541 (128bp); chr19:17642591–17642414 (178 bp)) (Extended Data Fig. 1a–e, Supplementary Note 2). The two cryptic exons share the same 3′ end but the 178-bp cryptic exon is 50 bp longer than the 128-bp cryptic exon at the 5′ end. The cryptic exons were almost completely absent in wild-type neuronal nuclei (Fig. 1f) and are not present in any of the known human isoforms of *UNC13A*[16]. Furthermore, analysis of ultraviolet cross-linking and immunoprecipitation (iCLIP) data for TDP-43[17] provides evidence that TDP-43 binds directly to the intron harbouring these cryptic exons (shown by mapped reads) (Extended Data Fig. 1g). Because of the much higher abundance of the 128-bp cryptic exon in the RNA-seq data that we analysed, we focused our analyses on this cryptic exon, which we refer to as CE. Intron 20–21 of *UNC13A* and the CE sequence are conserved

among most primates (Extended Data Fig. 2a, b) but not in mouse (Extended Data Fig. 2c, d), similar to *STMN2* and other cryptic splicing targets of TDP-43[2-4]. Together, these results suggest that TDP-43 functions to repress the inclusion of a cryptic exon in the *UNC13A* mRNA.

To determine whether TDP-43 directly regulates this *UNC13A* cryptic splicing event, we used short hairpin RNA (shRNA) to reduce TDP-43 levels in SH-SY5Y cells and quantitative PCR with reverse transcription (RT–qPCR) to detect CE inclusion in *UNC13A* transcript. CE was present in cells with TDP-43 depletion but not in cells treated with control shRNA (Fig. 1e). Along with the increase in CE, there was a corresponding decrease in the amount of the canonical *UNC13A* transcript upon TDP-43 depletion (Fig. 1c). By immunoblotting, we also observed a marked reduction in UNC13A protein in TDP-43-depleted cells (Fig. 1d, e). Reducing levels of TDP-43 in motor neurons derived from induced pluripotent stem cells (iPSC-MNs) (Fig. 1f–h, Extended Data Fig. 3a, b, Supplementary Table 3) and excitatory neurons (i³Ns) derived from human induced pluripotent stem (iPS) cells (Extended Data Fig. 3c) also resulted in CE inclusion and a reduction in *UNC13A* mRNA and protein. We confirmed insertion of the cryptic exon sequences into the mature transcript by amplicon sequencing of the product of reverse transcription with PCR (RT–PCR) (Extended Data Fig. 3d; see Supplementary Note 2) and demonstrated that the insertions introduce premature stop codons (Extended Data Fig. 3f–i), consistent with the observed decrease in UNC13A protein. *UNC13A* gene expression is probably regulated at multiple levels beyond simply inclusion of the cryptic exons. Other aspects of the cryptic exon-inclusion event (for example, aberrant peptides produced from it) could cause defects, although we do not yet have evidence that such peptides are produced. Thus, lowering levels of TDP-43 in human cells and neurons causes inclusion of CE in the *UNC13A* transcript, resulting in decreased UNC13A protein.

*UNC13A* belongs to a family of genes originally discovered in *Caenorhabditis elegans* and was named on the basis of the uncoordinated (*unc*) movements exhibited by animals with mutations in these genes[18], owing to deficits in neurotransmitter release. *UNC13A* encodes a large multidomain protein that is expressed in the nervous system, where it localizes to most synapses in the central nervous system and neuromuscular junctions, and has an essential role in the vesicle priming step, prior to synaptic vesicle fusion[19-22]. UNC13A is an essential neuronal protein because mice lacking Unc13a (also called Munc13-1) exhibit functional deficits at glutamatergic synapses, demonstrated by a lack of fusion-competent synaptic vesicles, and die within a few hours of birth[21]. Our data suggest that depletion of TDP-43 leads to loss of this critical synaptic protein. As well as UNC13A, several other genes encoding synaptic proteins are mis-spliced upon TDP-43 depletion (Fig. 1b). We validated the splicing events in three of these genes (*KALRN*, *RAPGEF6* and *SYT7*) in induced pluripotent stem (iPS) cell-derived neurons (iNs) using RT–qPCR (Extended Data Fig. 4), providing evidence that disruption of synaptic function could be a major mechanism in the pathogenesis of ALS and FTD.

To extend our analysis of *UNC13A* cryptic exon inclusion to a larger collection of patient samples, we first analysed a series of 117 frontal cortex brain samples from the Mayo Clinic Brain Bank using RT–qPCR and a pair of primers that detects the shared 3′ end of the cryptic exons. We found a significant increase in *UNC13A* cryptic exon in the frontal cortices of patients with FTLD-TDP compared with healthy controls (Fig. 2a, Extended Data Fig. 5a). Next, we analysed brain samples from the New York Genome Center (NYGC). We interrogated RNA-seq data from 1,151 tissue samples from 413 individuals (with multiple tissues per individual; see Supplementary Note 3), 330 of whom are patients with ALS or FTD. We detected *UNC13A* splice variants in nearly 50% of the frontal and temporal cortical tissues donated by patients with neuropathologically confirmed FTLD-TDP (Fig. 2b) and in some of the patients with ALS whose pathology has not been confirmed (Extended Data Fig. 5b). Notably, we did not observe *UNC13A* splice variants in any of the samples from patients with FTLD associated with FUS

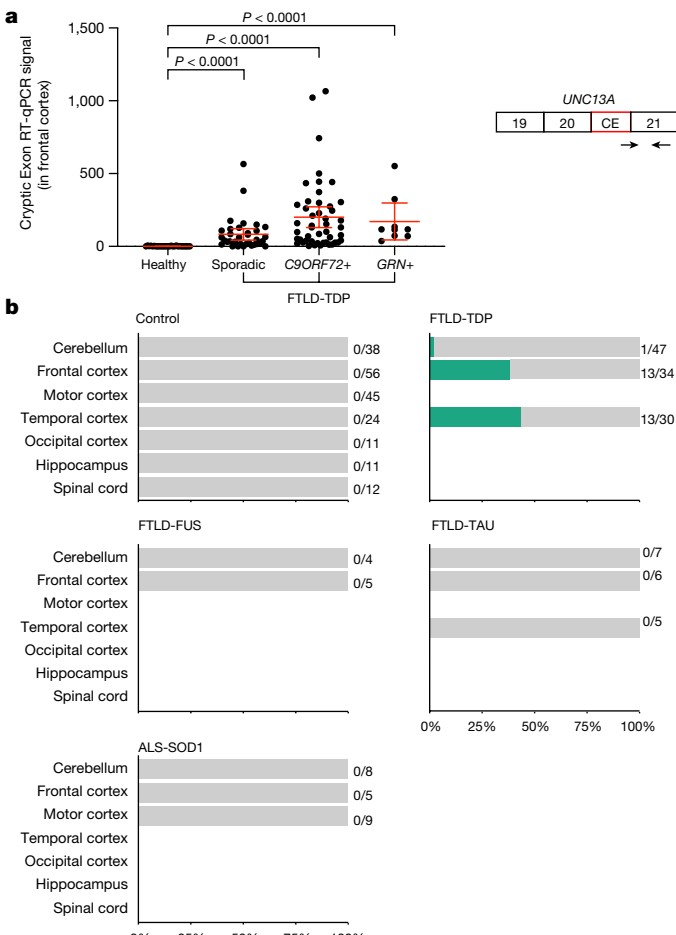

**Fig. 2 | *UNC13A* CE inclusion in human TDP-43 proteinopathies. a**, *UNC13A* CE expression level is increased in the frontal cortices of patients with FTLD-TDP. *GAPDH* and *RPLP0* were used to normalize the RT–qPCR (two-tailed Mann–Whitney test, mean ± 95% confidence interval). The schematic to the right shows the localization of the primer pair (arrows) used for the RT–qPCR assay. Healthy: *n* = 27; sporadic FTLD-TDP: *n* = 34; *C9ORF72*+ FTLD-TDP: *n* = 47; *GRN*+ FTLD-TDP: *n* = 9. **b**, *UNC13A* CE is detected in nearly 50% of frontal cortical tissues and temporal cortical tissues from neuropathologically confirmed FTLD-TDP patients in bulk RNA-sequencing from the NYGC ALS Consortium cohort. CE is absent in tissues from healthy controls and patients with FTLD-FUS, FTLD-TAU or ALS-SOD1.

(FTLD-FUS) (*n* = 9) or TAU (FTLD-TAU) (*n* = 18) or ALS associated with SOD1 (ALS-SOD1) (*n* = 22), nor in any of the control samples (*n* = 197) (Fig. 2b). Using the same criteria, we detected the known *STMN2* cryptic exon[3,4] in tissues from these patients, and the majority of the *UNC13A* splice variant containing tissues also contained the *STMN2* splice variant (Supplementary Table 2). Hyperphosphorylated TDP-43 (pTDP-43) is a key feature of the pathology of these diseases[1]. We found a strong association between higher levels of pTDP-43 and higher levels of *UNC13A* CE inclusion in patients with FTLD-TDP (Spearman's ρ = 0.610, *P* < 0.0001) (Extended Data Fig. 2c). Thus, *UNC13A* CE inclusion is a robust and specific facet of pathobiology in TDP-43 proteinopathies.

To visualize the *UNC13A* CE within single cells in the human brain, we designed custom BaseScope in situ hybridization probes that specifically bind to the exon 20–CE junction (Fig. 3a) or the exon 20–exon 21 junction (Extended Data Fig. 6c). We used these probes for in situ hybridization, combined with immunofluorescence for NeuN (to detect neuronal nuclei) and TDP-43. We stained sections from the medial frontal pole of four patients with FTLD-TDP and three controls (Supplementary Table 4). In neurons showing loss of nuclear TDP-43 and

**a**

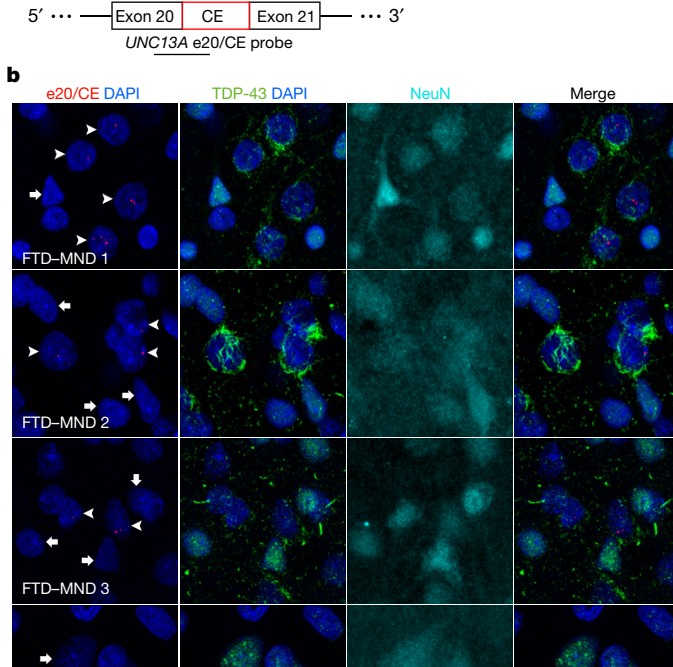

**b**

Fig. 3 | *UNC13A* cryptic splicing is associated with loss of nuclear TDP-43 in patients with FTD and motor neuron disease. a, The design of the *UNC13A* e20/CE BaseScope probe targeting the alternatively spliced *UNC13A* transcript. Each Z binds to the transcript independently, and both must be in close proximity for successful signal amplification, ensuring binding specificity. b, BaseScope in situ hybridization using the *UNC13A* e20/CE probe, combined with immunofluorescence for TDP-43 and NeuN, was performed on sections from the medial frontal pole of patients with FTD and motor neuron disease (FTD–MND) and healthy controls. Representative images illustrate the presence of *UNC13A* CE (arrowheads) in neurons showing depletion of nuclear TDP-43. Neurons with normal nuclear TDP-43 in patients and controls show no CE signal (arrows). Images are maximum intensity projections of a confocal image *z*-stack. Scale bar, 10 μm. Images representative of six non-overlapping images from each individual. We optimized *UNC13A* probes on two cases and two controls in three separate experiments, with similar findings.

accompanying cytoplasmic TDP-43 inclusions, we observed *UNC13A* CE-containing mRNA splice variants in the nucleus (Fig. 3b, Extended Data Fig. 6a). We observed between one and four CE-containing mRNA puncta per nucleus. We did not observe puncta in the cytoplasm, perhaps because CE introduces a premature stop codon, which could lead to nonsense-mediated decay. Controls, however, had universally normal nuclear TDP-43 staining and showed no evidence of *UNC13A* cryptic splicing (Fig. 3b, Extended Data Fig. 6b). Next, we sought to determine whether TDP-43 nuclear depletion is associated with reduced expression of canonical *UNC13A* mRNA. In control brain tissue, *UNC13A* mRNA was widely expressed in neurons across cortical layers (Extended Data Fig. 6d, e). In patients, we saw a trend for reduced *UNC13A* mRNA in neurons showing TDP-43 pathology compared with neighbouring neurons with normal nuclear TDP-43 (Extended Data Fig. 6d, e), consistent with the RT–qPCR data (Fig. 1c, f). These findings suggest that cryptic splicing of *UNC13A* is absent from controls and, in patients, is seen exclusively in neurons showing depleted nuclear TDP-43.

*UNC13A* is one of the top hits for ALS and FTD–ALS in multiple genome-wide association studies[5,6,23–26] (GWAS). Single nucleotide polymorphisms (SNPs) in *UNC13A* are associated with increased risk

of sporadic ALS[5] and sporadic FTLD-TDP pathology, especially type B, the subtype associated with FTD–ALS[6]. In addition to increasing susceptibility to ALS, SNPs in *UNC13A* are associated with shorter survival in patients with ALS[27–30]. But the mechanism by which genetic variation in *UNC13A* increases risk for ALS and FTD is unknown. Notably, the two most significantly associated SNPs, rs12608932 (A>C) and rs12973192 (C>G), are both located in the same intron that we found harbours CE, with rs12973192 located in CE itself (Fig. 4a, c). This immediately suggested that these SNPs (or other nearby genetic variations tagged by these SNPs) might make *UNC13A* more vulnerable to CE inclusion upon TDP-43 depletion. To test this, we analysed the percentage of RNA-seq reads that mapped to intron 20–21 that support the inclusion of CE (Extended Data Fig. 7a, b). Among the seven patients included in the initial splicing analysis (Fig. 1a), two out of three who were homozygous (G/G) and the one patient who was heterozygous (C/G) for the risk allele at rs12973192, showed inclusion of CE in almost every *UNC13A* mRNA that was mapped to intron 20–21. By contrast, the three other patients who were homozygous for the reference allele (C/C) showed much less inclusion of CE (Extended Data Fig. 7a, b). Another way to directly assess the effect of the *UNC13A* risk alleles on CE inclusion is to measure allele imbalance in RNAs from individuals who are heterozygous for the risk allele. Two of the iPSC-MN lines that we used to detect CE inclusion upon TDP-43 knockdown (iPSC-MN1 and iPSC-MN3; Fig. 1h) are heterozygous (C/G) at rs12973192. We performed amplicon sequencing of the RT–PCR product that spans CE and analysed the allele distribution from these two samples (Extended Data Fig. 3d, e) as well as the one patient sample from the original RNA-seq dataset (Fig. 1a) that is heterozygous (C/G) at rs12973192 (Extended Data Fig. 7b). We found a significant difference between the percentage of risk and reference alleles in the spliced variant, with higher inclusion of the risk allele (Fig. 4b, Extended Data Fig. 7c).

Given this evidence for an effect of the risk allele on CE inclusion, we extended our analysis by genotyping patients with FTLD-TDP harbouring CE (*n* = 86) in the Mayo Clinic Brain Bank dataset for the *UNC13A* risk alleles at rs12973192 and rs12608932. Because these two SNPs are in high linkage disequilibrium in the European population[31], we consider them to be on the same haplotype. Thus, we refer to the haplotype that contains reference alleles at both SNPs as the reference haplotype, and the haplotype that contains risk alleles as the risk haplotype. We excluded the one patient who is homozygous for the reference allele (C/C) at rs12973192 but heterozygous (A/C) at rs12608932. The remaining patients (*n* = 85) have exactly the same number of risk alleles at both loci, indicating that it's very likely that they are carriers of the reference haplotype or the risk haplotype. Using a multiple linear regression model, we found a strong positive correlation ($\beta$ = 0.175, *P* = 0.0290) between the number of risk haplotypes and the abundance of *UNC13A* CE inclusion measured by RT–qPCR (Extended Data Fig. 7e). Together, these data suggest that genetic variation in *UNC13A* that increases risk for ALS and FTD in humans promotes CE inclusion upon nuclear depletion of TDP-43.

GWAS SNPs typically do not cause the trait but rather 'tag' other neighbouring genetic variation[32]. Thus, a major challenge in human genetics is to go from a GWAS hit to identifying the causative genetic variation that increases risk for disease[33]. A LocusZoom[34] plot (Fig. 4a) generated using results from an ALS GWAS[35] suggests that the strongest association signal on *UNC13A* is indeed in the region surrounding the two lead SNPs (rs12973192 and rs12608932). To identify other genetic variants in intron 20–21 that might also cause risk for disease by influencing CE inclusion but were not included in the original GWAS, we analysed genetic variants identified in whole genome sequencing data of 297 ALS patients of European descent (Answer ALS; https://www.answerals.org). We searched for novel genetic variants that could be tagged by the two SNPs by looking for other loci in intron 20–21 that are in linkage disequilibrium with both rs12608932 and rs12973192. We found one that fit these criteria: rs56041637 (Extended Data Fig. 7d).

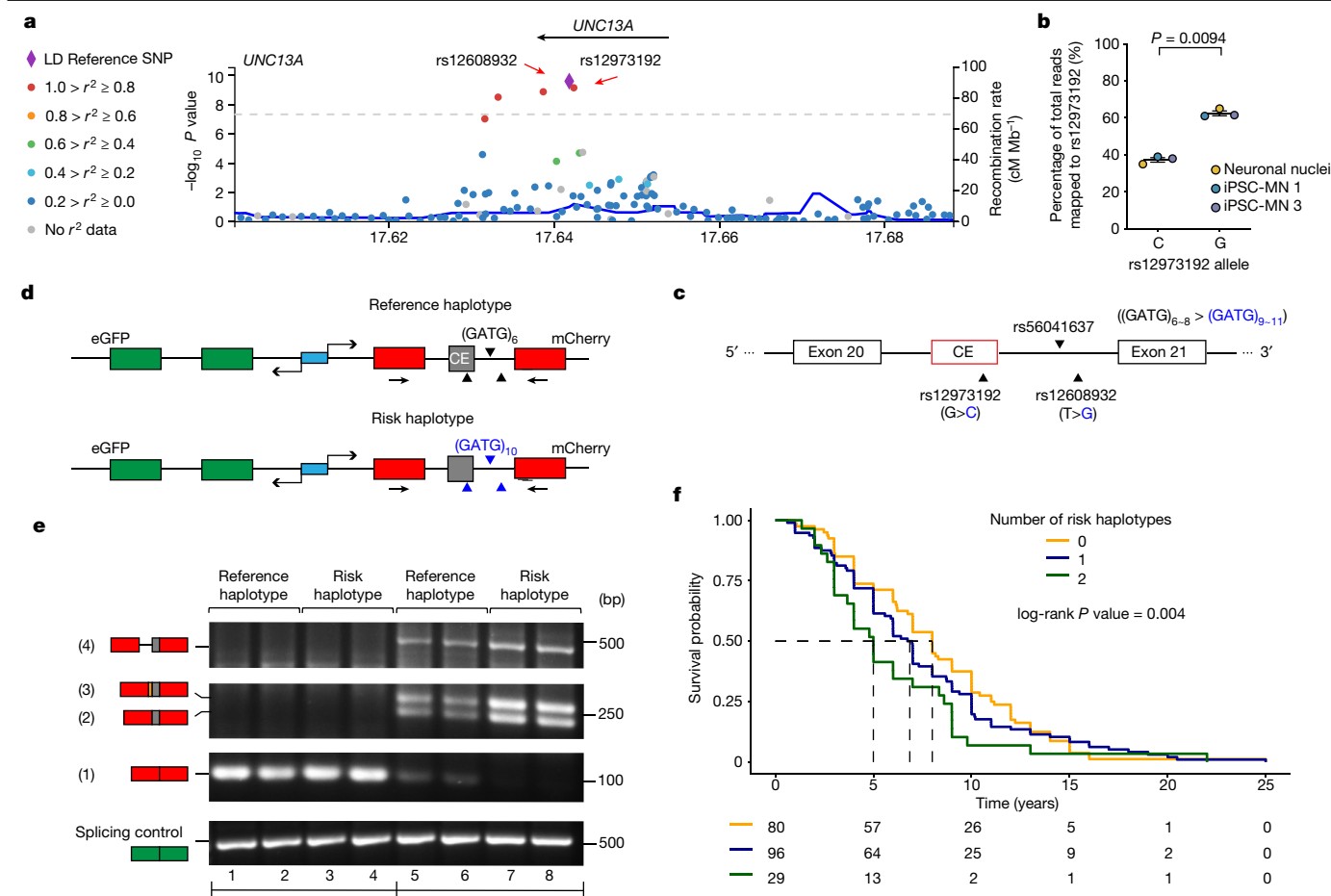

**Fig. 4 | *UNC13A* risk haplotype associated with ALS or FTD susceptibility potentiates CE inclusion when TDP-43 is dysfunctional. a**, LocusZoom plot showing SNPs associated with ALS or FTD in *UNC13A*. SNPs are coloured on the basis of levels of linkage equilibrium; SNPs rs12608932 and rs12973192 are in strong linkage disequilibrium (LD). **b**, There is a higher percentage inclusion of the risk allele (G) at rs12973192 in the *UNC13A* splice variant (*n* = 3 biologically independent samples; two-sided paired *t*-test; mean ± s.e.m.). Quantification in Extended Data Fig. 7c. **c**, Location of rs56041637 relative to the two known FTD–ALS GWAS hits and *UNC13A* CE. **d**, Design of *UNC13A* CE minigene reporter constructs and location of the primer pair used for RT–PCR. Black (reference alleles) and blue (risk alleles) triangles represent the genetic variants as shown in **c**. **e**, Splicing of minigene reporters was assessed in wild-type (WT) and TDP-43⁻/⁻ HEK 293T cells. In addition to the inclusion of CE (2), some splice variants

showed inclusion of one of the other two cryptic splicing products (3 and 4) (Extended Data Figs. 1a–e, 3a, Supplementary Note 2). The risk haplotype-carrying minigene showed an almost complete loss of canonical splicing product (1) and an increase in alternatively spliced products (2, 3 and 4). *n* = 2 independent cell culture experiments for each condition. **f**, Top, survival curves of FTLD-TDP patients stratified on the basis of the number of risk haplotypes. Heterozygous (1) and homozygous (2) patients had shorter survival time after disease onset (*n* = 205, Mayo Clinic Brain Bank; score (log-rank) test, *P* = 0.004). Dashed lines mark median survival for each genotype. Risk haplotype effect is modelled additively using Cox multivariable analysis adjusted for genetic mutations, sex and age at onset. Bottom, risk table. Summary results of the analysis are shown in Extended Data Fig. 10b.

rs56041637 is a CATC-repeat insertion in intron 20–21. Most frequently there are six CATC repeats in this region. In the patient dataset, we observed that patients who are homozygous for the risk alleles at both rs12608932 and rs12973192 tend to have 3 to 5 additional CATC repeats; patients who are homozygous for reference alleles at both rs12608932 and rs12973192 tend to have only 0 to 2 additional repeats (Fig. 4c; CATC is shown as GATG because *UNC13A* is on the reverse strand). Thus, in addition to the two lead GWAS SNPs (rs12608932 and rs12973192), we now nominate rs56041637 as potentially contributing to risk for disease by making *UNC13A* more vulnerable to CE inclusion when TDP-43 is depleted from the nucleus.

To directly test whether these three variants in *UNC13A*—which are part of the risk haplotype—increase CE inclusion upon TDP-43 depletion, we synthesized minigene reporter constructs (Fig. 4d). The reporter uses a bidirectional promoter to co-express eGFP containing a canonical intron and mCherry that is interrupted by *UNC13A* intron 20–21 from either the reference haplotype or the risk haplotype.

Because *UNC13A* is on the reverse strand, the reference alleles and the risk alleles are the reverse complement of the genotypes reported on dbSNP—for example, in intron 20–21 of *UNC13A*, the reference allele at rs12973192 is G and the corresponding risk allele is C. We transfected wild-type and TDP-43-knock-out (TDP43⁻/⁻) HEK 293T cells[36] with each minigene reporter construct. Using RT–PCR, we found that both versions of intron 20–21 were efficiently spliced out in wild-type cells (Fig. 4e, lanes 1–4). However, in TDP43⁻/⁻ cells there was a decrease in completely intron-free splicing products and a concomitant increase in cryptic splicing products (Fig. 4e, lanes 5–6). Of note, in TDP-43⁻/⁻ cells transfected with the minigene construct harbouring the risk haplotype in the intron, there was an even greater decrease in complete intron 20–21 splicing, and a proportional increase in cryptic splicing products (Fig. 4e, lanes 7–8; see Supplementary Note 2). The transcript levels of the eGFP control remained constant across all conditions, verifying equal reporter expression levels and the integrity of the splicing machinery independent of TDP-43.

To define the effect of each individual risk variant on splicing, we generated six additional minigene reporters, each carrying a different combination of the reference and risk alleles (individually, two at a time, or all three). Using RT–qPCR, we found that the risk variant in CE (rs12973192) had the strongest effect on CE inclusion in cells lacking TDP-43. The other variants also contributed to mis-splicing, but in a non-additive way, and the largest effect was with the construct harbouring all three risk variants (Extended Data Fig. 8). Expression of full-length TDP-43 rescued the splicing defects, whereas an RNA binding-deficient mutant did not (Extended Data Fig. 9). Together, these results provide direct functional evidence that TDP-43 regulates splicing of *UNC13A* intron 20–21 and that genetic variants associated with ALS and FTD susceptibility in humans potentiate cryptic exon inclusion when TDP-43 is dysfunctional.

To directly test whether the risk variants affected TDP-43 binding, we performed quantitative electrophoretic mobility shift assays (EMSA). We incubated radioactively labelled RNA probes containing reference or risk versions of sequences within the *UNC13A* CE or intronic region with recombinant full-length TDP-43. This resulted in a mobility shift, indicating that TDP-43 can bind to these sequences (Extended Data Fig. 10a). TDP-43 binds the reference and risk versions of the CE probe with similar affinity but the intronic risk allele results in a minor reduction in affinity (Extended Data Fig. 10a). TDP-43 had lower affinity for the probe containing the risk allele at rs56041637 compared with the probe containing the reference allele (Fig. 4g). Overall, TDP-43 has a much higher affinity for the intron sequence compared with the exon or repeat sequence. The diminished binding affinity of TDP-43 to risk alleles of the intron and repeat sequence may contribute to the increased cryptic splicing found in ALS and FTD. We note that these in vitro binding results are somewhat different from those reported in the accompanying Article by Brown and colleagues[37]. Our use of full-length TDP-43, which can be prone to aggregation (although our maltose-binding protein (MBP)-tagged TDP-43 is soluble) but contains additional domains that are important for TDP-43 function, differs from the one used by Brown and colleagues, which contains only the TDP-43 RNA-recognition motifs (RRMs). Future studies will be required to explore how TDP-43 regulates the cryptic splicing of *UNC13A* and other splicing targets and the effect of different genetic variations on TDP-43 binding in vivo.

To examine whether these SNPs affect survival in patients with FTLD-TDP, we evaluated the association of the risk haplotype with survival time after disease onset using data from the Mayo Clinic Brain Bank ($n = 205$). Using Cox multivariable analysis adjusting for other factors known to influence survival (genetic mutations, sex and age at onset), the risk haplotype was associated with survival time under an additive model (Fig. 4f). The number of risk haplotypes an individual carries was a strong prognostic factor (Extended Data Fig. 10b). The association remained significant under a dominant model (Extended Data Fig. 10c, d) and a recessive model (Extended Data Fig. 10e, f), indicating that carrying the risk haplotype reduces patient survival time after disease onset, consistent with previous analyses[27–30]. Thus, as suggested by previous studies[28,30,38], genetic variants in *UNC13A* that increase cryptic exon inclusion are associated with decreased survival in patients.

Here we have found that TDP-43 regulates a cryptic splicing event in the FTD–ALS risk gene *UNC13A*. The most significant genetic variants associated with disease risk are located within the intron harbouring the cryptic exon itself. Brain samples from patients with FTLD-TDP carrying these SNPs exhibited more *UNC13A* CE inclusion than those from patients with FTLD-TDP lacking the risk alleles. These risk alleles seem insufficient to cause CE inclusion because CE is not detected extensively in RNA-seq data from healthy control samples[16] (GTEx) and our functional studies indicate that TDP-43 dysfunction is required for substantial CE inclusion. Instead, the *UNC13A* risk alleles exert a TDP-43 loss-of-function-dependent disease-modifying effect.

We propose that *UNC13A* risk alleles might act as a kind of Achilles' heel, not causing problems until TDP-43 becomes dysfunctional. The discovery of a novel TDP-43-dependent cryptic splicing event in a bona fide FTD–ALS risk gene opens up a multitude of new directions for validating *UNC13A* as a biomarker and therapeutic target in ALS and FTD. This cryptic exon inclusion event—similar to that of *STMN2*[3,4]—is not conserved in mouse, so will require studies in human neuron models to test whether blocking *UNC13A* cryptic splicing is sufficient to rescue phenotypes associated with loss of TDP-43 function. It is possible that a full rescue of TDP-43 function will require restoration of more than one cryptic splicing target (for example, *STMN2*, *UNC13A* and perhaps some of the others (Fig. 1b)). But the human genetics data (Fig. 4f) showing a dose-dependent decrease in survival in individuals carrying *UNC13A* risk alleles indicate that *UNC13A* is a key target of TDP-43. We note that *UNC13A* is more highly expressed in the frontal cortex (transcripts per million (TPM) = 530.2) than in the spinal cord[16] (TPM = 35.54). One picture that might emerge is that the cryptic target *STMN2* could have a key role in lower motor neurons in the spinal cord, whereas *UNC13A* could have a key role in the brain. Perhaps some combination of effects could contribute to ALS or FTD. It remains unknown why TDP-43 pathology is associated with ALS, FTLD-TDP, or even other aging-related neuropathological changes[39]. TDP-43-dysfunction-related cryptic splicing plays out across the diverse regional and neuronal landscape of the human brain. It is tempting to speculate that in addition to *STMN2* and *UNC13A*, there could be specific portfolios of other important cryptic exon splicing events (and genetic variations that increase or decrease susceptibility to some of these events) that contribute to heterogeneity in clinical manifestation of TDP-43 dysfunction.

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

# Methods

All materials used in this study are available upon request.

## RNA-seq alignment and splicing analysis
The detailed pipeline v2.0.1 for RNA-seq alignment and splicing analysis is available on https://github.com/emc2cube/Bioinformatics/sh_RNAseq.sh. FASTQ files were downloaded from the Gene Expression Omnibus (GEO) database (GSE126543). Adaptors in FASTQ files were removed using trimmomatic (0.39) (ILLUMINACLIP:TruSeq3-PE.fa:2:30:10 LEADING:3 TRAILING:3 SLIDINGWINDOW:4:15 MINLEN:36). The quality of the resulting files was evaluated using FastQC (v0.11.9). RNA-seq reads were mapped to the human (hg38) using STAR v2.7.3a following ENCODE standard options, read counts were generated using RSEM v1.3.1, and differential expression analysis was performed in R v4.0.2 using the DESeq2 package v1.28.1[40].

## Splicing analysis
**MAJIQ.** Alternative splicing events were analysed using MAJIQ (2.2) and VOILA[13]. In brief, uniquely mapped, junction-spanning reads were used by MAJIQ with the following parameters: 'majiq build -c config–min-intronic-cov 1–simplify', to construct splice graphs for transcripts by using the UCSC transcriptome annotation (release 82) supplemented with de novo detected junctions. Here, de novo refers to junctions that were not in the UCSC transcriptome annotation but had sufficient evidence in the RNA-seq data (–min-intronic-cov 1). Distinct local splice variations (LSVs) were identified in gene splice graphs, and the MAJIQ quantifier 'majiq psi' estimated the fraction of each junction in each LSV, denoted as percent spliced in (PSI or $\Psi$), in each RNA-seq sample. The changes in each junction's PSI ($\Delta$PSI or $\Delta\Psi$) between the two conditions (TDP-43-positive neuronal nuclei versus TDP-43-negative neuronal nuclei) were calculated by using the command 'majiq deltapsi'. The gene splice graphs and the posterior distributions of PSI and $\Delta$PSI were visualized using VOILA.

**LeafCutter.** LeafCutter is available as commit 249fc26 on https://github.com/davidaknowles/leafcutter. Using RNA-seq reads aligned as previously described, reads that span exon–exon junctions and map with a minimum of 6 nt into each exon were extracted from the alignment (bam) files using filter_cs.py with the default settings. Intron clustering was performed using the default settings in leafcutter_cluster.py. Differential excision of the introns between the two conditions (TDP-43-positive neuronal nuclei versus TDP-43-negative neuronal nuclei) were calculated using leafcutter_ds.R.

## Sashimi plot
RNA-seq densities along the exons were plotted using the sashimi_plot function included in the MISO package (misopy 0.5.4). In the sashimi plot, introns are scaled down by a factor of 15 and exons are scaled down by a factor of 5. RNA-seq read densities across exons are scaled by the number of mapped reads in the sample and are measured in RPKM units. Slight modifications were made to plot_gene.py and plot_settings.py within the sashimi_plot directory of the MISO package to highlight the RNA-seq density plot. The modified sashimi_plot directory is available at (https://github.com/rosaxma/TDP-43-UNC13A-2021).

## Cell culture
SH-SY5Y (ATCC) cells were grown in DMEM/F12 media supplemented with Glutamax (Thermo Scientific), 10% fetal bovine serum and 10% penicillin–streptomycin at 37 °C, 5% $CO_2$. For shRNA treatments, cells were plated on day 0, transduced with shRNA on day 2 followed by media refresh on day 3, and collected for readout (RT–qPCR and immunoblotting) on day 6. HEK 293T TDP-43 knockout cells and parent HEK 293T cells were generated as described[36]. The cells were cultured in DMEM medium (Gibco 10564011) supplemented with 10% fetal bovine serum,

1% penicillin–streptomycin, 2 mM L-glutamine (Gemini Biosciences) and 1× MEM non-essential amino acids solution (Gibco) at 37 °C, 5% $CO_2$.

## iPS cell maintenance and differentiation into iPSC-MNs
iPS cell lines were obtained from public biobanks (GM25256-Corriell Institute; NDS00262, NDS00209-NINDS) and maintained in mTeSR1 media (StemCell Technologies) on Matrigel (Corning). iPS cells were fed daily and split every 4–7 days using ReLeSR (StemCell Technologies) according to the manufacturer's instructions. Differentiation of iPS cells into motor neurons was carried out as previously described[41]. In brief, iPS cells were dissociated and placed in ultra-low adhesion flasks (Corning) to form 3D spheroids in media containing DMEMF12/Neurobasal (Thermo Fisher), N2 Supplement (Thermo Fisher), and B-27 Supplement-Xeno free (Thermo Fisher). Small molecules were added to induce neuronal progenitor patterning of the spheroids, (LDN193189, SB-431542, Chir99021), followed by motor neuron induction (with retinoic acid, Smo agonist and DAPT). After 14 days, neuronal spheroids were dissociated with Papain and DNAse (Worthington Biochemical) and plated on poly-D-lysine/laminin coated plates in Neurobasal Medium (Thermo Fisher) containing neurotrophic factors (BDNF, GDNF and CNTF; R&D Systems). For viral transductions, neuronal cultures were incubated for 18 h with media containing lentivirus particles for shScramble, or shTDP-43. Infection efficiency of over 90% was assessed by RFP expression. Neuronal cultures were analysed for RNA and protein 7 days post transduction.

## shRNA cloning, lentiviral packaging, and cellular transduction for detecting the *UNC13A* splice variant
shRNA sequences were originated from the Broad GPP Portal (TDP-43: AGATCTTAAGACTGGTCATTC, scramble: GATATCGCTTCTACTAG-TAAG). To clone, complementary oligonucleotides were synthesized to generate 4-nt overhangs, annealed and ligated into pRSITCH (Tet inducible U6) or pRSI16 (constitutive U6) (Cellecta). Ligations were transformed into Stbl3 chemically competent cells (Thermo Scientific) and grown at 30 °C. Large scale plasmid generation was performed using Maxiprep columns (Promega), with purified plasmid used as input for lentiviral packaging with second generation packaging plasmids psPAX2 and pMD2.G (Cellecta), transduced with Lipofectamine 2000 (Invitrogen) in Lenti-X 293T cells (Takara). Viral supernatant was collected at 48 and 72 h post transfection and concentrated using Lenti-X Concentrator (Takara). Viral titer was established by serial dilution in relevant cell lines and readout of percentage of BFP+ cells by flow cytometry, with a dilution achieving a minimum of 80% BFP+ cells selected for experiments.

## Immunoblotting
SH-SY5Y cells and iPSCs-MNs were transfected and treated as above before lysis. Cells were lysed in ice-cold RIPA buffer (Sigma-Aldrich R0278) supplemented with a protease inhibitor cocktail (Thermo Fisher 78429) and phosphatase inhibitor (Thermo Fisher 78426). After pelleting lysates at maximum speed on a table-top centrifuge for 15 min at 4 °C, bicinchoninic acid (Invitrogen 23225) assays were conducted to determine protein concentrations. 60 μg (SH-SY5Y) and 30 μg (iPSCs-MNs) protein of each sample was denatured for 10 min at 70 °C in LDS sample buffer (Invitrogen NP0008) containing 2.5% 2-mercaptoethanol (Sigma-Aldrich). These samples were loaded onto 4–12% Bis–Tris gels (Thermo Fisher NP0335BOX) for gel electrophoresis, then transferred onto 0.45-μm nitrocellulose membranes (Bio-Rad 162-0115) at 100 V for 2 h using the wet transfer method (Bio-Rad Mini Trans-Blot Electrophoretic Cell 170-3930). Membranes were blocked in Odyssey Blocking Buffer (LiCOr 927-40010) for 1 h then incubated overnight at room temperature in blocking buffer containing antibodies against UNC13A (1:500, Proteintech 55053-1-AP), TDP-43 (1:1,000, Abnova H00023435-M01), or GAPDH (1:1,000, Cell Signaling Technologies 5174S). Membranes were subsequently incubated in blocking buffer containing horseradish peroxidase (HRP)-conjugated anti-mouse IgG

(H+L) (1:2,000, Fisher 62-6520) or HRP-conjugated anti-rabbit IgG (H+L) (1:2,000, Life Technologies 31462) for 1 h. ECL Prime kit (Invitrogen) was used for development of blots, which were imaged using ChemiDox XRS+ System (Bio-Rad). The intensity of bands was quantified using Fiji, and then normalized to the corresponding controls.

## RNA extraction, cDNA synthesis and RT–qPCR or RT–PCR for detecting the *UNC13A* splice variant in iPSC-MNs

Total RNA was extracted using RNeasy Micro kit (Qiagen) per manufacturer's instructions, with lysate passed through a QIAshredder column (Qiagen) to maximize yield. RNA was quantified by Nanodrop (Thermo Scientific), with 75 ng used for cDNA synthesis with SuperScript IV VILO Master Mix (Thermo Scientific). Quantitative PCR was run with 6 ng cDNA input in a 20 µl reaction using PowerTrack SYBR Green Master Mix (Thermo Scientific) with readout on a QuantStudio 6 Flex using standard cycling parameters (95 °C for 2 min, 40 cycles of 95 °C for 15s and 60 °C for 60 s), followed by standard dissociation (95 °C for 15 s at 1.6 °C s$^{-1}$, 60 °C for 60 s at 1.6 °C s$^{-1}$, 95 °C for 15 s at 0.075 °C s$^{-1}$). $\Delta\Delta C_t$ was calculated with the housekeeper gene *RPLPO* as control and relevant shScramble as reference; measured $C_t$ values greater than 40 were set to 40 for visualizations. See Supplementary Table 6 for primers.

PCR was conducted with 15 ng cDNA input in a 100 µl reaction using NEBNext Ultra II Q5 Master Mix (New England Biolabs), with the following cycling parameters: initial denaturation: 98 °C for 30 s; 40 cycles: 98 °C for 10 s, 64 °C for 30 s, 72 °C for 20 s; final extension: 72 °C for 2 min. The resulting products were visualized on a 1.5% TAE gel. See Supplementary Table 6 for primers.

## Human iPS cell-derived neurons for detecting *UNC13A* splice variants

cDNA was available from CRISPRi-i³Neuron iPS cells (i³N) generated from our previous publication[11], in which TDP-43 is downregulated to about 50%. RT–qPCR was performed using SYBR GreenER qPCR SuperMix (Invitrogen). Samples were run in triplicate, and RT–qPCR reactions were run on a QuantStudio 7 Flex Real-Time PCR System (Applied Biosystems). Relative quantification was determined using the $\Delta\Delta C_t$ method and normalized to the endogenous controls *RPLPO* and *GAPDH*. We normalized relative transcript levels for wild-type *UNC13A* to that of the neurons treated with control sgRNA (mean set to 1). See Supplementary Table 6 for primers.

## Cell culture for validating additional splicing events in iPS cell-derived neurons

We used an induced neuron (iN) system previously established for rapidly differentiating human iPS cells into functional cortical neurons[42]. In brief, iPS cells (without disease mutation) were cultured using feeder-free conditions on Matrigel (Fisher Scientific CB-40230) using mTeSR1 media (Stemcell Technologies 85850). Cells were transduced with a Tet-On induction system that allows expression of the transcription factor NGN2. Cells were dissociated on day 3 of differentiation and replated on Matrigel-coated tissue culture plates in Neurobasal Medium (Thermo Fisher) containing neurotrophic factors, BDNF and GDNF (R&D Systems) with viral transductions for shScramble or shTDP-43. RNA and protein were extracted 7 days after transduction.

## shRNA cloning, lentiviral packaging, and cellular transduction for validating additional splicing events

The lentiviral plasmid targeting *TARDBP* (Millipore-Sigma TRCN0000016038) and Scramble (CAACAAGATGAAGAGCACCAA) were packaged using third generation packaging plasmids (pMDLg/pRRE, pRSV-Rev, pMD2.G) and transduced with Lipofectamine 3000 (Invitrogen) into HEK 293T cells cultured under standard conditions (DMEM, 10% FBS, penicillin–streptomycin). Viral supernatant was collected at 48 and 72 h post-transfection and concentrated 1:100 using Lenti-X Concentrator (Takara).

## RNA extraction, cDNA synthesis and RT–qPCR for validating additional splicing events

Total RNA was extracted using RNeasy Micro kit (Qiagen) and reverse transcribed into cDNA using High-Capacity cDNA Reverse Transcription Kits (Invitrogen). Quantitative PCR was run with 2 ng cDNA input in a 10 µl reaction using PowerTrack SYBR Green Master Mix (Thermo Scientific) with readout on a QuantStudio 6 Flex using standard cycling parameters. $\Delta\Delta C_t$ was calculated with *RPLPO* or *GAPDH* as housekeeper gene controls and relevant shScramble transduced condition as reference; measured $C_t$ values greater than 40 were set to 40 for visualizations. See Supplementary Table 6 for primers used for detecting mis-spliced transcripts and normal splicing transcripts, and primers used for internal controls.

## Amplicon sequencing of the splice variants

Splice variants in iPSC-MNs were established by PCR amplification from UNC13A exon 19 to exon 21 (UNC13A_19_21 FWD 5′–3′= CAACCTGG ACAAGCGAACTG, UNC13A_19_21 RVS 5′–3′= GGGCTGTCTCATCGTAG TAAAC). Resulting products were purified using Wizard SV Gel and PCR Clean-Up columns (Promega) and submitted for NGS (Amplicon EZ, Genewiz). Adaptors in FASTQ files were removed using trimmomatic (0.39) (ILLUMINACLIP:TruSeq3-PE.fa:2:30:10 LEADING:3 TRAILING:3 SLIDINGWINDOW:4:15 MINLEN:36). The quality of the resulting files was then evaluated using FastQC (v0.11.9). The sequencing reads were then mapped to the human (hg38) using STAR v2.7.3a following ENCODE standard options. Uniquely mapped reads were then filtered for using the command 'samtools view -b -q 255'. The Sashimi Plot were then generated using the sashimi plot function in IGV (2.8.0) with the minimum junction coverage set to 20.

## Post-mortem brain tissues for detecting *UNC13A* splice variant

Post-mortem brain tissues from patients with FTLD-TDP and cognitively normal control individuals were obtained from the Mayo Clinic Florida Brain Bank. Diagnosis was independently ascertained by trained neurologists and neuropathologists upon neurological and pathological examinations, respectively. Written informed consent was given by all participants or authorized family members and all protocols were approved by the Mayo Clinic Institution Review Board and Ethics Committee. Complementary DNA (cDNA) obtained from 500 ng of RNA (RIN ≥ 7.0) from medial frontal cortex was available from a previous study, as well as matching pTDP-43 data from the same samples[43]. Following standard protocols, RT–qPCR was conducted using SYBR GreenER qPCR SuperMix (Invitrogen) for all samples in triplicates. RT–qPCR reactions were run in a QuantStudio 7 Flex Real-Time PCR System (Applied Biosystems). Relative quantification was determined using the $\Delta\Delta C_t$ method and normalized to the endogenous controls *RPLPO* and *GAPDH*. We normalized relative transcript levels to that of the healthy controls (mean set to 1). See Supplementary Table 6 for primers.

## Quantification of *UNC13A* splice variants in bulk RNA sequencing

RNA-seq data generated by NYGC ALS Consortium cohort were downloaded from the NCBI Gene Expression Omnibus (GEO) database (GSE137810, GSE124439, GSE116622 and GSE153960). We used the 1658 available and quality-controlled samples classified as described[11]. After pre-processing and aligning the reads to human (hg38) as described previously, we estimated the expression of the full-length UNC13A using RSEM (v1.3.2). PCR duplicates were removed using MarkDuplicates from Picard Tools (2.23.0) using the command 'MarkDuplicates REMOVE_DUPLICATES=true CREATE_INDEX=true'. We then filtered for uniquely mapped reads using the command 'samtools view -b -q 255'. Reads that span either exon 19–exon 20 junction, exon 20–CE junction, CE–exon 21 junction or exon 20–exon 21 junction were quantified using bedtools (2.27.1) using the command 'bedtools intersect -split'. Because of the relatively low level of expression of *UNC13A* in

post-mortem tissues and the heterogeneity of the tissues, it is possible that not all tissues have enough detectable *UNC13A* for us to detect the splice variants. Since *UNC13A* contains more than 40 exons and RNA-seq coverages of mRNA transcripts are often not uniformly distributed[44], we looked at reads spanning the exon 19–exon 20 junction, which is included in both the canonical isoform variant and the splice variant, and there is a strong correlation (Pearson's *r* = 0.99) between the numbers of reads mapped to the exon 19–exon 20 junction and the exon 20–exon 21 junction. We observed that samples that have at least 2 reads spanning either exon 20–CE junction or CE–exon 21 junction have at least either *UNC13A* TPM = 1.55 or 20 reads spanning exon 19–exon 20 junction. Therefore, we selected the 1,151 samples that had a TPM ≥ 1.55, or at least 20 reads mapped to the exon 19–exon 20 junction as samples suitable for *UNC13A* splice variant analysis.

### In situ hybridization *UNC13A* CE analysis in postmortem brain samples

**Patients and diagnostic neuropathological assessment.** Postmortem brain tissue samples used for this study were obtained from the University of California San Francisco (UCSF) Neurodegenerative Disease Brain Bank (Supplementary Table 4). Supplementary Table 4 provides demographic, clinical, and neuropathological information. Consent for brain donation was obtained from subjects or their surrogate decision makers in accordance to the Declaration of Helsinki, and following a procedure approved by the UCSF Committee on Human Research. Brains were cut fresh into 1 cm thick coronal slabs, and alternate slices were fixed in 10% neutral buffered formalin for 72 h. Blocks from the medial frontal pole were dissected from the fixed coronal slabs, cryoprotected in graded sucrose solutions, frozen, and cut into 50 μm thick sections as described previously[45]. Clinical and neuropathological diagnosis were performed as described previously[45]. Subjects were selected on the basis of clinical and neuropathological assessment. Patients selected had a primary clinical diagnosis of behavioural variant frontotemporal dementia (bvFTD) with or without amyotrophic lateral sclerosis or motor neuron disease and a neuropathological diagnosis of FTLD-TDP, type B. We excluded subjects if they had a known disease-causing mutation, post-mortem interval ≥ 24 h, Alzheimer's disease neuropathologic change > low, Thal amyloid phase > 2, Braak neurofibrillary tangle stage > 4, CERAD neuritic plaque density > sparse, and Lewy body disease > brainstem predominant[45].

**In situ hybridization and immunofluorescence.** To detect single RNA molecules, a BaseScope Red Assay kit (ACDBIO, USA) was used. One 50 μm thick fixed frozen tissue section from each subject was used for staining. Experiments were performed under RNase-free conditions as appropriate. Probes that target the transcript of interest, *UNC13A*, specific to either the mRNA (exon 20–exon 21 junction) or the cryptic exon containing spliced target (exon 20–cryptic exon junction) were used. Positive (*Homo Sapiens* PPIB) and negative (*Escherichia coli* DapB) control probes were also included. In situ hybridization was performed based on vendor specifications for the BaseScope Red Assay kit. In brief, frozen tissue sections were washed in PBS and placed under an LED grow light (HTG Supply, LED-6B240) chamber for 48 h at 4 °C to quench tissue autofluorescence. Sections were quickly rinsed in PBS and blocked for endogenous peroxidase activity. Sections were transferred on to slides and dried overnight. Slides were subjected to target retrieval and protease treatment and advanced to ISH. Probes were detected with TSA Plus-Cy3 (Akoya Biosciences), and subjected to immunofluorescence staining with antibodies to TDP-43 (rabbit polyclonal, Proteintech, RRID: AB_615042, dilution 1:4,000, catalogue (cat.) no. 10782-2-AP) and NeuN (Guinea pig polyclonal, Synaptic Systems, dilution 1:500; cat. no. 266004), and counterstained with DAPI (Life Technologies) for nuclei.

**Image acquisition and analysis.** *Z*-stack images were captured using a Leica SP8 confocal microscope with an 63× oil immersion objective (1.4 NA). For RNA probes, image capture settings were established during initial acquisition based on PPIB and DAPB signal and remained constant across *UNC13A* probes and subjects. TDP-43 and NeuN image capture settings were modified based on staining intensity differences between cases. For each case, 6 non-overlapping *Z*-stack images were captured across cortical layers 2–3. RNA puncta for the UNC13A cryptic exon were quantified using the 'analyze particle' plugin in ImageJ. In brief, all images were adjusted for brightness using similar parameters and converted to maximum intensity *Z*-projections, images were adjusted for auto-threshold (intermodes), and puncta were counted (size: 6-infinity, circularity: 0–1).

### Linkage disequilibrium analysis

Recalibrated VCF files of 297 ALS patients of European descent generated by GATK HaplotypeCallers were downloaded from Answer ALS in July 2020 (https://www.answerals.org). VCFtools (0.1.16) were used to filter for sites that are in intron 20–21. The filtered VCF files were merged using BCFtools (1.8). Since there are sites that contain more than 2 alleles, we tested for genotype independence using the chi-squared statistics by using the command 'vcftools–geno-chisq-min-alleles 2–max-alleles 8'. We found two additional SNPs, rs56041637 (*P* < 0.0001 with rs12608932, *P* < 0.0001 with rs12973192), and rs62121687 (*P* < 0.0001 with rs12608932, *P* < 0.0001 with rs12973192) that are in linkage disequilibrium with both. However, since rs62121687 was included in a GWAS and has a *P*-value[35] of 0.0186585, we excluded it from further analysis.

### Determination of rs12608932 and rs12973192 SNP genotype in human postmortem brain

Genomic DNA (gDNA) was extracted from human frontal cortex using Wizard Genomic DNA Purification Kit (Promega), according to the manufacturer's instructions. TaqMan SNP genotyping assays were performed on 20 ng of gDNA per assay, using a commercial pre-mixture consisting of a primer pair and VIC or FAM-labelled probes specific for each SNP (cat. no. 4351379, assay ID 43881386_10 for rs12608932 and 11514504_10 for rs12973192, Thermo Fisher Scientific), and run on a QuantStudio 7 Flex Real-Time PCR system (Applied Biosystems), according to the manufacturer's instructions. The PCR programs were 60 °C for 30 s, 95 °C for 10 min, 40 cycles of 95 °C for 15 s and, 60 °C (rs12973192) or 62.5 °C for 1 min (rs12608932), and 60 °C for 30 s.

### Splicing reporter assay

Minigene constructs were designed in silico, synthesized by GenScript and sub-cloned into a vector with the GFP splicing control. HEK 293T TDP-43 knockout cells and the parent HEK 293T cells were seeded into standard P12 tissue culture plates (at $1.6 \times 10^5$ cells per well), allowed to adhere overnight, and transfected with the indicated splicing reporter constructs (400 ng per well) using Lipofectamine 3000 transfection reagent (Invitrogen). Each reporter comprised one of the splicing modules (shown in Fig. 4d), which is expressed from a bidirectional promoter. Twenty-four hours after transfection, RNA was extracted from these cells using PureLink RNA Mini Kit (Life Technologies) according to the manufacturer's protocol, with on-column PureLink DNase (Invitrogen) treatment. The RNA was reverse transcribed into cDNA using the High Capacity cDNA Reverse Transcription Kit (Invitrogen) according to the manufacturers' instructions. PCRs were performed using OneTaq 2X Master Mix with Standard Buffer (NEB) with the following cycling parameters: denaturation: 94 °C for 30 s; 30 cycles: 94 °C for 20 s, 54 °C for 30 s, 68 °C for 30 s; final extension: 68 °C for 5 min on a Mastercycler Pro (Eppendorf) thermocycler PCR machine. PCR products were separated by electrophoresis on a 1.5% TAE gel and imaged ChemiDox XRS+ System (Bio-Rad). See Supplementary Table 6 for primers.

### Assay to assess the effect of variants at rs12973192, rs12608932 and rs56041637 on splicing

Additional minigene constructs shown in Extended Data Fig. 8 were either generated using site-directed mutagenesis (New England

Biolabs, E0554S) or synthesized by GenScript, and sub-cloned into the vector with the GFP splicing control. HEK 293T TDP-43 knockout cells and the parent HEK 293T cells were seeded into standard P12 tissue culture plates (at $5 \times 10^5$ cells per well), allowed to adhere overnight and transfected with the indicated splicing reporter constructs (400 ng per well) using Lipofectamine 3000 transfection reagent (Invitrogen). Twenty-four hours after transfection, RNA was extracted from these cells using PureLink RNA Mini Kit (Life Technologies) according to the manufacturer's protocol, with on-column PureLink DNase treatment. The RNA was reverse transcribed into cDNA using the High Capacity cDNA Reverse Transcription Kit (Invitrogen) according to the manufacturers' instructions. The *UNC13A* cryptic exon signal was measured using a pair of primers that detect the junction of the CE and the immediately downstream mCherry exon. The splicing of eGFP was measured using a pair of primers that detect the junction of the first and second exons of eGFP. A pair of primers that mapped within the second exon of eGFP was used to measure the transfection efficiency of the splicing reporter construct and was used as a normalizer. $\Delta\Delta C_t$ was calculated using the cryptic exon signal level or the splicing of eGFP in the HEK 293T TDP-43 knockout cells expressing the reference haplotype-carrying reporter as reference. See Supplementary Table 6 for primers.

## Rescue of UNC13A splicing using TDP-43 overexpression constructs

HEK 293T TDP-43 knockout cells and the parent (wild-type) HEK 293T cells were seeded into standard P12 tissue culture plates (at $5 \times 10^5$ cells per well), allowed to adhere overnight and transfected with the splicing reporter construct carrying the reference haplotype (400 ng per well; Fig. 4e) and the indicated TDP-43 overexpression constructs (600 ng per well) using Lipofectamine 3000 transfection reagent (Invitrogen). Twenty-four hours after transfection, RNA was extracted from these cells using PureLink RNA Mini Kit (Life Technologies) according to the manufacturer's protocol, with on-column PureLink DNase treatment. The RNA was reverse transcribed into cDNA using the High Capacity cDNA Reverse Transcription Kit (Invitrogen) according to the manufacturers' instructions. Quantitative PCR was run with 8 ng cDNA input in a 10 µl reaction using PowerTrack SYBR Green Master Mix (Thermo Scientific) with readout on a QuantStudio 6 Flex using standard cycling parameters.

The *UNC13A* cryptic exon signal was measured using a pair of primers that detect the junction of the CE and the mCherry exon immediately downstream of it. A pair of primers that are mapped within the second exon of eGFP was used to measure the transfection efficiency of the splicing reporter construct, and was used as a normalizer. $\Delta\Delta C_t$ was calculated using the cryptic exon signal level in the wild-type HEK 293T cells without TDP-43 overexpression constructs as reference. See Supplementary Table 6 for primers.

The expression levels of the overexpression constructs were measured using a pair of primers that detect the second exon of TDP-43. The primers do detect the endogenous TDP-43 but since the HEK 293T TDP-43 knockout cells do not have TDP-43 expression as shown previously[36], using the primers do not interfere with the measurement of the expression levels of TDP-43 constructs in the knockout cells. $\Delta\Delta C_t$ was calculated using the TDP-43 expression level in the HEK 293T TDP-43 knockout cells with full length TDP-43 overexpression constructs as reference. *RPLP0* and *GAPDH* were used as internal controls. See Supplementary Table 6 for primers.

## Generation of pTB *UNC13A* minigene construct

The pTB *UNC13A* minigene construct containing the human *UNC13A* cryptic exon sequence and the nucleotide flanking sequences upstream (50 bp at the of end of intron 19, the entirety of exon 20, and the entirety of intron 20 sequence upstream of the cryptic exon) and downstream (approximately 300-bp intron 20) of the cryptic exon were amplified from human genomic DNA using the following primers: FWD 5'–3', AGGTCATATGCACTGCTATAGTGGGAAGTTC and RVS 5'–3', CTTACATA TGTAATAACTCAACCACACTTCCATC; and subcloned into the NdeI site of the pTB vector. We have previously used a similar approach to study TDP-43 splicing regulation of other TDP-43 targets[46].

## Rescue of *UNC13A* splicing using the pTB minigene and TDP-43 overexpression constructs

HeLa cells were grown in Opti-MEM I Reduced Serum Medium, GlutaMAX Supplement (Gibco) plus 10% fetal bovine serum (Sigma), and 1% penicillin/streptomycin (Gibco). For double-transfection and knockdown experiments, cells were first transfected with 1.0 µg of pTB *UNC13A* minigene construct and 1.0 µg of one of the following plasmids: GFP, GFP-TDP-43 or GFP-TDP-43 5FL constructs to express GFP-tagged TDP-43 proteins have been previously described[46,47], in serum-free media and using Lipofectamine 2000 following the manufacturer's instructions (Invitrogen). Four hours following transfection, media was replaced with complete media containing siLentfect (Bio-Rad) and siRNA complexes (AllStars Neg. Control siRNA or siRNA against *TARDBP* 3' untranslated region, a region not included in the TDP-43 overexpression constructs) (Qiagen) following the manufacturer's protocol. Cycloheximide (Sigma) was added at a final concentration of $100 \ \mu g \ ml^{-1}$ at 6 h prior to collecting the cells. Then RNA was extracted from the cells using TRIzol Reagent (Zymo Research), following the manufacturer's instructions. Approximately 1 µg of RNA was converted into cDNA using the High Capacity cDNA Reverse Transcription Kit with RNA inhibitor (Applied Biosystems). The RT–qPCR assay was performed on cDNA (diluted 1:40) with SYBR GreenER qPCR SuperMix (Invitrogen) using QuantStudio7 Flex Real-Time PCR System (Applied Biosystems). All samples were analysed in triplicates. The RT–qPCR program was as follows: 50 °C for 2 min, 95 °C for 10 min, and 40 cycles of 95 °C for 15 s and 60 °C for 1 min. For dissociation curves, a dissociation stage of 95 °C for 15 s, 60 °C for 1 min and 95 °C for 15 s was added at the end of the program. Relative quantification was determined using the $\Delta\Delta C_t$ method and normalized to the endogenous controls *RPLP0* and *GAPDH*. We normalized relative transcript levels for wild-type *UNC13A* and GFP to that of the control siRNA condition (mean set to 1). See Supplementary Table 6 for primers.

## In vitro TDP-43 binding studies

**Cloning.** The plasmid encoding TDP43 as a C-terminal MBP-tagged protein (TDP43–MBP–His$_6$) was purchased from Addgene (#104480).

**Bacterial growth and protein expression.** The wild-type TDP-43 expression plasmid was transformed into *E. coli* One Shot BL21 Star (DE3) cells (ThermoFisher). Transformed *E. coli* were grown at 37 °C in 1 l of LB media supplemented with 0.2% dextrose and 50 µg ml$^{-1}$ kanamycin until absorbance at 600 nm reached 0.5–0.6. The culture was then incubated at 4 °C for 30–45 min. TDP-43 expression was induced with 1 mM IPTG for 16 h at 4 °C. Cells were collected by centrifugation.

**Recombinant TDP-43 purification.** Wild-type TDP-43–MBP was purified as described[48]. In brief, cell pellets were resuspended in lysis buffer 1 M NaCl, 20 mM Tris (pH 8.0), 10 mM imidazole, 10% glycerol and 2.5 mM 2-mercaptoethanol and supplemented with cOmplete, EDTA-free protease inhibitor cocktail tablets (Roche) then lysed via sonication. Cell lysates were centrifuged at 31,400*g* at 4 °C for 1 h, filtered, then purified with FPLC using a XK 50/20 column (Cytiva) packed with Ni-NTA agarose beads (Qiagen) which were equilibrated in lysis buffer. TDP-43 was recovered via a 0–80% gradient elution using 1 M NaCl, 20 mM TrisHCl (pH 8.0), 10 mM imidazole, 10% glycerol and 2.5 mM 2-mercaptoethanol as the base buffer and 1 M NaCl, 20 mM TrisHCl (pH 8.0), 500 mM imidazole, 10% glycerol, and 2.5 mM 2-mercaptoethanol as the elution buffer. Eluted protein was concentrated using Amicon Ultra-15 centrifugal filters, MWCO

50 kDa (Millipore), filtered and further purified with size-exclusion chromatography using a 26/600 Superdex 200 pg column (Cytiva) equilibrated with 300 mM NaCl, 20 mM TrisHCl (pH8.0) and 1 mM DTT. The second out of three peaks, as evaluated by absorbance at 280 nm, was collected, spin concentrated as before, aliquoted, flash frozen in liquid $N_2$, and stored at −80 C until further use. Protein concentrations were determined using absorbance at 280 nm (Nanodrop) and purity was determined by running samples on a 4–20% SDS–PAGE gel and visualized with Coomassie stain.

**Electrophoresis mobility shift assay**

EMSA was used to compare TDP-43 binding to the reference and risk RNA sequences for reference and risk alleles of CE (rs12973192), intron (rs12608932), and repeat sequences (rs56041637) (see Supplementary Table 5). Increasing TDP-43 concentrations ranging from 0–4 mM were incubated with a constant 1 nM concentration of RNA in buffer (50 mM Tris-HCl, pH 7.5, 100 mM KCl, 2 mM MgCl2, 100 mM β−mercaptoethanol, 0.1 mg ml$^{-1}$ BSA) for 30 min at room temperature. RNA is dual-labelled (Cy3 and Cy5) and contains an 18-nucleotide partial duplex on the 3′ end. Reactions were mixed with loading dye and run on a 6% non-denaturing polyacrylamide gel and imaged using fluorescence mode (Cy5) on a Typhoon scanner. Bound fractions were determined using the Analyze Gel plugin in ImageJ and normalized to the total intensity per lane. Apparent binding affinities were calculated using the 'Specific binding with Hill slope' function in Graphpad.

**Statistical methods**

Survival curves were compared using the coxph function in the survival (3.1.12) R package, which fits a multivariable Cox proportional hazards model that contains sex, reported genetic mutations and age at onset, and performs a score (log-rank) test. Effect sizes are reported as the hazard ratios. Proportional Hazards assumptions were tested using cox.zph function. The survival curves were plotted using ggsurvplot in suvminer (v.0.4.8) R package. Linear mixed effects models were analysed using lmerTest R package (3.1.3). Statistical analyses were performed using R (version 4.0.0), or Prism 8 (GraphPad), which were also used to generate graphs.

**Reporting summary**

Further information on research design is available in the Nature Research Reporting Summary linked to this paper.

## Data availability

The amplicon sequencing data has been deposited in the Gene Expression Omnibus (GEO) at GSE182976. RNA-seq data for splicing analysis is available at GSE126543. RNA-seq data generated by NYGC ALS Consortium cohort is available at GSE137810, GSE124439, GSE116622 and GSE153960. Source data are provided with this paper.

## Code availability

All codes used in this study are available at (https://github.com/rosaxma/TDP-43-UNC13A-2021) and (https://doi.org/10.5281/zenodo.5770954).

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

**Acknowledgements** This work was supported by NIH grants R35NS097263(10) (A.D.G.), R35NS097273(17) (L.P.), U54NS123743 (A.D.G., L.P., M.P. and W.W.S.), P01NS084974 (L.P.), R01NS104437 (W.W.S.), RF1NS120992 (M.P.), F32 NS116208-02 (C.M.R.), T32 GM007231 (G. Mekonnen), 2T32AG047126-06A1 (T.A.), F32GM139268 (L.G.), RF1AG071326 (S. Myong.), RF1NS113636 (S. Myong), the Robert Packard Center for ALS Research at Johns Hopkins (L.P., J.S. and A.D.G.), P30AG06267 (R.C.P.), U01AG006786 (R.C.P.), 2T32HG000044-21 NIHGRI training grant (X.R.M.), the Brain Rejuvenation Project of the Wu Tsai Neurosciences Institute (A.D.G.), Target ALS (J.S.), Amyotrophic Lateral Sclerosis Association (J.S.), and the Office of the Assistant Secretary of Defense for Health Affairs through the Amyotrophic Lateral Sclerosis Research Program (W81XWH-20-1-0242 to J.S.). G.K. is supported by a fellowship from the Stanford Knight-Hennessy Scholars Program. A.B. is supported by a Fulbright Future Scholarship. Y.K. is supported by Milton Safenowitz Postdoctoral Fellowship Program from the Amyotrophic Lateral Sclerosis Association (21-PDF-582). S.P. is supported by a BrightFocus ADR Grant (A2020279F). T.A. is supported by a fellowship from the Takeda Science Foundation. The UCSF Neurodegenerative Disease Brain Bank receives funding support from NIH grants P30AG062422, P01AG019724, U01AG057195 and U19AG063911, as well as the Rainwater Charitable Foundation and the Bluefield Project to Cure FTD. Some of the computing for this project was performed on the Sherlock cluster. We would like to thank Stanford University and the Stanford Research Computing Center for providing computational resources and support that contributed to these research results.

**Author contributions** X.R.M., M.P., L.P. and A.D.G. designed the experiments. X.R.M. and A.D.G. wrote the paper. X.R.M., M.P., Y.K. performed experiments and analysed data. S.C.V. and W.W.S. performed and analysed the *UNC13A* BaseScope experiments on patient samples. G.K. performed the UNC13A immunoblotting experiments. C.M.R. helped perform RT–PCR analyses. F.H., D.W.W., K.K., G.M., S.M., N.S. and E.M.G. performed TDP-43 knockdown experiments and analysed data. A.B. and H.B.S. performed some of the minigene reporter assays. C.G. analysed additional TDP-43 target genes. T.A. generated and analysed induced neurons for TDP-43 knockdown experiments. B.B.C. helped with *UNC13A* human genetics analyses. G.M., L.G., J.D.R., J.S. and S.M. performed TDP-43 RNA-binding studies and analysed data. K.J-W., C.N.C. and S.P. generated TDP-43 splicing reporter constructs. B.O., N.R.G-R., B.F.B., D.S.K., R.C.P. and D.W.D. contributed patient samples.

**Competing interests** A.D.G. is a scientific founder of Maze Therapeutics. X.R.M. served as a consultant for Maze Therapeutics. M.P. and L.P. serve as consultants for Target ALS. F.H., B.B.C., D.W.W., K.K., G. Miller, S. Mekhoubad, N.S. and E.G. are employees of Maze Therapeutics, which has filed a patent (63/171,522) on methods to modulate splicing of *UNC13A*.

**Additional information**
**Correspondence and requests for materials** should be addressed to Leonard Petrucelli or Aaron D. Gitler.

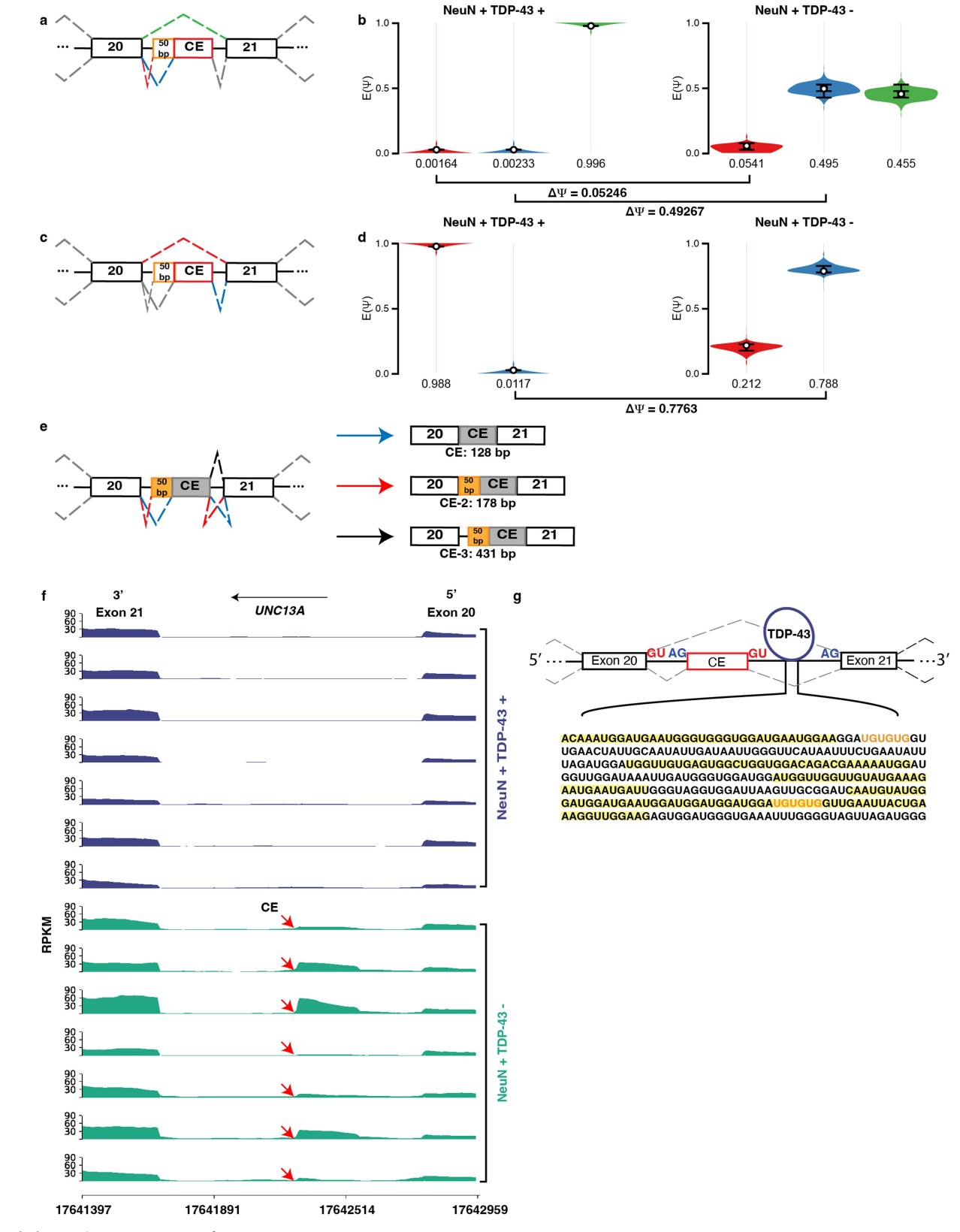

**Extended Data Fig. 1** | See next page for caption.

**Extended Data Fig. 1 | Splicing analysis using MAJIQ demonstrates inclusion of cryptic exon between exon 20 and exon 21 of *UNC13A*.**
**(a, b)** Depletion of TDP-43 introduces two alternative 3′ splicing acceptors in the intron 20–21: one is near chr19:17642591($\Delta\Psi = 0.05246$) and the other one is near chr19:17642541($\Delta\Psi = 0.49267$). **(c, d)** An alternative 5′ splicing donor is also introduced near chr19:17642414 ($\Delta\Psi = 0.7763$). See Supplementary Note 2. **(a, c)** Splice graphs showing the inclusion of the cryptic exon (CE) between exon 20 and exon 21 of *UNC13A*. **(b, d)** Violin plots corresponding to (a and c) respectively. Each violin in (b and d) represents the posterior probability distribution of the expected relative inclusion (PSI or Ψ) for the color matching junction in the splice graph. The tails of each violin represent the 10th and 90th percentile. The box represents the interquartile range with the line in the middle indicating the median. The white circles mark the expected PSI (E[Ψ]). The change in the relative inclusion level of each junction between two conditions is referred to as $\Delta\Psi$ or $\Delta PSI$[13]. **(e)** The three versions of cryptic exons resulting from the loss of TDP-43. **(f)** Visualization of RNA-sequencing alignment between exon 20 and exon 21 in UNC13A (hg38). Libraries were generated as described in Fig. 1a. CE, cryptic exon. **(g)** iCLIP for TDP-43 (from Tollervey et al.[17]) indicates that TDP-43 binds to intron 20–21. The sequence shown is an example of a region in intron 20-21 that is frequently bound by TDP-43 (shown by mapped reads from ERR039843, ERR039845 and ERR039855).

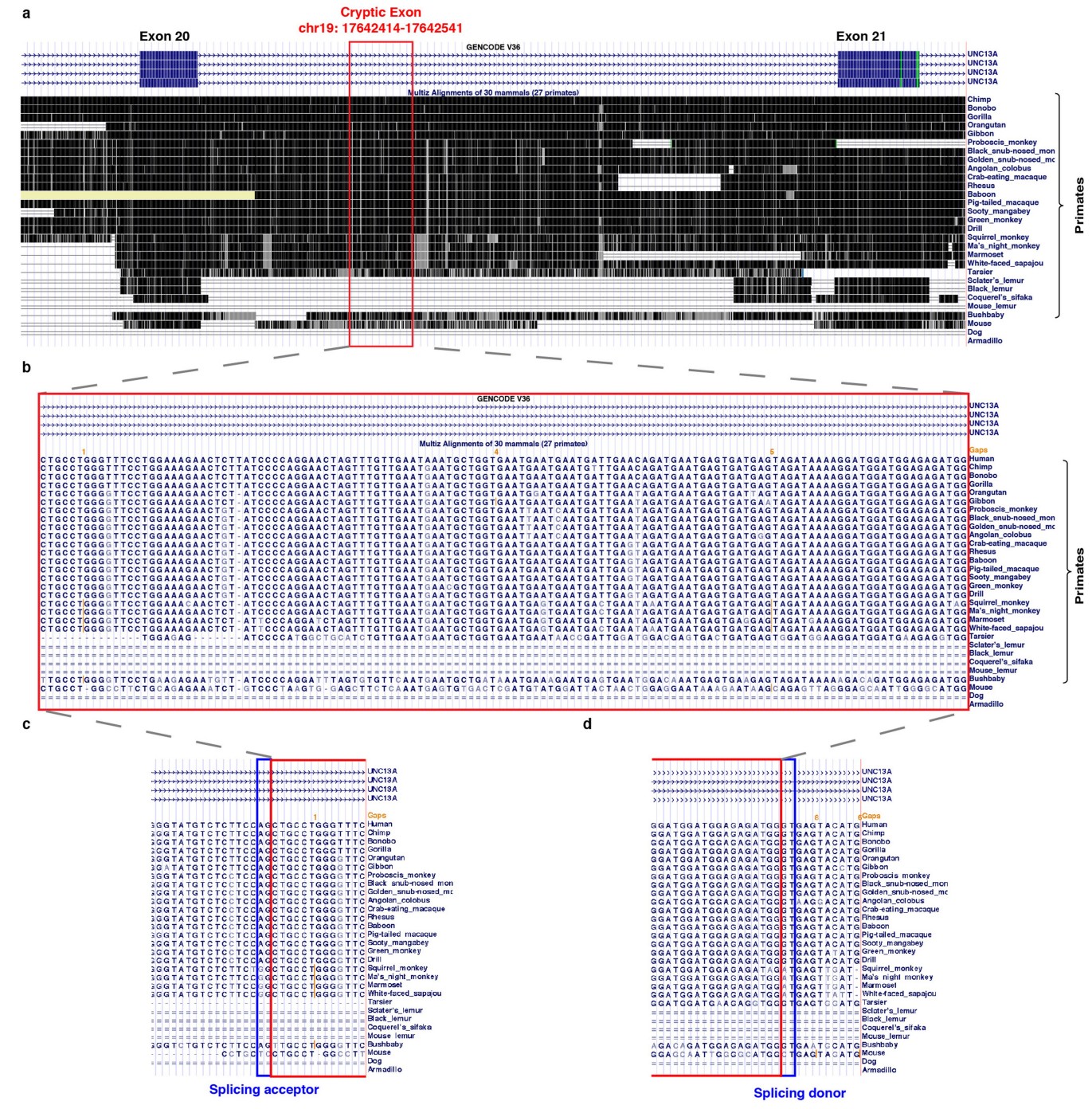

**Extended Data Fig. 2 | Intron 20-21 of *UNC13A* is conserved among most primates.** The Primates Multiz Alignment & Conservation track on UCSC[49] genome browser (http://genome.ucsc.edu) includes 30 mammals, 27 of which are primates. **(a)** Exon 20 and exon 21 of *UNC13A* is well conserved among mammals. The location of the 128 bp cryptic exon is highlighted in red.

However, intron 20-21 **(a)**, the cryptic exon **(b)**, and the splicing acceptor site (highlighted in blue) upstream of the cryptic exon **(c)** and splicing donor site (highlighted in blue) downstream of the cryptic exon **(d)** are only conserved in primates.

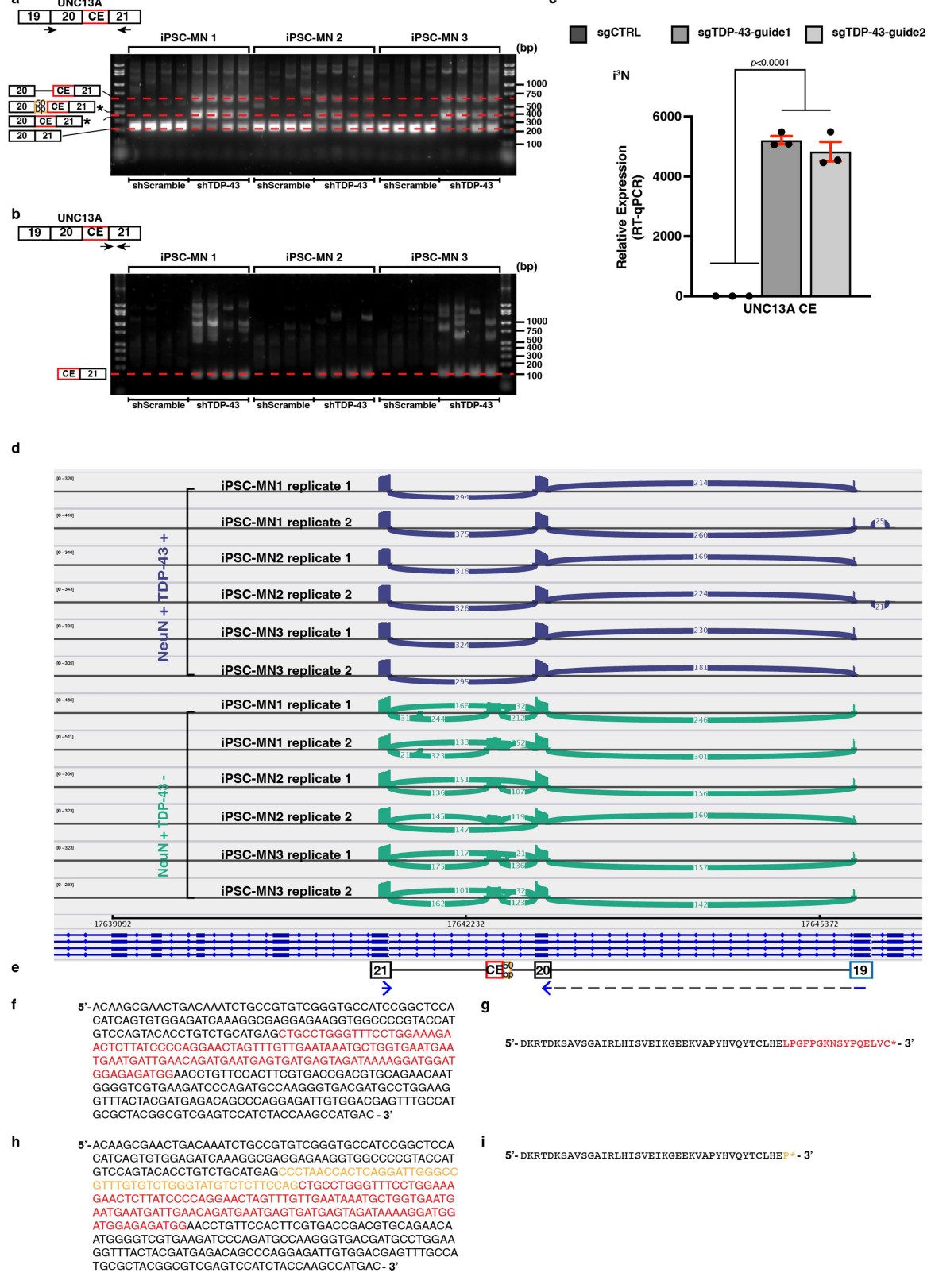

**Extended Data Fig. 3** | See next page for caption.

**Extended Data Fig. 3 | Depletion of TDP-43 from iPSC derived motor neurons (iPSC-MNs) and iPSC derived neurons (i³Ns) leads to cryptic exon inclusion in *UNC13A*. (a)** RT-PCR confirmed the expression of the cryptic exon-containing *UNC13A* splice variant upon TDP-43 depletion in three independent iPSC-MNs (n = 4 independent cell culture experiments for each condition). Gel picture shows results from all 4 experiments performed. In addition to the splice variant containing the 128 bp and 178 bp cryptic exons, we also detected inclusion of the complete intron upstream of the cryptic exon (Fig. 4e, Supplementary Note 2). The 128 bp and 178 bp cryptic exons cannot be distinguished here but they are detected through amplicon sequencing the corresponding band **(d)**. The PCR products represented by each band are marked to the left of each gel. The location of the PCR primer pair used is shown on top of each gel image. **(b)** The PCR primer pairs spanning the cryptic exon and exon 21 junction confirms cryptic exon inclusion only occurs upon TDP-43 knockdown. For gel source data, see Supplementary Fig. 1c, d. **(c)** RT-qPCR analyses confirmed the inclusion of *UNC13A* cryptic exon upon TDP-43 depletion in iPSC derived neurons (i3Ns). TDP-43 was depleted by expressing two different sgRNAs: sgTDP-43-guide1 and sgTDP-43-guide2 in i³Ns stably expressing CRISPR inactivation machinery (CRISPRi). *RPLPO* and *GAPDH* were used to normalize RT-qPCR. (n = 3 independent cell culture experiments for each condition; Ordinary one-way ANOVA with Dunnett's multiple comparisons test, mean ± s.e.m.). **(d)** Sashimi plot visualization of the alignment (hg38) of the amplicon (2 × 250 bp) sequencing reads of the sequences amplified using primers (blue) shown in **(e)**. Both the 128 bp and the 178 bp cryptic exons were supported by the sequencing reads. **(e)** Schematic of the exons amplified by the primers (blue). **(f, h)** DNA sequence of the 128 bp and 178 bp cryptic exons and their flanking exons. The sequences are color coded according to **(e)**. **(g, i)** The amino acid sequences correspond to the DNA sequences in **(f, h)**. The asterisks indicate stop codons are encountered.

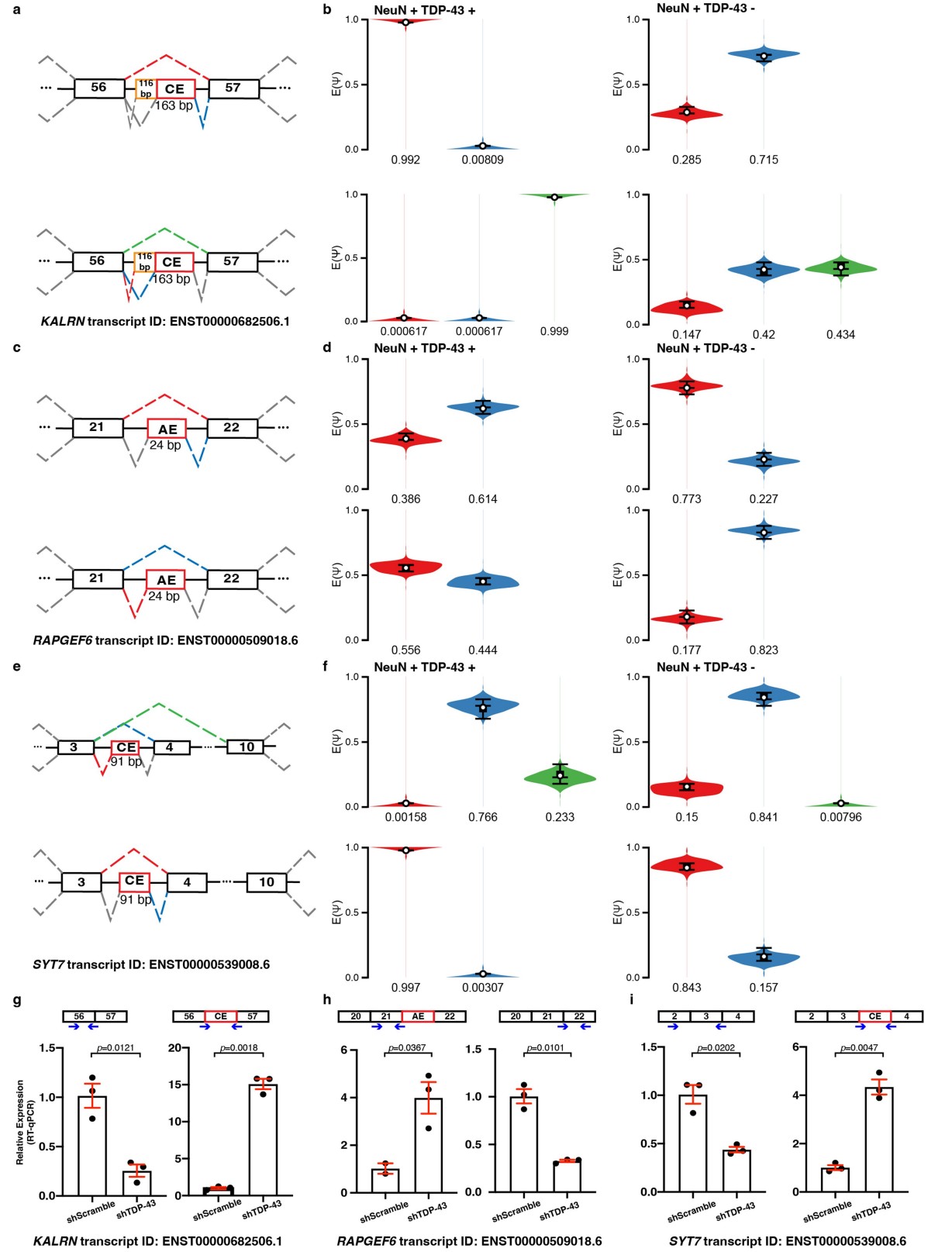

**Extended Data Fig. 4** | See next page for caption.

**Extended Data Fig. 4 | Validation of additional splicing targets.**
**(a–f)** Depletion of TDP-43 introduces cryptic exons into *KALRN* mRNA and *SYT7* mRNA; In *RAPGEF6*, TDP-43 depletion leads to a decrease in the usage of the exon AE (for alternative exon) between exon 21 and exon 22 of the isoform ENST0000509018.6. Exon AE does exist in some other isoforms of *RAPGEF6*, indicating the depletion of TDP-43 could lead to changes in isoform composition. **(b, d, f)** Violin plots corresponding to **(a, c, e)**, respectively. Each violin in **(b, d, f)** represents the posterior probability distribution of the expected relative inclusion (PSI or Ψ) for the color matching junction in the splice graph. The tails of each violin represent the 10th and 90th percentile. The box represents the interquartile range with the line in the middle indicating the median. The white circles mark the expected PSI (E[Ψ]). The change in the relative inclusion level of each junction between two conditions is referred to as ΔΨ or ΔPSI[13]. **(g–i)** RT-qPCR analyses confirmed changes in exon usage upon TDP-43 depletion in iPSC-derived neurons. *RPLPO* and *GAPDH* were used to normalize RT-PCR. (n = 3 independent cell culture experiments for each condition, two sided-Welch Two Sample t-test, mean ± s.e.m.).

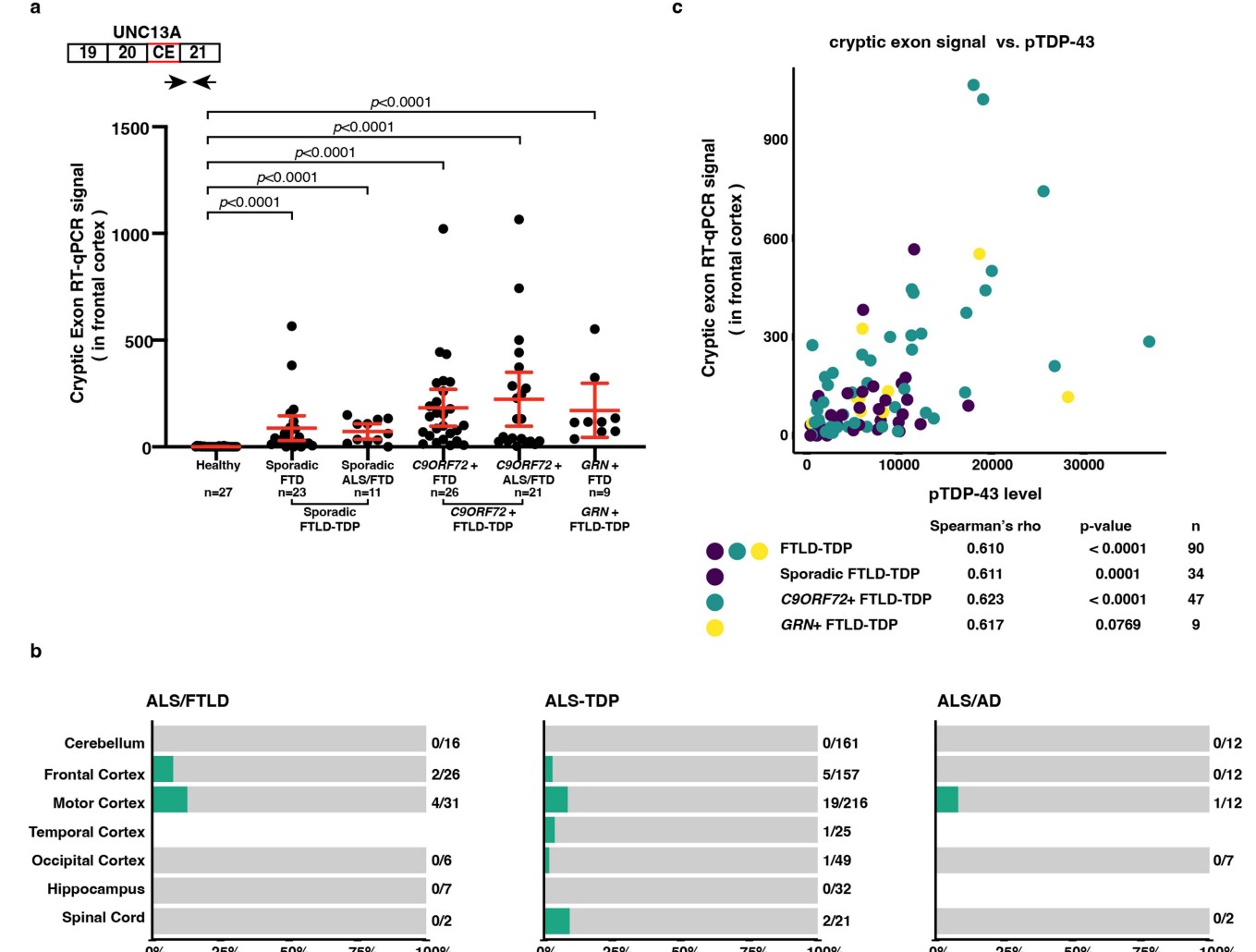

**Extended Data Fig. 5 | *UNC13A* cryptic exon inclusion is detected in disease relevant tissues of FTLD-TDP, ALS/FTLD, ALS-TDP and ALS/AD patients, and is correlated with phosphorylated TDP-43 levels in frontal cortices of FTLD-TDP patients. (a)** *UNC13A* cryptic exon expression level is significantly increased in the frontal cortices of patients with FTD and ALS/FTD clinical diagnoses (Mayo Clinic Brain Bank). *GAPDH* and *RPLP0* were used to normalize RT-qPCR (the sample size of each group is listed under the corresponding group; two-tailed Mann-Whitney test, mean ± 95% confidence interval). The schematic on top shows the localization of the primer pair (arrows) used for the RT-qPCR assay. **(b)** *UNC13A* splice variants are observed in ALS patients with unconfirmed pathology. ALS-FTLD refers to patients who have concurrent FTD and ALS. ALS patients were categorized based on whether they carry *SOD1* mutations (ALS-SOD1 (Fig. 2b) vs. ALS-TDP). ALS-AD refers to ALS patients with suspected Alzheimer's disease. The diagnoses of these patients (NYGC) are not neuropathologically confirmed. Therefore, it is unclear whether TDP-43 mislocalization is present. **(c)** *UNC13A* cryptic exon signal is positively correlated with phosphorylated TDP-43 levels in frontal cortices of FTLD-TDP patients in Mayo Clinic Brain Bank (Spearman's rho = 0.610, n = 90, p-values were calculated by one-sided t-test). Data points are colored according to patients' reported genetic mutations. The correlation within each genetic mutation group and the corresponding p-value and sample size is also shown.

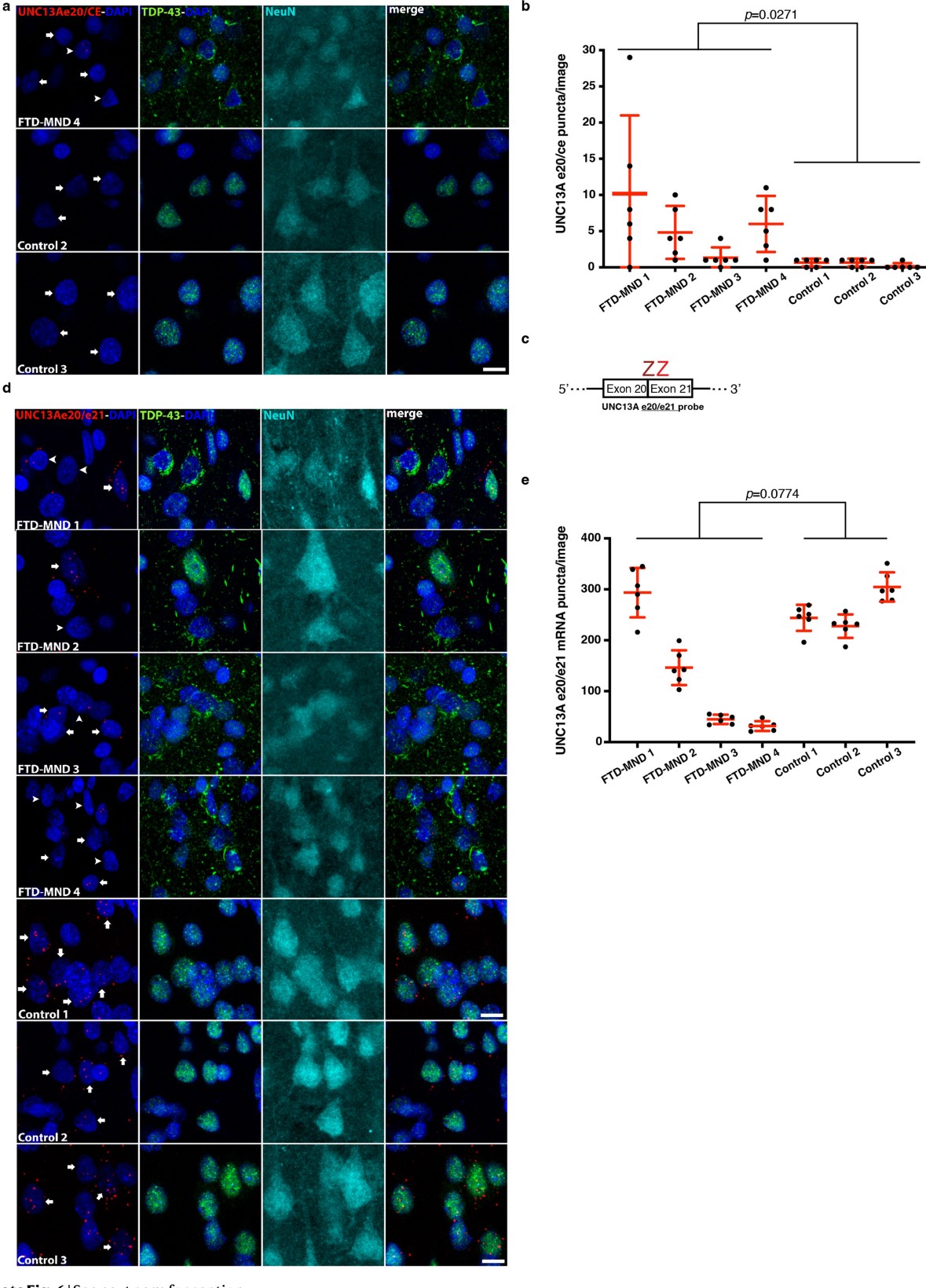

**Extended Data Fig. 6** | See next page for caption.

**Extended Data Fig. 6 | *UNC13A* cryptic splicing is associated with loss of nuclear TDP-43 in patients with FTD and motor neuron disease (MND).**
**(a)** Additional patients and control subjects used in the study (but not shown in Fig. 3), demonstrating *UNC13A* cryptic splicing. Scale bar equals 10 µm.
**(b)** Quantification of *UNC13A* cryptic exon BaseScope™ in situ hybridization. Six non-overlapping Z-stack images from layer 2–3 of medial frontal pole were captured, per subject, using a 63X oil objective and flattened into a maximum intensity projection image. Puncta counts per image were derived using the "analyze particle" plugin in ImageJ. Each data point represents the number of *UNC13A* cryptic exon puncta in a single image. Cryptic exon quantity varies between patients but always exceeds the technical background of the assay, as observed in controls. (n = 6 non-overlapping Z-stack images; Linear mixed model, mean ± s.d.). **(c)** The design of the UNC13A e20/e21 BaseScope™ probe targeting canonical *UNC13A* transcript. Each "Z" binds to the transcript independently. Both "Z"s must be in close proximity for successful signal amplification, ensuring binding specificity. **(d)** Representative images showing expression of *UNC13A* mRNA in layer 2–3 neurons from the medial frontal pole using the probe shown in **(b)**. *UNC13A* mRNA expression is restricted to neurons (arrows) and is decreased in cells exhibiting TDP-43 nuclear depletion. Arrowheads represent neurons with loss of nuclear TDP-43 and accompanying cytoplasmic inclusions, and arrows indicate neurons with normal nuclear TDP-43. Images are maximum intensity projections of a confocal image Z-stack. Scale bar equals 10 µm. **(e)** Quantification of *UNC13A* mRNA BaseScope™ in situ hybridization. *UNC13A* mRNA puncta were quantified as described in **(b)**. Each data point represents the number of *UNC13A* mRNA puncta in a single image. This suggests some variability in *UNC13A* mRNA levels potentially attributable to technical or biological factors. More importantly, the control *UNC13A* mRNA levels suggest that failure to detect *UNC13A* cryptic exons in controls is not due to nonspecific RNA degradation. There is variability of *UNC13A* mRNA detected per sample but we observe a trend of reduced *UNC13A* mRNA in patient samples compared to controls. (Linear mixed model, mean ± s.d.). The final BaseScope™ experimental run was performed once involving all the cases and controls. Six non-overlapping images were captured from each individual, and representative images are shown. *UNC13A* probes were first optimized by testing them on 2 cases and 2 controls in 3 separate pilot experiments, showing similar findings.

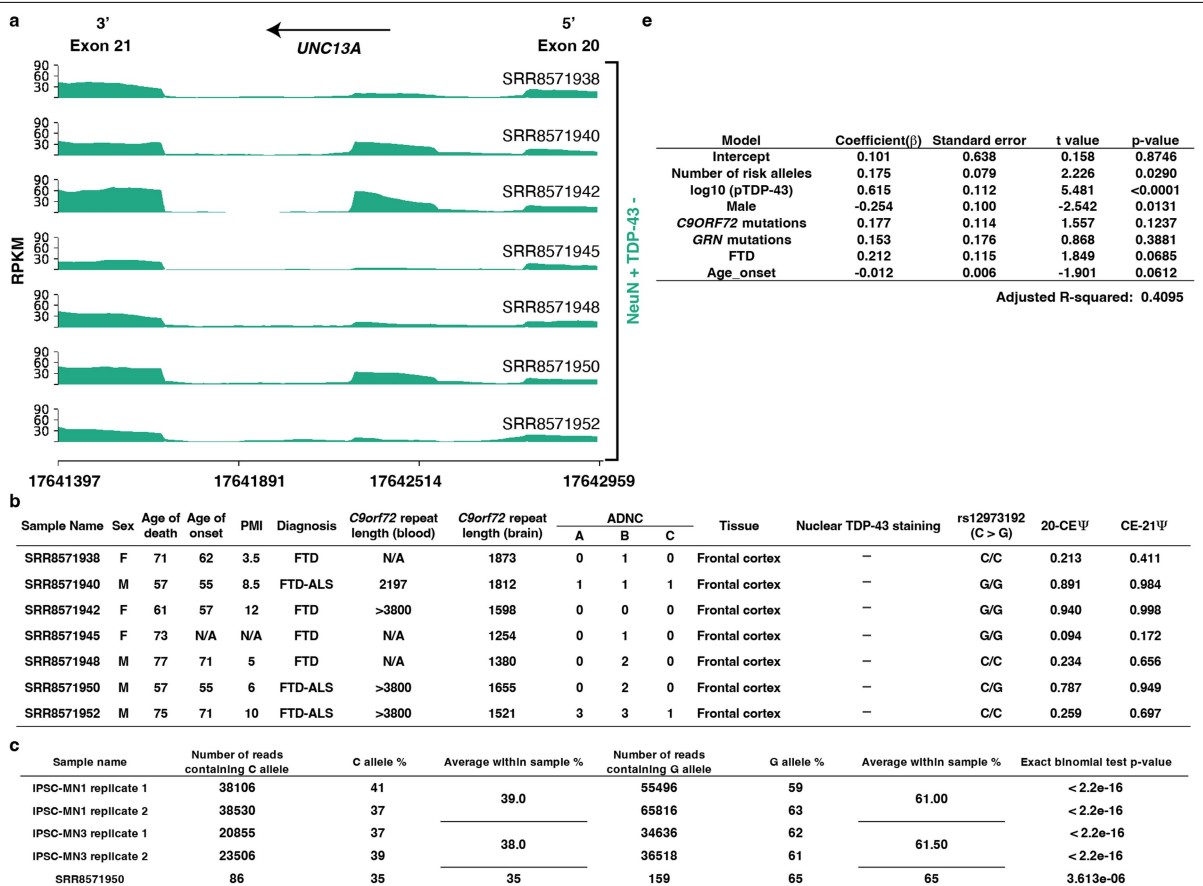

**a**

3' Exon 21 ← *UNC13A* 5' Exon 20

NeuN + TDP-43 -

**e**

| Model | Coefficient(β) | Standard error | t value | p-value |
|---|---|---|---|---|
| Intercept | 0.101 | 0.638 | 0.158 | 0.8746 |
| Number of risk alleles | 0.175 | 0.079 | 2.226 | 0.0290 |
| log10 (pTDP-43) | 0.615 | 0.112 | 5.481 | <0.0001 |
| Male | -0.254 | 0.100 | -2.542 | 0.0131 |
| *C9ORF72* mutations | 0.177 | 0.114 | 1.557 | 0.1237 |
| *GRN* mutations | 0.153 | 0.176 | 0.868 | 0.3881 |
| FTD | 0.212 | 0.115 | 1.849 | 0.0685 |
| Age_onset | -0.012 | 0.006 | -1.901 | 0.0612 |

Adjusted R-squared: 0.4095

**b**

| Sample Name | Sex | Age of death | Age of onset | PMI | Diagnosis | *C9orf72* repeat length (blood) | *C9orf72* repeat length (brain) | ADNC A | ADNC B | ADNC C | Tissue | Nuclear TDP-43 staining | rs12973192 (C > G) | 20-CEΨ | CE-21Ψ |
|---|---|---|---|---|---|---|---|---|---|---|---|---|---|---|---|
| SRR8571938 | F | 71 | 62 | 3.5 | FTD | N/A | 1873 | 0 | 1 | 0 | Frontal cortex | − | C/C | 0.213 | 0.411 |
| SRR8571940 | M | 57 | 55 | 8.5 | FTD-ALS | 2197 | 1812 | 1 | 1 | 1 | Frontal cortex | − | G/G | 0.891 | 0.984 |
| SRR8571942 | F | 61 | 57 | 12 | FTD | >3800 | 1598 | 0 | 0 | 0 | Frontal cortex | − | G/G | 0.940 | 0.998 |
| SRR8571945 | F | 73 | N/A | N/A | FTD | N/A | 1254 | 0 | 1 | 0 | Frontal cortex | − | G/G | 0.094 | 0.172 |
| SRR8571948 | M | 77 | 71 | 5 | FTD | N/A | 1380 | 0 | 2 | 0 | Frontal cortex | − | C/C | 0.234 | 0.656 |
| SRR8571950 | M | 57 | 55 | 6 | FTD-ALS | >3800 | 1655 | 0 | 2 | 0 | Frontal cortex | − | C/G | 0.787 | 0.949 |
| SRR8571952 | M | 75 | 71 | 10 | FTD-ALS | >3800 | 1521 | 3 | 3 | 1 | Frontal cortex | − | C/C | 0.259 | 0.697 |

**c**

| Sample name | Number of reads containing C allele | C allele % | Average within sample % | Number of reads containing G allele | G allele % | Average within sample % | Exact binomial test p-value |
|---|---|---|---|---|---|---|---|
| IPSC-MN1 replicate 1 | 38106 | 41 | 39.0 | 55496 | 59 | 61.00 | < 2.2e-16 |
| IPSC-MN1 replicate 2 | 38530 | 37 | | 65816 | 63 | | < 2.2e-16 |
| IPSC-MN3 replicate 1 | 20855 | 37 | 38.0 | 34636 | 62 | 61.50 | < 2.2e-16 |
| IPSC-MN3 replicate 2 | 23506 | 39 | | 36518 | 61 | | < 2.2e-16 |
| SRR8571950 | 86 | 35 | 35 | 159 | 65 | 65 | 3.613e-06 |

**d**

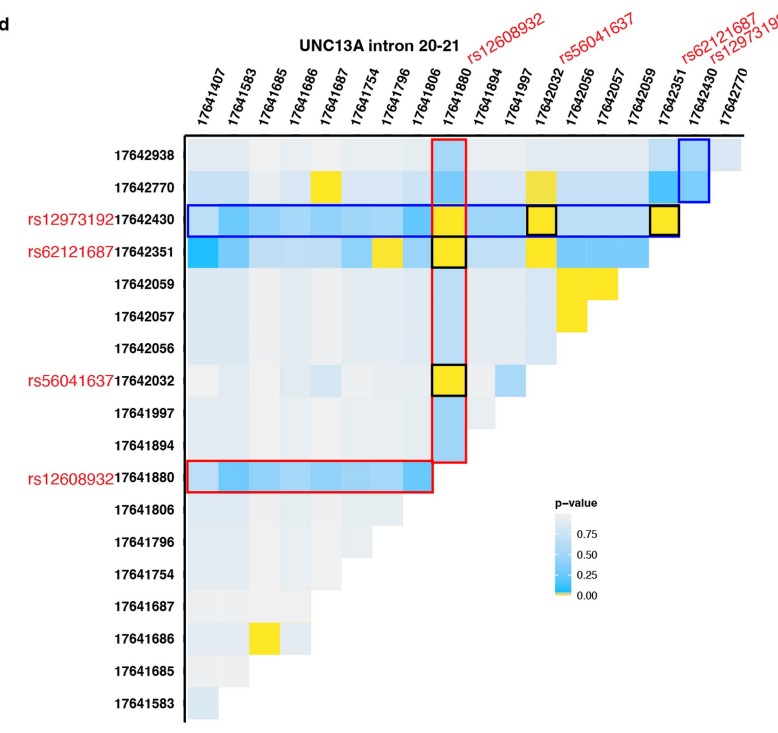

**Extended Data Fig. 7** | See next page for caption.

**Extended Data Fig. 7 | Levels of *UNC13A* cryptic exon inclusion are influenced by the number of risk haplotypes. (a)** Visualization of RNA-Seq alignment between exon 20 and exon 21 of *UNC13A*. RNA-Seq libraries were generated from TDP-43-negative neuronal nuclei as described in Fig. 1a. **(b)** Samples that are heterozygous (C/G) or homozygous (G/G) at rs12973192 have higher relative inclusion (Ψ) of the cryptic exon except for SRR8571945. Information about the patients were obtained from Liu et al.[12]. **(c)** Percentages of C and G alleles in the *UNC13A* spliced variants in TDP-43 depleted iPSC-MNs and SRR8571950 neuronal nuclei. Exact binomial test was done for each replicate to test whether the observed difference in percentages differ from what was expected if both alleles are equally included in the cryptic exon. **(d)** rs56041637 and rs62121687 are in strong linkage disequilibrium with both GWAS hits in intron 20-21 of *UNC13A* (Method). Along the axes of the heatplot are all loci that show variation among the 297 patients from Answer ALS in July 2020. Each tile represents the p-value from the corresponding Chi-Square test. P-value < 0.05 are shown in yellow and others are shown in blue or gray. Red and blue blocks highlight the associations of rs12608932 and rs12973192 with other genetic variants in intron 20-21 respectively. Significant associations common to both are circled in black. **(e)** The summary results of multiple linear regression modeling the effects of the number of *UNC13A* risk alleles on the abundance of *UNC13A* cryptic exon inclusion measured by RT-qPCR. A multivariable model was derived adjusting for phosphorylated TDP-43 levels (pTDP-43), sex, known genetic mutations, disease types, and the age of onset. As shown in Extended Data Fig. 5c, pTDP-43 levels have a strong effect on the abundance of *UNC13A* cryptic exon inclusion. Normality of residuals is tested by Shapiro-Wilk normality test (p-value = 0.2014).

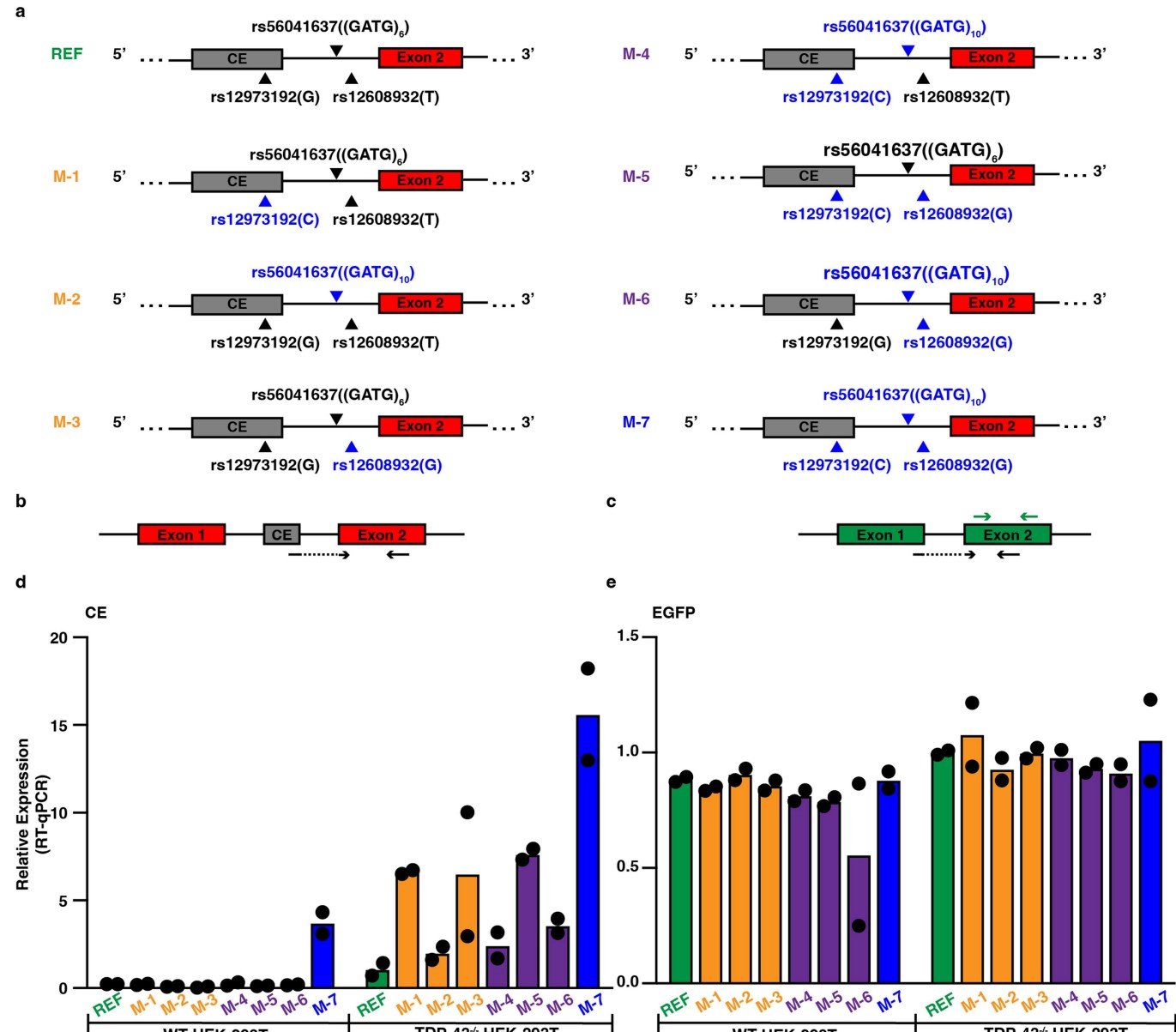

**Extended Data Fig. 8 | The impact of variants at rs12973192, rs12608932 and rs56041637 on splicing. (a)** Diagrams showing the design of the *UNC13A* minigene reporter constructs used to assess the impact of the variants at each locus. The complete design of the reporter construct is shown in Fig. 4d. For clarity, the mCherry and GFP exons that are closest to the promoter (blue in Fig. 4d) are labeled as exon 1, and the downstream exon are labeled as exon 2. REF is the reporter that carries the reference haplotype. M-1 to M-3 carry a single risk variant. M-4 to M-6 carry two risk variants. M-7 carries all three variants, the risk haplotype. **(b, c)** The locations of the RT-qPCR primer pairs used to detect the inclusion of the cryptic exon **(b)** and the splicing of EGFP

(**c**, shown in black). **(d, e)** The expression level of the cryptic exon **(d)** or the splicing of EGFP **(e)** in each condition is calculated with reference to the expression level of cryptic exon or the splicing of EGFP from the WT construct in TDP-43-/- HEK-293T cells. The expression of the reporter construct measured using a pair of primers aligned to the second exon of EGFP (**c**, shown in green) was used to normalize RT-qPCR. The cryptic exon expression levels of each pair of reporters expressed within the same cell line were compared. The splicing of EGFP remained constant across all conditions, verifying equal reporter expression levels and the integrity of the splicing machinery independent of TDP-43.

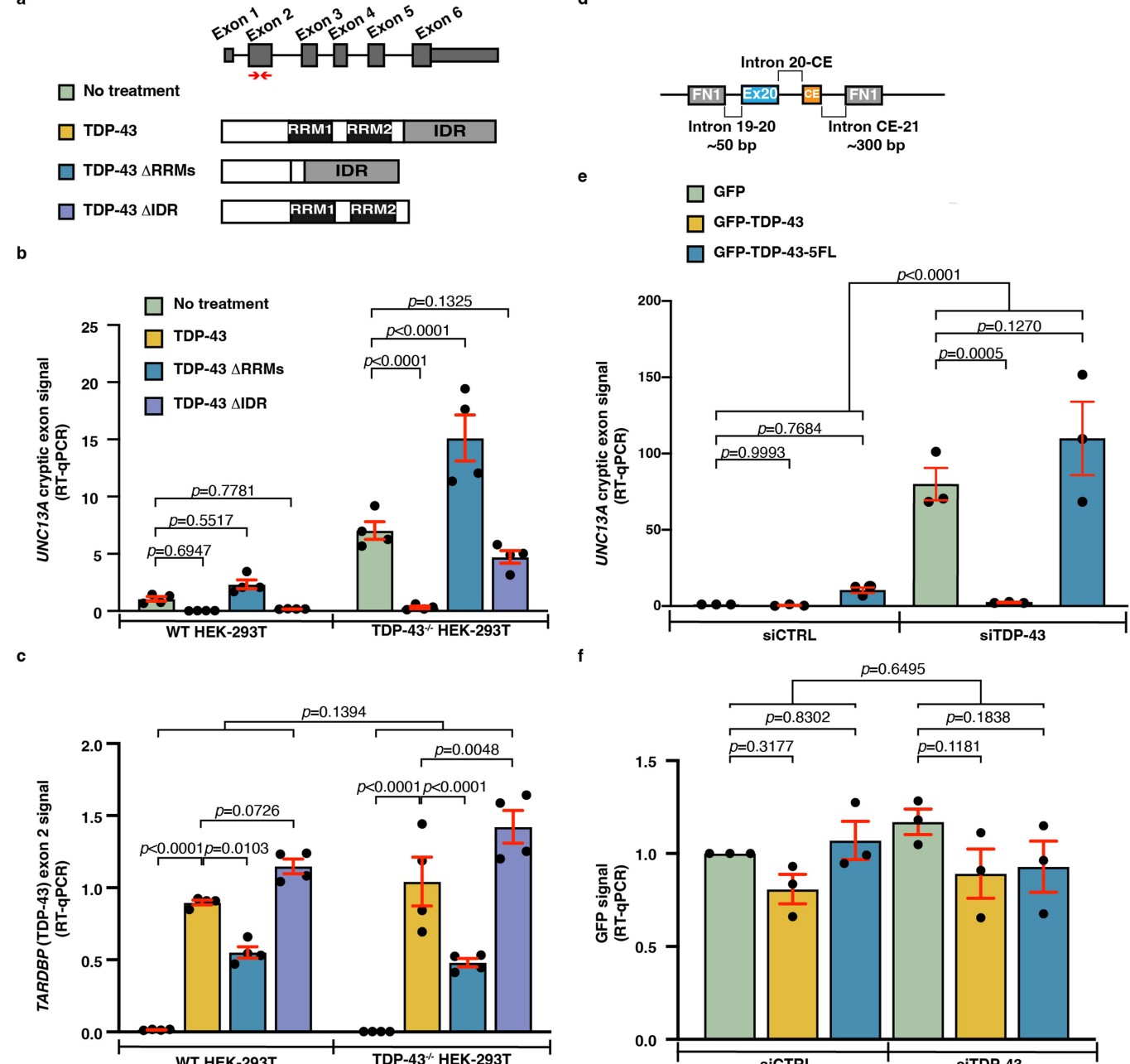

**Extended Data Fig. 9 | TDP-43-dependent minigene splicing reporter assay in HEK293T cells and HeLa cells. (a)** Schematic of various TDP-43 overexpression constructs used in HEK293T cells. RRM1&RRM2: RNA recognition motifs 1 & 2; IDR: intrinsically disordered region; adapted from[50]. The RT-qPCR primers (red arrows) for measuring the expression levels of the TDP-43 overexpression constructs are mapped to the second exon of *TARDBP*. The primer pair can detect all the TDP-43 overexpression constructs, including the endogenous TDP-43. Since the HEK293T TDP-43 knock-out cells do not have TDP-43[36], using the primers does not interfere with measurement of TDP-43 construct expression levels in TDP-43-/- HEK293T. **(b)** Expression of full-length TDP-43 rescued the splicing defects in HEK293T. TDP-43 lacking both RRMs (TDP-43 ΔRRMs) exacerbates the splicing defects and TDP-43 lacking the IDR (TDP-43 ΔIDR) has a much weaker rescue effect compared to full length TDP-43. **(c)** The expression levels of the second exon of TDP-43 across different conditions in HEK293T measured by RT-qPCR. The expression levels differ significantly, possibly due to the autoregulation of TDP-43[51]. Despite the variability, the full length TDP-43 is significantly better at reducing

the cryptic exon in *UNC13A* compared to TDP-43 ΔRRMs and TDP-43 ΔIDR. (n = 4 independent cell culture experiments for each condition in **(b, c)**). **(d)** Schematic of the pTB *UNC13A* minigene construct in HeLa cells. The pTB *UNC13A* minigene construct containing *UNC13A* cryptic exon sequence and the flanking sequences upstream (from 50 bp at the of end of intron 19 to the cryptic exon) and downstream (~300 bp intron 20) were expressed using the pTB vector, which we have previously used to study TDP-43 splicing regulation of other TDP-43 targets[47]. **(e)** Depletion of TDP-43 by siRNA in HeLa cells resulted in inclusion of the cryptic exon, which was rescued by expressing an siRNA-resistant form of TDP-43 (GFP-TDP-43) but not by an RNA-binding deficient mutant TDP-43 (GFP-TDP-43-5FL). **(f)** RT-qPCR of GFP demonstrating expressions of the constructs are similar across different conditions. (n = 3 independent cell culture experiments in **(e, f)**). *GAPDH* and *RPLPO* were used to normalize RT-qPCR in **(c)** and **(f)**. For all RT-PCR analysis, two-way ANOVA with Dunnett's's multiple comparisons test was used for all RT-qPCR analysis, mean ± s.e.m.).

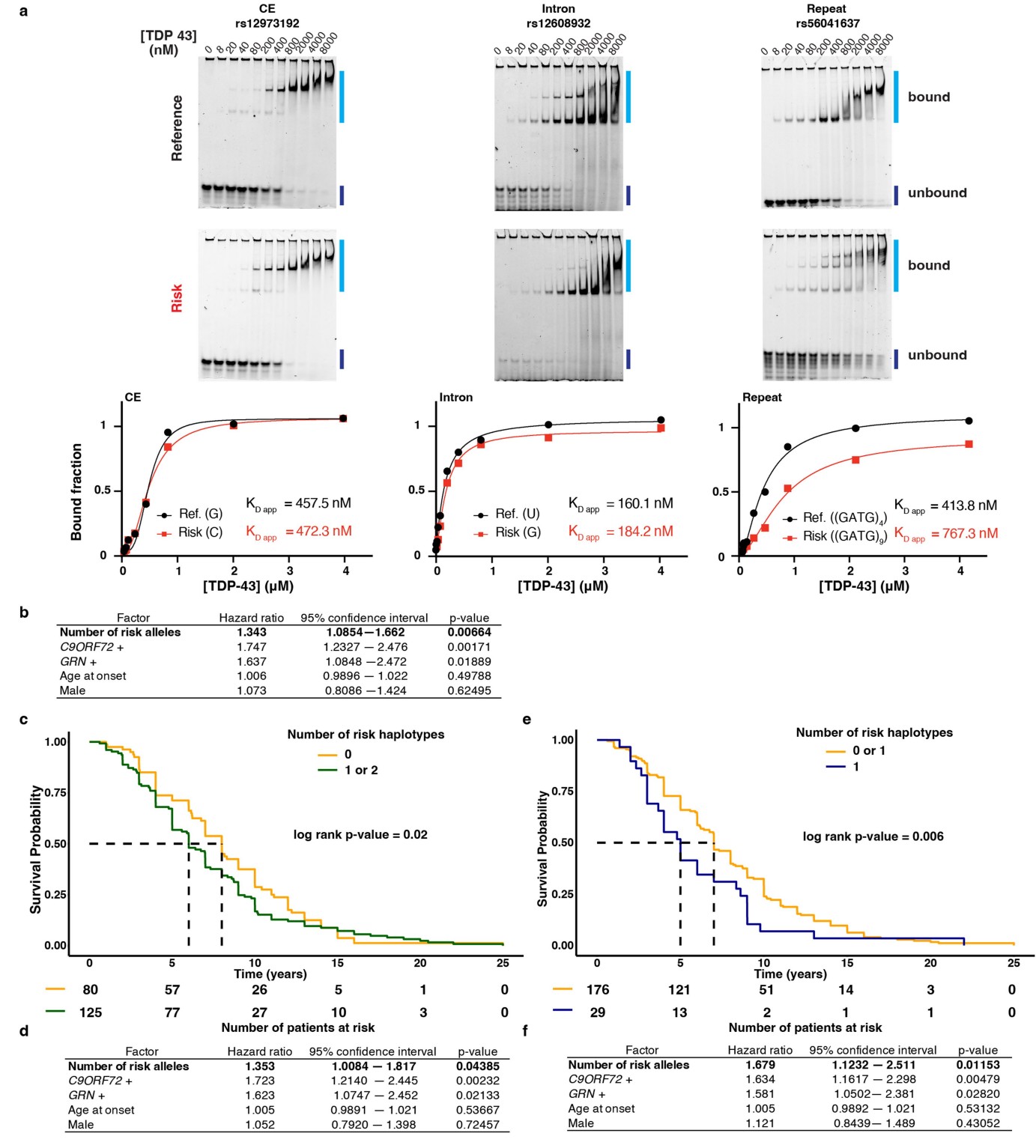

**Extended Data Fig. 10 | *UNC13A* risk haplotype is associated with diminished binding affinity of TDP-43 and reduced survival time of FTLD-TDP patients. (a)** Reference vs. risk allele RNA binding assay using electrophoretic mobility shift assay (EMSA). Exponentially increasing concentrations (0–8 µM) of purified TDP-43 were incubated with 1 nM Cy5 labeled RNA substrate for the reference and risk alleles of CE (rs12973192), intron (rs12608932), and repeat sequences (rs56041637) (see Methods). 8 mM values were excluded in plots due to significant aggregation at this concentration. The total bound population was quantified at each TDP-43 condition and used to plot the binding curve and calculate the apparent binding affinity, $K_{D app}$. Experiments were performed three times for introns and two times for exons and repeats, which produced similar results. For gel source data, see Supplementary Fig. 1g. **(b)** *UNC13A* risk haplotype is associated with reduced survival time of FTLD-TDP patients. Summary results of Cox multivariable analysis (adjusted for genetic mutations, sex and age at onset) of an additive model. **(c, e)** Survival curves of FTLD-TDP patients (n = 205, Mayo Clinic Brain Bank), according to a dominant model **(c)** and a recessive model **(e)** and their corresponding risk tables. Summary results of Cox multivariable analysis (adjusted for genetic mutations, sex and age at onset) of a dominant model **(d)** and a recessive model **(f)**. Both the dominant model **(c, d)** and the recessive model **(e, f)** show that the presence of the risk haplotype can reduce the survival of FTLD-TDP patients. Dashed lines mark the median survival for each genotype. Log rank p-values were calculated using Score (logrank) test. **(b, d, f)** The significance of each factor was calculated by Wald test.

# Reporting Summary

## Statistics

For all statistical analyses, confirm that the following items are present in the figure legend, table legend, main text, or Methods section.

| n/a | Confirmed | |
|---|---|---|
| ☐ | ☒ | The exact sample size (*n*) for each experimental group/condition, given as a discrete number and unit of measurement |
| ☐ | ☒ | A statement on whether measurements were taken from distinct samples or whether the same sample was measured repeatedly |
| ☐ | ☒ | The statistical test(s) used AND whether they are one- or two-sided<br>*Only common tests should be described solely by name; describe more complex techniques in the Methods section.* |
| ☐ | ☒ | A description of all covariates tested |
| ☐ | ☒ | A description of any assumptions or corrections, such as tests of normality and adjustment for multiple comparisons |
| ☐ | ☒ | A full description of the statistical parameters including central tendency (e.g. means) or other basic estimates (e.g. regression coefficient) AND variation (e.g. standard deviation) or associated estimates of uncertainty (e.g. confidence intervals) |
| ☐ | ☒ | For null hypothesis testing, the test statistic (e.g. *F*, *t*, *r*) with confidence intervals, effect sizes, degrees of freedom and *P* value noted<br>*Give P values as exact values whenever suitable.* |
| ☒ | ☐ | For Bayesian analysis, information on the choice of priors and Markov chain Monte Carlo settings |
| ☒ | ☐ | For hierarchical and complex designs, identification of the appropriate level for tests and full reporting of outcomes |
| ☐ | ☒ | Estimates of effect sizes (e.g. Cohen's *d*, Pearson's *r*), indicating how they were calculated |

*Our web collection on statistics for biologists contains articles on many of the points above.*

## Software and code

Policy information about availability of computer code

| Data collection | N/A |
|---|---|
| Data analysis | trimmomatic (0.39), FastQC (v0.11.9), STAR v2.7.3a, RSEM v1.3.1, R v4.0.0  and R v4.0.2, DESeq2(v1.28.1), MISO(misopy 0.5.4),, VCFtools (0.1.16), BCFtools (1.8), Prism 8; https://github.com/rosaxma/TDP-43-UNC13A-2021 |

For manuscripts utilizing custom algorithms or software that are central to the research but not yet described in published literature, software must be made available to editors and reviewers. We strongly encourage code deposition in a community repository (e.g. GitHub). See the Nature Portfolio guidelines for submitting code & software for further information.

## Data

Policy information about availability of data

All manuscripts must include a data availability statement. This statement should provide the following information, where applicable:

- Accession codes, unique identifiers, or web links for publicly available datasets
- A description of any restrictions on data availability
- For clinical datasets or third party data, please ensure that the statement adheres to our policy

The amplicon sequencing data has been deposited in the Gene Expression Omnibus (GEO) at GSE182976. RNA-Seq data for splicing analysis is available at GSE126543. RNA-Seq data generated by NYGC ALS Consortium cohort is available at GSE137810, GSE124439, GSE116622, and GSE153960

# Field-specific reporting

Please select the one below that is the best fit for your research. If you are not sure, read the appropriate sections before making your selection.

☒ Life sciences ☐ Behavioural & social sciences ☐ Ecological, evolutionary & environmental sciences

For a reference copy of the document with all sections, see nature.com/documents/nr-reporting-summary-flat.pdf

# Life sciences study design

All studies must disclose on these points even when the disclosure is negative.

| | |
|---|---|
| Sample size | No sample size calculation was performed; for in vitro experiments, sample sizes were choses such that statistical significance could be confidently established; for analysis using patient data, sample sizes were chosen based on data availability. |
| Data exclusions | No data from in vitro experiments were excluded. The data exclusion criteria for the following experiments/analyses were pre-established. For the in situ hybridization experiment using postmortem brain samples, subject were excluded if they had a known disease-causing mutation, post-mortem interval ≥ 24 h, Alzheimer's disease neuropathologic change > low, Thal amyloid phase > 2, Braak neurofibrillary tangle stage > 4, CERAD neuritic plaque density > sparse, and Lewy body disease > brainstem predominant. For the multiple linear regression analysis, we excluded one subject who does not carry the haplotypes we were studying. In the analysis that quantifies UNC13A splice variants in bulk RNA, we exclude samples whose UNC13A expression was too low for us to detect the splice variants. |
| Replication | All the experiments presented in the studies have been successfully replicated in independent cell cultures. The results of the experiments contributing to the main findings have been replicated in separate studies. |
| Randomization | Randomization is not applicable to cell culture experiments. Randomization is also not applied to the experiments that involve human postmortem tissues. |
| Blinding | Investigators were not blinded to the identity of the samples but findings have been replicated by independent investigators in multiple laboratories. |

# Reporting for specific materials, systems and methods

We require information from authors about some types of materials, experimental systems and methods used in many studies. Here, indicate whether each material, system or method listed is relevant to your study. If you are not sure if a list item applies to your research, read the appropriate section before selecting a response.

## Materials & experimental systems

| n/a | Involved in the study |
|---|---|
| ☐ | ☒ Antibodies |
| ☐ | ☒ Eukaryotic cell lines |
| ☒ | ☐ Palaeontology and archaeology |
| ☒ | ☐ Animals and other organisms |
| ☐ | ☒ Human research participants |
| ☒ | ☐ Clinical data |
| ☒ | ☐ Dual use research of concern |

## Methods

| n/a | Involved in the study |
|---|---|
| ☒ | ☐ ChIP-seq |
| ☒ | ☐ Flow cytometry |
| ☒ | ☐ MRI-based neuroimaging |

## Antibodies

| | |
|---|---|
| Antibodies used | HRP-conjugated anti-mouse IgG (H+L) (Fisher 62-6520), HRP-conjugated anti-rabbit IgG (H+L) (Life Technologies 31462) |
| Validation | HRP-conjugated anti-mouse IgG (H+L) (https://www.thermofisher.com/antibody/product/Goat-anti-Mouse-IgG-H-L-Secondary-Antibody-Polyclonal/62-6520). Extracts from multiple cell lines were first probed with appropriate mouse monoclonal antibodies and then probed with HRP-conjugated anti-mouse IgG (H+L) to verify specificity. HRP-conjugated anti-rabbit IgG (H+L) (Life Technologies 31462) (https://www.thermofisher.com/antibody/product/Goat-anti-Rabbit-IgG-H-L-Cross-Adsorbed-Secondary-Antibody-Polyclonal/31462). Extracts from multiple cell lines were first probed with appropriate rabbit polyclonal antibodies and then probed with HRP-conjugated anti-rabbit IgG (H+L) to verify specificity. |

## Eukaryotic cell lines

Policy information about cell lines

| | |
|---|---|
| Cell line source(s) | SH-SY5Y, HEK293T: ATCC; iPSCs: Coriell Repository |

| Authentication | Cell line authentication was performed by the supplier.<br>SH-SY5Y: STR profiling.<br>iPSC lines: GM25256 (karyotypic analysis), NDS00262(96 SNP assay), NDS00209(96 SNP assay).<br>HEK293T: STR profiling. |
| --- | --- |
| Mycoplasma contamination | Cells were not test for mycoplasma contamination by our labs. |
| Commonly misidentified lines<br>(See ICLAC register) | Cell lines were not listed in ICLAC register. |

## Human research participants

Policy information about studies involving human research participants

| Population characteristics | Mayo Clinic Brain Bank: patients with FTLD-TDP and cognitively normal control individuals; UCSF: see Supplementary Table 4 |
| --- | --- |
| Recruitment | Written informed consent was obtained before study entry from all subjects or their legal next of kin if they were unable to give written consent, and biological samples were obtained with Mayo Clinic Institutional Review Board (IRB) approval. |
| Ethics oversight | Mayo Clinic Institution Review Board and Ethics Committee, UCSF Committee on Human Research |

Note that full information on the approval of the study protocol must also be provided in the manuscript.

