## [Peer Review File · Nature]

Manuscript Title: TDP-43 represses cryptic exon inclusion in FTD/ALS gene UNC13A

Reviewer Comments & Author Rebuttals

Reviewer Reports on the Initial Version:

Referee #1

TDP-43 represses cryptic exon inclusion in FTD/ALS gene UNC13A (Rosa et al., 2021) is an interesting manuscript, which potentially provides a mechanistic understanding of one of the established risk loci for FTD/ALS and directly links it to the function and deposition of TDP43 within the brain tissue of affected individuals. There are relatively few examples where this form of molecular understanding has been achieved for complex diseases and therefore this paper will generate considerable interest. However, it will be of greatest interest to those focused on TDP43-related diseases and those interested in the development of novel therapies in the field. It is not clear to me the extent to which these findings or the approach taken are generalisable beyond this specific disease and within that the subtype characterised by TDP43 deposits.

A range of methods and approaches have been used to connect TDP43 function to the UNC13A locus, but a significant component of the whole paper and the initial starting point as presented in the manuscript is RNAseq data analysis. While these analyses are likely to be sound given the expertise of some of the authors of this manuscript, given the importance of this analysis to the paper, there is insufficient detail within the manuscript itself to assess this - though I appreciate that github links are provided I think this information needs to be in the manuscript itself when it is so crucial to the interpretation. I think this becomes particularly problematic when running what are in effect eQTL and ASE analyses at the UNC13A locus. It is simply not clear to me if covariate corrections have been made as would normally be expected for an eQTL analysis or if the ASE analyses has accounted for mapping biases which could easily drive false positive findings. More generally within the main text my impression was there were occasions where significance of a result was mentioned but without a p-value, test type or n, though this was available within the relevant figure.

In terms of further experimental work which I think would improve this study, it is now possible to run targeted long read RNAseq which could be performed across samples and which could allow more accurate delineation of the proposed abnormal UNC13A transcript/s containing the CE and the relative usage of transcripts. I think the inclusion of this form of data would significantly add to the work.

I also think it would be valuable to go back to iPSC/mouse models of pathogenic Mendelian forms of FTD/ALS due to TDP43 mutations. It strikes me that variability in the severity of the disease could be explained amongst these patients by variability at the UNC13A locus and this would substantiate the findings.

In summary, I think this is a very valuable and interesting piece of work, but one which would benefit from significant improvements in the methodological detail to enable it to be fully assessed.

In terms of specific comments:

"To identify changes associated with loss of TDP-43 from the nucleus, the authors used fluorescence-activated cell sorting (FACS) to enrich neuronal nuclei with and without TDP-43 from postmortem FTD/ALS patient brain tissue and then performed RNA-seq to compare the transcriptomic profiles between TDP-43-positive and TDP-43-negative neuronal nuclei. They identified a multitude of interesting differentially expressed genes 11."

- The validity of re-analysing this data set for differential splicing when the RNAseq data being used originates from the nuclei needs to be addressed. It may be that this is identifying a problem with the rate of RNA processing for the genes not an actual splicing defect through this analysis.

"There were 65 alternatively spliced genes in common between both analyses (Fig. 1b), 35 likely because each tool uses different definitions for transcript variations and different criteria to control for false positives. Among the alternatively spliced genes identified by both tools were STMN2 and POLDIP3, both of which have been extensively validated as bona fide TDP-43 splicing targets 8–10,14."

- Was there any statistically significant overlap between the genes with splicing changes as identified by LeafCutter and MAGIQ? Also these aren't really two independent methods for assessing splicing changes given that they both use a fundamentally similar approach ie. only apparently exon-exon junction reads are analysed.
- Furthermore, merging the output from these two tools is far from trivial and there is nothing in the methods to explain this. It is very possible that the two tools identify the same genes but because of an entirely different event. Has there been any attempt to map intron clusters?
- Was there any overall enrichment of genes previously implicated in FTD/ALS within the gene list? "This new exon (CE, for cryptic exon) was absent in wild type neuronal nuclei (Fig. 1c) and is not present in any of the known human isoforms of UNC13A 15. Furthermore, analysis of ultraviolet cross-linking and 45 immunoprecipitation (iCLIP) data for TDP-43 3 provides evidence that TDP-43 directly binds to the intron harboring this cryptic exon (shown by mapped reads) (Fig. 1d)"
- It is not clear to me how absence was defined in the wild type neuronal nuclei? Given that data will inevitably be sparse in nature and account needs to be taken of sequencing depth, number of nuclei captured etc this may not be trivial and the methods do not provide the detail necessary to judge this.

"Reducing levels of TDP-43 in iPSC-derived motor neurons (iPSC-MNs) (Fig. 1 h to j; Extended Data Fig. 3 a and b) and excitatory neurons (i3Ns) derived from human iPSCs (Extended Data Fig. 4) also resulted in cryptic exon inclusion in UNC13A and a reduction in UNC13A mRNA and protein. We confirmed insertion of the 128 bp cryptic exon sequence into the mature transcript by direct sequencing of the RT-PCR product (* in Extended Data Fig. 3a). Thus, lowering levels of TDP-43 in human cells and neurons causes inclusion of a cryptic exon in the UNC13A transcript, resulting in decreased UNC13A protein."

- It would be helpful to more clearly understand the relationship between i) the inclusion of the CE, ii) the drop in UNC13A levels and iii) the UNC13A protein levels. Given the fact that NMD will remove the CE transcripts, but this could trigger compensatory increase in UNC13A transcription, and in general correlations between RNA levels and protein levels are poor the protein level drops seem dramatic. To tease this out more carefully, I think correlations between these three different analyses should be performed.

To extend our analysis of UNC13A cryptic exon (CE) inclusion to a larger collection of patient samples, we first analyzed a series of 117 frontal cortex brain samples from the Mayo Clinic Brain 30 Bank and found a significant increase in UNC13A CE levels in FTLTDP patients compared to healthy controls (Fig. 2a and Extended Data Fig. 5a).

- As mentioned before, I do not think there is sufficient detail in the methods to assess how this was done.

Owing to noise generated from bulk sequencing, we scored the UNC13A splice variant as present if there were more than two reads spanning at least one of the exon-exon junctions. We identified 63 samples, from 49 patients, which met the above criteria. We detected UNC13A splice variants in nearly 50% of the frontal and temporal cortical tissues donated by patients with neuropathologically confirmed FTLTDP (Fig. 2b).

- Surely this analysis could be made more robust by actually just running LeafCutter or MAJIQ? Was this tried and didn't yield? If this is the case then I appreciate that detection of these abnormal junctions would be helpful, but then I think the authors should still be able to account for read depth specifically over the expected junction.

We also detected the splice variants in some of the ALS patients whose pathology has not been confirmed (Extended Data Fig. 5b). Notably, 45 we did not observe UNC13A CE in any of the samples from FTLTDP-FUS (n=9), FTLTDP-TAU (n=18) and ALS-SOD1 (n=22) patients, nor in any of

the control samples (n=197) (Fig. 2b).

- Without accounting for read depth across the canonical junction the significance of these findings is unclear.

To determine the relationship between pTDP-43 levels and UNC13A cryptic exon inclusion we analyzed data from 90 patients with FTLD-TDP in the Mayo Clinic Brain Bank for which we had RT-qPCR and pTDP-43 levels (Methods) from frontal cortices. We found a striking association between higher pTDP-43 levels and higher levels of UNC13A cryptic exon inclusion in patients with FTLD-TDP (Spearman's $\rho = 0.572$, $P < 0.0001$) (excluding 4 individuals who did not show 10 UNC13A CE signal) (Fig. 2c and figure using raw data, Extended Data Fig. 5c).

- I do not understand the reason for excluding the 4 patients. If those individuals have FTLD-TDP and have the data to run the analysis then the fact they don't have the UNC13A CE does not invalidate them.

Among the seven patients included in the initial splicing analysis (Fig. 1a), 2 out of 3 who were homozygous (G/G) and the one patient that was heterozygous (C/G) for the risk allele at rs12973192 showed inclusion of the cryptic exon in almost every UNC13A mRNA that was mapped to intron 20-21. In contrast, the other patients who were homozygous for the reference allele (C/C) showed much less inclusion of the cryptic exon (Extended Data Fig. 7 a and b).

- This amounts to a form of eQTL analysis but with a very small number of patients and without any detail on the control of common co-variables which would be expected for standard eQTL analyses and so it is hard to assess its value. I would advise the authors to either add the relevant detail and make explicit the caveats.

Two of the iPSC-MN lines that we used to detect cryptic exon inclusion upon TDP-43 knockdown (Fig. 1h, iPSC-MN1 and iPSC-MN3) are heterozygous (C/G) at rs12973192. We sequenced the RT-PCR product that spans the cryptic exon and analyzed the allele distribution from these two samples as well as the one patient sample from the original RNAseq dataset (Fig. 1a) that is heterozygous (C/G) at rs12973192 (Extended Data Fig. 7b). We found a significant difference between the percentage of C and G alleles in the spliced variant, with higher inclusion of the risk allele (Fig. 4b and Extended Data Fig. 7c).

- This is an ASE analysis performed in the manner in which ASE was originally used. Of course a lot hinges on the design of the primers as to the validity of this approach and I think the authors should make some comment on how this was approached to prevent biasing of the results.

The rest of the patients (n=85) have exactly the same number of risk alleles at both loci, indicating that it's very likely the patients are carriers of the reference haplotype or the risk haplotype. Using a linear regression model, we found a strong correlation between the number of risk haplotypes and the abundance of UNC13A cryptic exon inclusion (Fig. 4c and figure using raw data, Extended Data Fig. 7d). Taken together, these data suggest that genetic variation in UNC13A that increases risk for ALS and FTD in humans promote cryptic exon inclusion upon TDP-43 nuclear depletion.

- This is an eQTL analysis conducted in patient samples (has been done for AD, MDD etc) in which case it is subject to all the issues which are normally a concern ie. controlling for known and unknown co-variables, genetic covariates and reaching genome-wide significant p-values. This makes the p-value attained is not particularly impressive and the necessary details to assess this are not present in the manuscript as things stand, I would have to hunt through the code. I appreciate that the N isn't high so that might explain the p-val but then I think the authors need to recognise the limitations of this analysis.

(h). RPLP0 was used to normalize RTqPCR. (Linear mixed model, **** $P < 0.0001$; mean \pm s.e.m.). (f and i) Immunoblotting for UNC13A and TDP-43 protein levels in SH-SY5Y cells (f) and iPSC-MNs (i) treated with scramble shRNA (shScramble) or TDP-43 shRNA (n=3). GAPDH served as a loading control. (g and j) Quantification of the blots in (f and i), respectively (two-sided Welch Two Sample t-test, * $P < 0.05$, ** $P < 0.01$).

- What's the basis for the choice of RPLP0 for normalisation and why weren't multiple genes used for normalisation which is the usual way this type of analysis is done and in fact was done by the

authors in another section of the paper where GAPDH was also used.

FTD and motor neuron disease (MND). (a) The design 5 of the UNC13A e20/CE BaseScope™...

- When running the BaseScope analysis are there any attempts to control for common covariates likely to impact on detection eg. sex, age, RIN of the tissue etc. This may have occurred but it is not clear to me from the methods.

Methods: RNA-Seq alignment and splicing analysis Detailed pipeline v2.0.1 for RNA-Seq alignment and splicing analysis is available on <https://github.com/emc2cube/5> Bioinformatics/sh_RNAseq.sh.

- Given the key importance of the RNAseq data analysis to this paper, there is simply insufficient detail in the methods on the data sets used to run analysis and the analysis itself. I appreciate that the github link is present but that doesn't negate the need for more detail within the main manuscript.

-

Methods: MAJIQ – “Here, de novo refers to junctions that were not in the UCSC transcriptome annotation, but had sufficient evidence in the RNA-Seq data (--min-intronic-cov 1).

- I assume tis means that a single intronic read was sufficient but it would be useful to have this more explicitly explained

Methods: LeafCutter (commit 249fc26 on <https://github.com/davidaknowles/leafcutter>): Using the already aligned RNA-Seq reads as previously described, reads that span exon-exon junction and map with a minimum of 6 bp into each exon were extracted from the alignment (bam) files using filter_cs.py with the default settings. Intron clustering was performed using the default settings in 30 leafcutter_cluster.py. Differential excision of the introns between the two conditions (TDP-43-positive neuronal nuclei vs. TDP-43-negative neuronal nuclei) were calculated using leafcutter_ds.R

- Are the settings for exon-exon junction reads used by LeafCutter and MAJIQ filtered the same way eg. overhangs into the exon, limits on the size of the implied intron etc.

- From this information and that in the main manuscript it appears to me that you are filtering MAJIQ output for the delta psi but not the LeafCutter output, why and what is the justification for this approach?

- How are you matching the output from MAJIQ and LeafCutter? The clustering will be different and direction of effects could be different so this is not trivial.

Quantification of UNC13A splice : UNC13A contains more than 40 exons and RNA-Seq coverages of mRNA transcripts are often not uniformly distributed 43, we looked at reads spanning “Exon 19-Exon 20” junction, which is included in both the canonical isoform and the splice variant, and there is a strong correlation (Pearson’s $r = 0.99$) between the numbers of reads mapped to “Exon 19-Exon 20” junction and “Exon 20-Exon 21” junction. We observed that samples that have at least 2 reads spanning either “Exon 20-CE” junction or “CE-Exon 21” junction have at least either UNC13A TPM = 1.55 or 20 reads spanning “Exon 19- Exon 20” junction. Therefore, we selected the 1151 samples that had a TPM ≥ 1.55 , or at least 20 reads.

- This comes across as a slightly dirty analysis, why use two different ways of filtering read depth which are correlated but not equivalent? In particular, I don't see the value of the TPM cut off. More generally, I think this analysis comes across as overly complicated and there is insufficient detail.

Referee #2

In this study, Ma, Prudencio, Koike and colleagues report the inclusion of a novel cryptic exon (CE) in the gene UNC13A, an RNA target of TDP-43, in a subset of FTD and ALS postmortem patient samples, which results in a dramatic decrease of UNC13A mRNA and protein levels. Using a bioinformatic approach and experimental validation in cells, they show that CE inclusion is dependent on TDP-43 loss of function and is increased by two UNC13A small nucleotide polymorphisms (SNPs), previously shown to be associated with increased ALS and ALS-FTD risk. They also identify a novel UNC13A haplotype within the same intronic site that they propose is a

potential risk for developing these diseases. Collectively, the work shows that the UNC13A CE inclusion is a marker of TDP-43 dysfunction in TDP-43 proteinopathies and indicates a direct link between TDP-43 loss of function and disease.

Following the recent discovery of the abnormal inclusion of STMN2 CE upon TDP-43 loss of function and the resulting loss of STMN2 mRNA and protein with detrimental effects for neurons (Melamed et al, 2019; Klim et al, 2019), the current work supports a mechanistic link between TDP-43 pathology and defects in key neuronal proteins. The findings are novel and exciting as they set the stage for several applications, including potentially UNC13A-based diagnostic or patient stratification tests, as well as therapeutic approaches for these diseases.

The manuscript is clearly written and the experiments are carefully designed and well-controlled. Due to the strong relevance of these findings for human disease and the high quality of the study, I enthusiastically recommend this work for publication in Nature, after revision to address the following concerns:

1. My main concern is the lack of functional validation of the effect of UNC13A decrease in adult human neurons. While I appreciate that this is technically challenging, I think it is important to show that the direct effect of TDP-43 loss of function on UNC13A levels has significant consequences for neuronal maintenance and function.
2. The effect of the UNC13A SNPs on TDP-43 binding is not explored in this study, as the authors state that in most cases SNPs are just markers and not the actual genetic defect. Yet, the accompanying study by Brown et al showed that one of the previously identified SNPs (rs12973192) leads to a significantly lower binding affinity for TDP-43 than the wild type sequence (Fig. 4F-I, Brown et al), while, in contrast, there is enhanced TDP-43 binding on rs12608932 SNP (Fig. 4I, Brown, et al). In light of this somewhat puzzling result, I think it is very important that Ma and colleagues experimentally test the effect of the current SNPs and their novel haplotype rs56041637, on TDP-43 binding on this region.
3. As stated in the manuscript, rs12973192 and rs12608932 SNPs are mainly associated with increased ALS risk (van Es et al., 2009; Nicolas et al., 2018), but also shown to contribute to FTD in sporadic ALS (Placek et al., 2019). However, CE inclusion seems to be more frequent in FTLD-TDP (Figure 2) than in ALS/FTLD or ALS-TDP (Figure S5). How do the authors explain this result?
4. The authors hint at the possibility that UNC13A, STMN2 and potentially other cryptic exon splicing events may "contribute to heterogeneity in clinical manifestation of TDP-43 dysfunction", which is indeed an attractive hypothesis. In this context, I am wondering whether the apparent more frequent detection of UNC13A CE inclusion in cortical samples (compared to spinal cord samples) not only from FTLD-TDP (Figure 2), where this is expected but also from ALS/FTLD and ALS-TDP patients (Figure S5) may suggest that UNC13A has a more important role in the function of cortical neurons compared to motor neurons. Providing any evidence for or against this hypothesis will significantly strengthen the study, in my opinion.
5. Linked to the previous point, FISH experiments show the presence of CE inclusion and reduction of UNC13A mRNA in cortical samples (Figure 3 and Figure S6). Are these also present in affected motor neurons of ALS patients? Or is it possible that motor neurons are more resistant to UNC13A CE inclusion, upon TDP-43 aggregation and nuclear clearance? If that were the case, it would explain the more frequent detection of UNC13A CE in FTLD cases.
6. I find the detection of UNC13A CE foci very convincing and I applaud the authors for setting up this challenging assay. Given that the CE-containing transcript is expected to be unstable, I find it curious that their detection was possible at all and that it was confined to the nucleus. I am wondering if their presence and localization may be linked to errors in nucleocytoplasmic transport in human tissues that are now well documented in these diseases. One way to answer this

question would be to test if the UNC13A CE foci are also present in cultured cells after knockdown of TDP-43.

7. For the initial identification of UNC13A CE the authors using two different pipelines for splicing analyses (MAJIQ and LeafCutter) and found a rather small fraction (<50%) of overlapping alternatively spliced genes. What is the reason for this discrepancy? Are there specific advantages, or drawbacks to either of these pipelines? Despite the confirmed STMN2, UNC13A and POLDIP3 the question is important in light of additional events that are potentially relevant, but maybe were only identified by one of the two pipelines.

8. The authors use bulk RNA-seq data to find UNC13A CE inclusion and to determine the specificity of this event to FTLD and ALS cases with TDP-43 pathology (Figure 2 and S5). To validate their pipeline, which is based on an arbitrary cutoff, it would be important to show that with this same method they can identify other alternatively spliced events, known to be linked to TDP-43 loss of function in the same datasets, such as STMN2 CE inclusion, as they did in Figure 1b.

9. It is not clear to me which SNP(s) are included in the two reporter minigenes that the authors developed. Is it all three at the same time? Different minigenes with either rs12973192, rs12608932, or rs56041637 alone would give information on the role of each SNP in increasing CE inclusion. These would also be useful in the context of point #2 above.

10. The authors state that "our functional studies indicate that TDP-43 dysfunction is required for UNC13A cryptic exon inclusion [...] the UNC13A risk alleles exert a TDP-43 loss-of-function-dependent disease-modifying effect". Said functional studies consisted of downregulating TDP-43 levels or transiently overexpressing an RNA-binding TDP-43 mutant. Perhaps more relevant would be to test the levels of UNC13A CE inclusion in a model of TDP-43 misregulation at endogenous TDP-43 levels. For example, it was recently shown that inhibition of deacetylases leads to increased acetylation of the RRM of TDP-43 disrupting the RNA binding properties of the protein (Ye et al, Science 2020).

Minor concerns:

- I recommend the authors to show the precise location of the previously reported TDP-43 binding site on the UNC13A intron in Figure 1d. This will be very helpful for the readers. I note that the binding motifs that are currently highlighted are not necessarily identical with the binding site.
- Page 3, line 11: reference to Figure 1f should also contain Figure 1g (its quantification), which is otherwise not referenced in the text.
- Extended Data Fig. 2a: the conservation of UNC13A among mammals is not clear from the image. Do black boxes represent conserved residues? Overall this figure is very difficult to read.
- I encourage the authors to make all raw data, code, and materials used in the study publicly available upon publication and not only upon request. I also encourage them to add more details on their analyses whenever possible for full transparency. For example, I could not find the filter_cs.py file. Is this available? Is it part of LeafCutter?

Referee #3

Amyotrophic lateral sclerosis (ALS) and frontotemporal dementia (FTD) are two neurodegenerative disorders with a common etiology; the RNA binding protein TDP-43 undergoes a shift from a nuclear- to a cytosolic localization, thereby losing its ability to function as a splicing repressor. The downstream effect involves mis-splicing events, for example the unwanted inclusion of so-called cryptic exons (CEs) in mature RNA. These CEs would normally be spliced out, and their inclusion results in downstream errors in protein homeostasis.

Munc13-1 is a central presynaptic protein. It is involved in a preparation step synaptic vesicles must undergo to make them release competent. Munc13-1 is the central isoform in the majority of synapses at the central nervous system (CNS), as well as in the neuromuscular junction. Its knock-out in mice is a cause for one of the most severe phenotypes in synaptic biology; in excitatory hippocampal neurons, it results in a 90% block of neurotransmission, and the concomitant removal of Munc13-2 leads to a complete block of neurotransmitter release, in both excitatory and inhibitory CNS neurons (Augustin et al., 1999; Varoqueaux et al., 2002). Recently, Munc13-1 has been shown to be arranged in clusters that are thought to represent discrete SV release sites (Reddy-Alla et al., 2017; Sakamoto et al., 2018). The number of clusters is associated with the strength of the synapse; synapses with many clusters will release more glutamate, making them stronger, whereas synapses with less clusters will release less glutamate, making them weaker. Work in the *Drosophila* neuromuscular junction also showed that the amount of the Munc13-1 homolog UNC13A is scalable under plasticity (Bohme et al., 2019). Current interpretations of these data suggests that the levels of Munc13-1 in synapses may regulate the strength of neurotransmitter release. In addition to synaptic strength, Munc13-1 is a regulator of synaptic plasticity, translating elevations in Ca²⁺ and dynamic changes in membrane lipid concentrations into changes in the efficacy of neurotransmission during activity (Lipstein et al., 2013; Shin et al., 2010). Finally, two cases of human brain disease have been described, with the loss of Munc13-1 leading to a severe brain disease and premature death (Engel et al., 2016), and one variation described to result in a neurodevelopmental and a neuropsychiatric disorder (Lipstein et al., 2017).

A genetic association between non-coding SNPs in the UNC13A gene locus and ALS has emerged over a decade ago. Interestingly, the SNPs have been associated with increased disease risk and with increased disease progression, which makes the mechanism by which these contribute to disease significant and important to elucidate. To date, however, the molecular and cellular mechanisms linking these SNPs to the disease, and, more importantly, whether these indeed involve the Munc13 gene product, remained elusive.

In this exciting manuscript, Ma and colleagues establish these mechanisms and links. They demonstrate that the nuclear loss of TDP-43 leads to a CE inclusions in the UNC13A RNA, resulting in substantial reduction in protein expression. The supporting data includes:

- A splicing analysis of a previously published RNA-Seq dataset identified 65 mis-spliced transcripts, amongst which a highly significant hit is the UNC13A RNA. The major mis-splicing event is the inclusion of an CE between exon 20 and 21 (Figure 1).
- TDP-43 directly binds to the intronic sequence that includes the CE (Figure 1).
- In vitro depletion of TDP-43 in three cell lines leads to inclusion of the CE, reduction of mRNA levels and protein abundance (Figure 1).
- RNA analysis in patient-derived material identifies the CE inclusion as a common event specific to TDP-43-mediated disease (Figure 2).
- The levels of phosphorylated TDP-43, which are correlated with its abundance in the cytoplasm, are positively correlated with the levels of CE inclusion (Figure 2).
- In situ hybridization identifies nuclear RNA transcripts of UNC13A including the CE in patient-derived material (Figure 3). Canonical UNC13A mRNA is reduced in TDP-43 affected cells from patient tissue.
- The presence of risk SNPs, that are located in close proximity to the CE, is strongly associated with CE inclusion in mRNA, also in patient material (Figure 4).
- Identification of yet another risk region (rs56041637), associated with the two SNPs.
- Minigene assay demonstrating TDP-43-dependent and risk SNPs extravagated mis-splicing of the CE.

The paper is clear, precise, very well-written, and exciting. The experimental array used and the conclusions drawn appear justified and robust to me, although I am not an expert in the full spectrum of methodologies used, and others may be better suited to determine whether all

controls are included.

The major novelty is the univocal demonstration of a functional relationship between TDP-43 and Munc13-1. This exciting finding is highly significant, as it is associated with SNPs that exert a substantial effect on disease course. Because Munc13s are primarily synaptic proteins, the data presented here draws a strong connection between presynaptic dysfunction and the neurodegenerative disorders ALS and FTD, expanding the already identified link between presynaptic function and neurodegeneration in tauopathies and alpha-synucleinopathies. In fact, some data the authors decided not to follow up at this point (Fig 1b, see comment below) makes this link very strong. Notably, the clear demonstration that the RNA irregularities in the UNC13A transcript can be found in patient-induced material is extremely convincing of the relevance of the finding. This paper opens a new avenue for mechanistic studies of ALS/FTD in synaptic biology and its findings are likely to be followed up by many.

A major element that the current paper does not address is whether and how reducing the expression levels of the Munc13 proteins may lead to cellular pathology and/or to disease course modulation. It is important to note, that while a relationship between the reduction of Munc13-1/2 expression has been functionally established in hippocampal and striatal neurons (no Munc13s = no synaptic transmission (Varoqueaux et al., 2002)), and in the neuromuscular junction (no Munc13s = changes in synaptic transmission (Varoqueaux et al., 2005)) no such relationship has been established with regards to cell survival. In fact, while mice lacking the two isoforms die at birth due to a block of synaptic transmission, their CNS appears fully intact (Varoqueaux et al., 2002). In CNS-derived cultures or in organotypic slices in vitro, no degenerative effects have been observed (Broeke et al., 2010; Imig et al., 2014). In a single human patient with a Munc13-1 protein loss, neuromuscular architecture was fully preserved (Engel et al., 2016). Moreover, in mice, heterozygosity results in 50% reduction in protein abundance, but not in observable changes in synaptic function or in animal behavior, although this has not been studied in great detail (Augustin et al., 1999; Varoqueaux et al., 2005). Because conditional mouse models to reduce the expression of Munc13 isoforms in the adult nervous system (to levels lower than 50%) have not been published yet, the effects of such reduction at the adult nervous system have not been studied and are difficult to address. As such, this goes beyond the scope of the current manuscript. I would therefore recommend the paper for publication, provided that the following major points will be addressed:

1. Presynaptic dysfunction as a disease-modifying element: In addition to Munc13-1, there is a surprisingly high number of central presynaptic proteins that appear to be mis-spliced (Fig. 1B: CADPS, ERC2, STXB1, STBP5L, Syt7, several of the GEFs, are just those that I identify, see Taoufiq et al., PNAS 2020 or Koopmans et al., Neuron 2019). This is fascinating - all these proteins have been demonstrated to exert a major role in presynaptic function. A short description of the genetic nature of the mis-splicing events would be highly relevant to the synapse biology community and support the conclusion that presynaptic dysfunction is emerging a major mechanism in the pathogenesis of ALS/FTD.
2. Is the UNC13A CE introducing a stop codon? What is the mechanism to RNA decay? The statements in page 4 line 22 are not clear. Also, the supplemental note indicating three types of CEs is important and should be referred to in the main text.
3. In the minigene assay, how many CATC repetitions are included? Is there also a quantitative relationship there? Is this repeat affecting TDP-43 binding to that region? Or is it this just another marker?
4. It is not clear whether and how the survival time analysis takes into account the third, newly-identified risk region. If not, the novelty of this analysis is not clear in view of the references mentioned by the authors. Please clarify
5. Can the authors provide some evidence for reduced protein expression in patient-derived material?
6. The authors clearly demonstrate that TDP-43 pathology is necessary for detecting CE inclusion, and that in healthy individuals such inclusion cannot be seen. They also demonstrate that the

fraction of CE-containing transcripts is increased as a function of increased pTDP-43 levels. Can the authors provide data for the time course of CE inclusion in patients, to test how this relates to disease progression? It would be fascinating to align such data to the time course of cell degeneration.

Referee #4

In the manuscript "TDP-43 represses cryptic exon (CE) inclusion in FTD/ALS gene UNC13A", Ma et al. reveal that UNC13A gene is a target of the splicing regulator TDP-43. Loss of TDP-43 from the nucleus resulted in the inclusion of a cryptic exon splicing event in UNC13A transcript and reduced UNC13A protein expression. In addition, the authors showed that UNC13A genetic variants strongly associated with FTD/ALS risk are within the CE region, and these variants increase CE splicing when TDP-43 dysfunctions.

General Comments:

Overall, this study demonstrates a strong connection between UNC13A genetic variants and loss of TDP-43 function. The results are of immediate interest to people in neurodegeneration field. In general, the manuscript is clearly written and the findings are novel. The study provides a new explanation of how UNC13A genetic variants increase ALS/FTD risk and change survival. My main remaining question is to what extent this UNC13A CE event contributes to ALS/FTD pathology, which may be beyond the scope of this work.

Specific comments:

Figure 1F, I wonder whether a double band was detected for UNC13A protein after shTDP-43 KD? What is the estimated protein size for UNC13A CE transcript?

Figure 4g, Ext Figure 8b May the authors switch to two-way ANOVA to estimate the means of qPCR readout based on two categorical variables (siRNA and GFP constructs)?

Ext Figure 6c Please add a plot with quantitative measurement (e.g., Figure 3c).

"To discover cryptic splicing targets regulated by TDP-43 that may also play a role in disease pathogenesis, we utilized a recently generated RNA sequencing (RNA-Seq) dataset... Here, we re-analyzed the data to identify novel 30 alternative splicing events impacted by the loss of nuclear TDP-43. We performed splicing analyses using two pipelines, MAJIQ 12 and LeafCutter 13, designed to detect novel splicing events (Fig. 1a)." Comprehensive data mining of publicly available NGS datasets using the latest bioinformatics approaches is critical to building this paper's hypothesis. The authors stated that "All data used in this study are available upon request. All codes used in this study are available upon request." Can the authors upload their data processing codes to the GitHub repository so that the readers can repeat the published results (recommended by Nature Portfolio journals)? Also, I wonder whether the authors can release their own RNAseq data (e.g., a series of 117 frontal cortex brain samples from the Mayo Clinic Brain Bank) to a data portal?

Minor comments:

"we scored the UNC13A splice variant as present if there were more than two reads spanning at least one of the exon-exon junctions." Are those reads uniquely mapped?

Ext Figure4 Can the authors use one-way ANOVA test? Looks like there is no significant difference between sgTDP-43-guide1 and sgTDP-43-guide2, is that correct?

Referee #5

A. In this study, Ma et al mine the Liu et al RNAseq dataset from TDP-43 mutant brain and identified incorporation of an out-of-frame CE unexpectedly incorporated in UNC13, a gene previously implicated in FTD/ALS through GWAS. The GWAS SNPs occur proximal to the CE, and Ma demonstrates in human brain, neuronal cell lines, and iPSC derived motor neurons that there is incorporation of this unusual CE. Most convincing is that patient FTLT-DTP but not other genetic forms of FTLT samples show CE inclusion using two methods. Their data suggest that UNC13A risk SNPs, or a 4bp STR in linkage disequilibrium, increase the rate of CE incorporation following TDP-43 depletion.

B. This study links several previous studies in meaningful ways, and helps explain why the UNC13A SNPs are disease-associated. This may be one of the first examples of a GWAS peak that is later discovered to mediate to cryptic exon incorporation.

C. In general the approaches appropriately support the conclusions. Quality of data is excellent. The presentation could be further improved, as suggested below.

D. Fig 1b, all the genes are $P(\Delta\Psi > 0.1) > 0.95$. Is this what authors define as highly significant? Or does UNC13A have a higher p-value indicated by the color? Other parts are clearly documented and minor suggestions are attached below.

E. The authors conclusion from data mining of existing data and in vitro analysis is robustly presented in large-scale in vivo screening in patient-derived samples. In general the data are valid and reliable, with minor suggestions below.

F.

1. The minigene reporter assay would be more informative if the specific risk variants could be further functionally interrogated to determine which one(s) bias CE incorporation. Does the critical variant influence TDP-43 binding in the author's hands?

2. The finding that the SNPs affect survival in FTLT-DTP patients is positive, but it would be interesting to understand if they have no effect on survival in other forms of disease. If they still have an impact would the authors consider that there are genes other than TDP-43 influencing CE incorporation?

3. Does population VAF for UNC14B risk (maybe you can find them in 1000Genome/gnomAD) correlate with the prevalence rate of ALS/FTD in different populations (PMID 22231873, 30879893, and 27185810)?

4. Page 3 Line 15, Data Fig. 3a and 3b, the authors mention direct sequencing, but the figure shows only RT-PCR results.

5. Extended Data Fig. 4. Both bars with significant changes appear to refer to guide2. Was guide1 non-functional?

G. Recommend indicating in Fig. 1a the data is generated in Liu et al. PMID 31042469.

H.

-Extended Fig 2a the authors might want to highlight where the CE is?

-Page 3 Line 45, are these calculated with the same criteria?

-Fig. 2c and Extended Data Fig. 5c are probably redundant as log-transformation does not change the rank so Spearman correlation should have exactly the same rho and p-value.

-Why SNP genotyping instead of direct Sanger sequencing for genotyping the SNPs?

-There are only 600 bp between rs12608932 and rs12973192, so why is not possible to genotype the entire haplotype with a single Sanger reaction?

- Ref 38, 42 has a different ref format.

Author Rebuttals to Initial Comments:

Point-by-Point Response to Referees' Comments

Our responses are in blue.

Introduction to the Revision

We thank the Referees for their rigorous and enormously detailed and helpful comments, which we have considered carefully. We note that all five Referees appreciated the excitement, significance and impact of our study and raised key comments and suggestions to help us strengthen the conclusions presented in our paper. Their valuable input led us to perform additional experiments and controls that have now increased our confidence in the conclusions that we present in the revised manuscript. We also note that the accompanying manuscript by Brown et al, which we had not read before initial submission, reports findings remarkably concordant to ours, further underscoring the robustness of the results. Together, both papers now provide compelling evidence directly connecting one of the most common genetic risks factors for FTD/ALS (*UNC13A* polymorphisms) to the most common pathology (TDP-43 mislocalization). In addition to responding in detail to each of the Referees' comments below, here are some of the major additions that we have incorporated into the revised manuscript:

1. In addition to *UNC13A*, we validated several other pre-synaptic cryptic splicing targets of TDP-43. Referee #3 pointed out the exciting possibility of presynaptic dysfunction as a disease-modifying element and noted that, "In addition to Munc13-1, there is a surprisingly high number of central presynaptic proteins that appear to be mis-spliced (Fig. 1B: CADPS, ERC2, STXBP1, STBP5L, Syt7, several of the GEFs, are just those that I identify, see Taoufiq et al., PNAS 2020 or Koopmans et al., Neuron 2019). This is fascinating - all these proteins have been demonstrated to exert a major role in presynaptic function. A short description of the genetic nature of the mis-splicing events would be highly relevant to the synapse biology community and support the conclusion that presynaptic dysfunction is emerging a major mechanism in the pathogenesis of ALS/FTD." We too are excited by these findings. In Extended Data 6, we now present validation experiments that we performed for 3 additional pre-synaptic cryptic splicing targets (*KALRN*, *RAPGEF6*, and *SYT7*) in human iPSC-derived neurons, and include the specific details about these missplicing events. These are the ones that we are most interested in but we strongly suspect that others in the field will now want to follow up on these and others and that the focus on pre-synaptic function in ALS/FTD will open up new directions for the field.

2. Referees #2 and #5 both asked us to extend our minigene reporter construct experiments. In the initial manuscript, we tested two versions of these, one with part of the *UNC13A* gene with the reference haplotype (reference versions of the 3 SNPs) and one with the risk haplotype (risk versions of the 3 SNPs). We found diminished canonical splicing and increased cryptic splicing in the construct harboring the risk haplotype (Fig. 4). The Referees asked us to test all different SNPs individually in this assay. We did this as suggested and we also tested all permutations (single SNPs, doubles, and all 3). In total, we tested 8 constructs in the revised manuscript. We show these new data in Extended Data Fig. 10, which indicate that the risk variant at rs12973192 is the strongest contributor to cryptic exon inclusion but the other ones do contribute and are not simply additive (Extended Data Fig. 10). Because of linkage disequilibrium and the general principle of linked variants, we were and still are hesitant to pin all of the effects on one SNP and these new results support that notion. Our findings indicate

that the *UNC13A* risk haplotype that is associated with increased risk for ALS and FTD/ALS causes increased cryptic splicing in the face of TDP-43 loss of function. The lead GWAS SNP that is located right in the cryptic exon itself (rs12973192) can impact cryptic splicing by itself but there are likely additional variations contributing to cryptic splicing. Future efforts by us and others to fine map the locus will be informative.

3. In the accompanying manuscript by Brown et al, the authors perform several analyses of TDP-43 binding to the *UNC13A* intronic region harboring the risk variants. Referee #2 pointed out, “the effect of the *UNC13A* SNPs on TDP-43 binding is not explored in this study, as the authors state that in most cases SNPs are just markers and not the actual genetic defect. Yet, the accompanying study by Brown et al showed that one of the previously identified SNPs (rs12973192) leads to a significantly lower binding affinity for TDP-43 than the wild type sequence (Fig. 4F-I, Brown et al), while, in contrast, there is enhanced TDP-43 binding on rs12608932 SNP (Fig. 4I, Brown, et al). In light of this somewhat puzzling result, I think it is very important that Ma and colleagues experimentally test the effect of the current SNPs and their novel haplotype rs56041637, on TDP-43 binding on this region.” To address this suggestion, we performed quantitative electrophoretic mobility shift assays (EMSA). We used purified recombinant TDP-43 and performed quantitative electrophoretic mobility shift assays (EMSA) with six different *UNC13A* probes (cryptic exon reference SNP vs risk SNP; intron reference SNP vs risk SNP; intron reference repeat vs risk repeat). The probes we used were slightly different than the ones used by Brown et al (ours are longer). These new data are in Fig. 4g and Extended Data Fig. 13. The quantified apparent binding affinity ($K_{D,app}$) reveals that while TDP-43 binds the reference and risk versions of the cryptic exon with similar affinity, the intronic risk allele results in a minor reduction in affinity (Fig. 4x). TDP-43 had lower affinity for a probe containing the longer repeat version compared to the reference short repeat sequence (Fig. 4g). Overall, we find in these *in vitro* binding assays that TDP-43 has a much higher affinity for the intron sequence compared to the exon or repeat sequence. The diminished binding affinity of TDP-43 to risk alleles of the intron and repeat sequence may contribute to the increased cryptic exon splicing found in ALS and FTD. Future studies will be required to explore how TDP-43 regulates the cryptic splicing of *UNC13A* and other splicing targets and the impact of different genetic variations on TDP-43 binding *in vivo*. These results are somewhat different than those reported in the accompanying manuscript but the experiments are slightly different (we use full-length TDP-43, which on one hand can be aggregation prone (though our MBP-tagged TDP-43 is soluble) and the other team does not (they use a construct only containing TDP-43’s RRM). In any case, we do not want to over interpret results from *in vitro* binding studies, especially with SNPs that are common variants, which would necessarily be expected to have small effects, which could still be meaningful in terms of lifetime risk for FTD/ALS. We have added these new data to the revised manuscript and are cautious about interpretation.

Referee #1

TDP-43 represses cryptic exon inclusion in FTD/ALS gene *UNC13A* (Rosa et al., 2021) is an interesting manuscript, which potentially provides a mechanistic understanding of one of the established risk loci for FTD/ALS and directly links it to the function and deposition of TDP43 within the brain tissue of affected individuals. There are relatively few examples where this

form of molecular understanding has been achieved for complex diseases and therefore this paper will generate considerable interest. However, it will be of greatest interest to those focused on TDP43-related diseases and those interested in the development of novel therapies in the field.

We thank the Referee for appreciating that we have provided a mechanistic understanding of an FTD/ALS genetic risk factor and connected it to the key pathology in the diseases (TDP-43). We also thank the Referee for appreciating that this work will be of considerable interest and has direct relevance for development of therapeutic strategies.

1. It is not clear to me the extent to which these findings or the approach taken are generalisable beyond this specific disease and within that the subtype characterised by TDP43 deposits.

In addition to providing an understanding of the established risk loci in FTD/ALS, we hope this manuscript also illustrates the importance of understanding GWAS hits in the context of diseases. Since the exact etiology of many diseases are unclear, one can hypothesize that many GWAS hits that are mapped to non-coding regions of the genome may only exert their pathogenic effects in conjunction with certain perturbations (in the case of FTD/ALS: TDP-43 mislocalization from the nucleus to the cytoplasm) and cause changes in splicing regulation, gene expression, chromatin structures, etc. We refer to this phenomenon in the discussion portion when we present the concept that the *UNC13A* risk SNP may act as an Achilles' heel. In any case, since TDP-43 mislocalization is the universal pathology in nearly all ALS cases (~98% of cases) and nearly half of FTD cases, we think that these findings are very relevant to a broad number of these cases. Also, TDP-43 pathology is becoming appreciated in Alzheimer disease and even an aging-associated neurodegenerative disease (LATE) and we hope that our findings (and others in the field) will motivate studying emergent roles of cryptic splicing in these diseases and others.

2. In terms of further experimental work which I think would improve this study, it is now possible to run targeted long read RNAseq which could be performed across samples and which could allow more accurate delineation of the proposed abnormal *UNC13A* transcript/s containing the CE and the relative usage of transcripts. I think the inclusion of this form of data would significantly add to the work.

Thank you for your suggestion. We agree that targeted long read RNA-Seq would give us accurate information about the transcripts containing the CE and it would be very useful for testing other potential splicing targets. However, we did perform amplicon sequencing using the Illumina 2 × 250 bp platform. The locations of the primers used for amplification and the alignment results have been illustrated in Extended Data Fig. 5. and the sequencing files have been uploaded to GSE182976 . Since the expected size of the amplicon with the cryptic exon inclusion is 355 bp or 405 bp, the read length is sufficient to characterize the relative usages of the transcript with or without the CE.

3. I also think it would be valuable to go back to iPSC/mouse models of pathogenic Mendelian forms of FTD/ALS due to TDP43 mutations. It strikes me that variability in the severity of the disease could be explained amongst these patients by variability at the *UNC13A* locus and this would substantiate the findings.

Thank you for this suggestion. It would have been an interesting aspect to explore. However, we cannot perform the experiments in the mouse models in this case. As illustrated in Extended Data Fig 2., the *UNC13A* cryptic exon containing intron is not conserved in mice (similar to other TDP-43-regulated cryptic splicing targets, like *STMN2*). Therefore, it is challenging to recapitulate the cryptic exon inclusion in existing mouse models. In terms of iPSC models of Mendelian forms of FTD/ALS, it seems that the Referee is suggesting that we perform an experiment to compare the extent of *UNC13A* cryptic exon inclusion across different iPSC neuron models that have different causative mutations (e.g., *TARDBP*, *C9orf72*, etc.) and then attempt to correlate these to disease severity in the patients from which the cells were obtained. This is an interesting suggestion though there has not been a good model that can recapitulate TDP-43 depletion from the nucleus. However, it is possible that the TDP-43 with point mutations could still cause splicing defects. While we do think it could be interesting to explore whether the variability in the severity of the disease could be explained by the variability at the *UNC13A* locus, due to the inherent limitations of the iPSC models and a plethora of other confounders (age, genetic modifiers, comorbidity, etc.), we think it would be better to test the hypothesis in a well-controlled study on real patients, which is slightly out of scope.

4. “To identify changes associated with loss of TDP-43 from the nucleus, the authors used fluorescence-activated cell sorting (FACS) to enrich neuronal nuclei with and without TDP-43 from postmortem FTD/ALS patient brain tissue and then performed RNA-seq to compare the transcriptomic profiles between TDP-43-positive and TDP-43-negative neuronal nuclei. They identified a multitude of interesting differentially expressed genes 11.”

- The validity of re-analysing this data set for differential splicing when the RNAseq data being used originates from the nuclei needs to be addressed. It may be that this is identifying a problem with the rate of RNA processing for the genes not an actual splicing defect through this analysis.

Thank you for pointing this out. It was indeed one of our concerns at the beginning, but we reasoned that first, since splicing takes place in the nucleus and often happens co-transcriptionally, we would detect final splicing products by looking at the data originated from the nucleus; second, if it is an artifact associated with the nucleus origin of the data, we would detect the splice variant in the nucleus of the control condition (nucleus with TDP-43) as well. We have used multiple independent methods that do not rely on analysis of RNA from nuclei, beyond the initial dataset to validate that there is a splicing defect in *UNC13A* upon TDP-43 dysfunction. These include, qRT-PCR of splicing products following TDP-43 knockdown in SH-SY5Y cells and in iPSC-MNs, direct sequencing of mature splicing products, splicing mini-gene reporter assays, qRT-PCR analysis of splicing products in bulk

RNA from patient samples, and *in situ* visualization of splicing products in control and FTD/ALS patient brain sections.

5. “There were 65 alternatively spliced genes in common between both analyses (Fig. 1b), 35 likely because each tool uses different definitions for transcript variations and different criteria to control for false positives. Among the alternatively spliced genes identified by both tools were STMN2 and POLDIP3, both of which have been extensively validated as bona fide TDP-43 splicing targets 8–10,14.”

- Was there any statistically significant overlap between the genes with splicing changes as identified by LeafCutter and MAGIQ? Also these aren't really two independent methods for assessing splicing changes given that they both use a fundamentally similar approach ie. only apparently exon-exon junction reads are analysed.

In the revised manuscript we have added a Supplementary Note to compare the two methods in detail. Briefly, despite both tools relying on junction spanning reads, the units of comparison used by the tools are different. MAJIQ identifies and compares LSVs (local splicing variations); LeafCutter, on the other hand, detects intron inclusions and collapses all overlapping intron to an “intron cluster” for which the inclusion levels are normalized and quantified. Compared to MAJIQ, LeafCutter also does not detect alternative first and last exons, or intron retentions. Since the goal of this study is not to compare the two tools, we did not intend to make any claims on how similar the outputs are. Therefore, we did not conduct statistical tests to show whether the overlaps in the outputs are significant. That said, it is very possible with careful tweaking of the parameters, we would get more overlap in their outputs. However, a detailed comparison between these two analysis methods is beyond the scope of this paper.

6. Furthermore, merging the output from these two tools is far from trivial and there is nothing in the methods to explain this. It is very possible that the two tools identify the same genes but because of an entirely different event. Has there been any attempt to map intron clusters?

We agree with this and have now included a Supplementary Table in which the outputs from the two tools are event matched and the coordinates of the corresponding regions are provided in the Supplementary Table 1. There are 66 overlapping genes, among which CASP contains 3 different regions, SETD5 contains 2 regions and SYNE1 contains 2 regions. Only 3 genes (APLP2, EPB41L1 and KIF3A) from the output of LeafCutter did not reach a delta PSI larger than 10% (which was the cutoff for the MAJIQ outputs). However, this can likely be attributed to the normalization in intron clusters.

7. Was there any overall enrichment of genes previously implicated in FTD/ALS within the gene list?

We did not see an overall enrichment of FTD/ALS genes, indicating that other previously implicated FTD/ALS genes may not be affected by splicing defects caused by the loss of

TDP-43. However, we did identify *UNC13A* (a major FTD/ALS risk gene) and, as we suggest in the discussion portion of the manuscript, we think that using our list of new TDP-43 cryptic splicing targets to look for additional genetic variations that increase risk of FTD/ALS could be fruitful. In other words, some of the other splicing targets we have discovered might turn out to be disease genes themselves and we are excited for us and others to test this hypothesis in the future.

8. “This new exon (CE, for cryptic exon) was absent in wild type neuronal nuclei (Fig. 1c) and is not present in any of the known human isoforms of *UNC13A* 15. Furthermore, analysis of ultraviolet cross-linking and 45 immunoprecipitation (iCLIP) data for TDP-43 3 provides evidence that TDP-43 directly binds to the intron harboring this cryptic exon (shown by mapped reads) (Fig. 1d)”

- It is not clear to me how absence was defined in the wild type neuronal nuclei? Given that data will inevitably be sparse in nature and account needs to be taken of sequencing depth, number of nuclei captured etc this may not be trivial and the methods do not provide the detail necessary to judge this.

Thank you for pointing this out. The absence is defined by the lack of RNA-Seq reads that span the exon 20-CE junction or the CE-exon21 junction as indicated by LeafCutter and MAJIQ. According to Liu et al (2019), RNA-Seq libraries of both TDP-43 negative and TDP-43 positive nuclei have ~75 million unique reads. In our paper, Fig. 1c is generated using the sashimi-plot function in the MISO package. Read densities across exons are scaled by the number of mapped reads in the sample and are measured in RPKM units. We have also incorporated the suggestion in the manuscript (Methods, p. 52).

Comment 9: “Reducing levels of TDP-43 in iPSC-derived motor neurons (iPSC-MNs) (Fig. 1 h to j; Extended Data Fig. 3 a and b) and excitatory neurons (i3Ns) derived from human iPS cells (Extended Data Fig. 4) also resulted in cryptic exon inclusion in *UNC13A* and a reduction in *UNC13A* mRNA and protein. We confirmed insertion of the 128 bp cryptic exon sequence into the mature transcript by direct sequencing of the RT-PCR product (* in Extended Data Fig. 3a). Thus, lowering levels of TDP-43 in human cells and neurons causes inclusion of a cryptic exon in the *UNC13A* transcript, resulting in decreased *UNC13A* protein.”

- It would be helpful to more clearly understand the relationship between i) the inclusion of the CE, ii) the drop in *UNC13A* levels and iii) the *UNC13A* protein levels. Given the fact that NMD will remove the CE transcripts, but this could trigger compensatory increase in *UNC13A* transcription, and in general correlations between RNA levels and protein levels are poor the protein level drops seem dramatic. To tease this out more carefully, I think correlations between these three different analyses should be performed.

This is an interesting point and the Referee is correct that RNA levels do not always reflect protein levels. So far, we do not have evidence for a compensatory increase in *UNC13A*

transcription when TDP-43 level is decreased. Indeed, we see a decrease in total *UNC13A* mRNA levels upon TDP-43 knockdown. We do think, however, that the regulation of *UNC13A* is likely more complicated than we had indicated in the original manuscript and there are certainly other levels of regulation beyond the CE. In the revised manuscript we have added more caution in interpreting the CE formation, decrease in canonical transcripts, and decrease in protein levels, to reflect the possibility for other levels of regulation (pg. 3). We also mention the possibility that other aspects of the CE (e.g., aberrant peptides produced from it) could cause defects, though we do not have evidence yet that these are produced.

This figure shows a time-course experiment we performed in neurons derived from human iPS cells. We showed that there is an increase in cryptic exon inclusion 5 days after knocking down TDP-43 using shRNA. Notably, the level of normal splicing, measured by primers detecting the canonical exon 20- exon 21 junction, and total *UNC13A* level started to decrease 3 days after shRNA transduction. The total *UNC13A* level was measured by a pair of primers that detect exon 20, meaning that it would detect both the mature transcript and the nascent transcript (the RNA had been treated with DNase to remove genomic DNA). We expect to see similar expression levels of total *UNC13A* from both the shTDP-43 knock-down and control conditions (shScramble) if there was a compensatory increase in *UNC13A* transcription. By day 14, we started to see similar levels of *UNC13A* normal splicing and total *UNC13A*. Rather than a recovery of *UNC13A* splicing and expression, we think it is more likely caused by death of cells that were most affected by knocking out TDP-43. In fact, there were not enough cells remaining in the culture for us to harvest enough protein for immunoblotting. However, we do not think iNs serve as an accurate representation of what happens in the human nervous system longitudinally. For example, in the manuscript we showed that there is a decrease in the canonical exon 20-exon 21 junction (Extended Data Fig.8) and an increase in

cryptic exon inclusion (Fig. 3) in almost all neurons exhibiting TDP-43 mislocalization.

10. To extend our analysis of *UNC13A* cryptic exon (CE) inclusion to a larger collection of patient samples, we first analyzed a series of 117 frontal cortex brain samples from the

Mayo Clinic Brain 30 Bank and found a significant increase in UNC13A CE levels in FTLD-TDP patients compared to healthy controls (Fig. 2a and Extended Data Fig. 5a).

- As mentioned before, I do not think there is sufficient detail in the methods to assess how this was done.

We have added more details to the manuscript and the Methods (“Post-mortem brain tissues for detecting UNC13A splice variant” section in Methods) (p. 57).

11. Owing to noise generated from bulk sequencing, we scored the UNC13A splice variant as present if there were more than two reads spanning at least one of the exon-exon junctions. We identified 63 samples, from 49 patients, which met the above criteria. We detected UNC13A splice variants in nearly 50% of the frontal and temporal cortical tissues donated by patients with neuropathologically confirmed FTLD-TDP (Fig. 2b).

Surely this analysis could be made more robust by actually just running LeafCutter or MAJIQ? Was this tried and didn't yield? If this is the case then I appreciate that detection of these abnormal junctions would be helpful, but then I think the authors should still be able to account for read depth specifically over the expected junction.

Thank you for pointing this out. In the revised manuscript we have now added a Supplementary Note explaining the rationale behind this analysis. Briefly, due to the heterogeneity of the bulk RNA-Sequencing data and the sparsity of the reads mapped to UNC13A CE, we cannot use LeafCutter or MAJIQ for detection. We show here that there is no significant difference between the read depth of samples with cryptic exon inclusion (CE samples) and samples without cryptic exon inclusion (non-CE samples) at the exon 19-20 junction and the exon 20-21 junction. And there's no significant difference between the TPMs of the CE samples and those of the non-CE samples.

CE samples vs. non-CE samples	P- value (Mann-Whitney test)
Exon 19-20 junction	0.3706
Exon 20-21 junction	0.4475
TPM	0.1180

12. We also detected the splice variants in some of the ALS patients whose pathology has not been confirmed (Extended Data Fig. 5b). Notably, 45 we did not observe UNC13A CE in any of the samples from FTLD-FUS (n=9), FTLD-TAU (n=18) and ALS-SOD1 (n=22) patients, nor in any of the control samples (n=197) (Fig. 2b).

- Without accounting for read depth across the canonical junction the significance of these findings is unclear.

Thank you for pointing this out. We compared the read depths at the canonical junction (exon 20-21 junction) of samples with cryptic exon inclusion (CE samples) and that of samples from the FTLD-FUS, FTLD-TAU, ALS-SOD1, and the control samples. The results are presented in the table below. Most of the pairs except “CE samples vs. control samples” do not have a significant difference in read depth. However, both the mean and median read depth of the control samples are higher than those of the CE samples (control samples vs. CE samples” mean: 136.8 vs. 102.3; median: 122 vs. 94).

	P- value (Mann-Whitney test)
CE samples vs. FTLD-FUS samples	0.7052
CE samples vs. FTLD-TAU samples	0.5017
CE samples vs. ALS-SOD1 samples	0.9256
CE samples vs. control samples	0.0100

13. To determine the relationship between pTDP-43 levels and UNC13A cryptic exon inclusion we analyzed data from 90 patients with FTLD-TDP in the Mayo Clinic Brain Bank for which we had RT-qPCR and pTDP-43 levels (Methods) from frontal cortices. We found a striking association between higher pTDP-43 levels and higher levels of UNC13A cryptic exon inclusion in patients with FTLD-TDP (Spearman’s rho = 0.572, P < 0.0001) (excluding 4 individuals who did not show 10 UNC13A CE signal) (Fig. 2c and figure using raw data, Extended Data Fig. 5c).

- I do not understand the reason for excluding the 4 patients. If those individuals have FTLD-TDP and have the data to run the analysis then the fact they don’t have the UNC13A CE does not invalidate them.

We completely agree and have included the 4 patients in the updated graph. We initially removed the 4 patients because having 0s in a data set makes log transformation tricky. However, since log transformation is not necessary here, we are instead presenting the new scatter plot using just raw data.

14: Among the seven patients included in the initial splicing analysis (Fig. 1a), 2 out of 3 who were homozygous (G/G) and the one patient that was heterozygous (C/G) for the risk allele at rs12973192 showed inclusion of the cryptic exon in almost every UNC13A mRNA that was mapped to intron 20-21. In contrast, the other patients who were homozygous for the reference allele (C/C) showed much less inclusion of the cryptic exon (Extended Data Fig. 7 a and b).

- This amounts to a form of eQTL analysis but with a very small number of patients and without any detail on the control of common co-variables which would be expected for standard eQTL analyses and so it is hard to assess its value. I would advise the authors to either add the relevant detail and make explicit the caveats.

We agree that the lack of correction for the covariates is a concern. However, this is just an interesting observation that we share. And despite the dataset not being big, it is a direct demonstration that the SNP may have an effect on splicing without any experimental manipulation. Our knowledge of detailed information of the patients is from the original paper (Liu et al., 2019)] and has been clarified in the manuscript (Extended Data Fig. 9b). There are not any obvious confounders based on our limited knowledge of the patients.

16. Two of the iPSC-MN lines that we used to detect cryptic exon inclusion upon TDP-43 knockdown (Fig. 1h, iPSC-MN1 and iPSC-MN3) are heterozygous (C/G) at rs12973192. We sequenced the RT-PCR product that spans the cryptic exon and analyzed the allele distribution from these two samples as well as the one patient sample from the original RNAseq dataset (Fig. 1a) that is heterozygous (C/G) at rs12973192 (Extended Data Fig. 7b). We found a significant difference between the percentage of C and G alleles in the spliced variant, with higher inclusion of the risk allele (Fig. 4b and Extended Data Fig. 7c).

- This is an ASE analysis performed in the manner in which ASE was originally used. Of course a lot hinges on the design of the primers as to the validity of this approach and I think the authors should make some comment on how this was approached to prevent biasing of the results.

Thank you for the comments. We have shown the primer design in Extended Data Fig. 5. Since the primers are located in the exons flanking the intron that contains the cryptic exon, it would include all the different kinds of the splicing events (Extended Data Fig. 3), It has been shown in other studies that the mapped allelic reads may be biased to the allele represented by the reference genome (Degner et al., 2009). However, in our case, the reference allele at rs12973192 is C allele but we have a higher percentage of G allele. Therefore, the bias to the reference allele does not exist in this specific case.

Comment 17: The rest of the patients (n=85) have exactly the same number of risk alleles at both loci, indicating that it's very likely the patients are carriers of the reference haplotype or the risk haplotype. Using a linear regression model, we found a strong correlation between the number of risk haplotypes and the abundance of UNC13A cryptic exon inclusion (Fig. 4c and figure using raw data, Extended Data Fig. 7d). Taken together, these data suggest that genetic variation in UNC13A that increases risk for ALS and FTD in humans promote cryptic exon inclusion upon TDP-43 nuclear depletion.

- This is an eQTL analysis conducted in patient samples (has been done for AD, MDD etc) in which case it is subject to all the issues which are normally a concern ie. controlling for known and unknown co-variates, genetic covariates and reaching genome-wide significant p-values. This makes the p-value attained is not particularly impressive and the necessary details to assess this are not present in the manuscript as things stand, I would have to hunt through the code. I appreciate that the N isn't high so that might explain the p-val but then I think the authors need to recognise the limitations of this analysis.

We agree with the comments and have updated the linear model to include all the known covariates. The results have been updated in the text and the table showing the results from the multiple regression is shown in Extended Data Table 4. However, because we were only looking at the association between one genotype and the RT-qPCR signals of the cryptic exon, the data does not have the complexity for us to account for hidden covariates using methods such as PCA or PEER (Stegle et al., 2010). Additionally, since we were only testing whether the number of this one specific risk haplotype is correlated with the intensity of RT-qPCR signals, rather than scanning the loci on the entire chromosome or in the genome, the p-value here can be used as a statistical confidence measure.

18. (h). RPLP0 was used to normalize RTqPCR. (Linear mixed model, ****P<0.0001; mean \pm s.e.m.). (f and i) Immunoblotting for UNC13A and TDP-43 protein levels in SH-SY5Y cells (f) and iPSC-MNs (i) treated with scramble shRNA (shScramble) or TDP-43 shRNA (n=3). GAPDH served as a loading control. (g and j) Quantification of the blots in (f and i), respectively (two-sided Welch Two Sample t-test, *P<0.05, **P<0.01).

- What's the basis for the choice of RPLP0 for normalisation and why weren't multiple genes used for normalisation which is the usual way this type of analysis is done and in fact was done by the authors in another section of the paper where GAPDH was also used.

For the specific experiments in SH-SY5Y cells and iPSC-MNs, GAPDH returned much higher Ct values (Ct = 35 ~ 39) relative to our test primers for STMN2 (Ct = 19 ~ 25), indicating that using GAPDH as a control in dilute samples could lead to misleading or non-linear Ct values. And the Ct values of RPLP0 appeared to be unaffected by differences in samples and experimental treatments.

19: When running the BaseScope analysis are there any attempts to control for common covariates likely to impact on detection eg. sex, age, RIN of the tissue etc. This may have occurred but it is not clear to me from the methods.

Age and sex are reported in Extended Data Table 1. Age across the cases is well matched. Sex matching was not feasible based on case availability; we prioritized matching for age. Adjustment for sex was not performed because samples are too small, but we are not emphasizing statistical differences here. Although we lack RINs for these particular (fixed) tissue blocks, the canonical *UNC13A* BaseScope data address potential variation in RNA integrity.

20. Methods: RNA-Seq alignment and splicing analysis Detailed pipeline v2.0.1 for RNA-Seq alignment and splicing analysis is available on [https://github.com/emc2cube/5 Bioinformatics/sh_RNAseq.sh](https://github.com/emc2cube/5_Bioinformatics/sh_RNAseq.sh).

- Given the key importance of the RNAseq data analysis to this paper, there is simply insufficient detail in the methods on the data sets used to run analysis and the analysis itself. I appreciate that the github link is present but that doesn't negate the need for more detail within the main manuscript.

Thank you for the suggestion. We have included more details in the main manuscript (and in Supplementary Notes 1-3 and have uploaded all the scripts used for the analysis to <https://github.com/rosaxma/TDP-43-UNC13A-2021>.

21. Methods: MAJIQ – “Here, de novo refers to junctions that were not in the UCSC transcriptome annotation, but had sufficient evidence in the RNA-Seq data (--min-intronic-cov 1).

- I assume tis means that a single intronic read was sufficient but it would be useful to have this more explicitly explained

We have provided the rationale for this choice in the Supplementary note 1. We included the paragraph from the supplementary note here:

“ We increased the value assigned to **--min-intronic-cov** at the build step of MAJIQ from the default 0.01 to 1. This parameter sets the minimum number of average reads per position in one of the bins in the introns to be considered to have sufficient coverage at that position. MAJIQ sets the size of the bin as the readlength of the RNA-Seq so that the sizes of bins are equivalent to the sizes of the junctions (junctions are defined by reads that span the exon boundaries). Since the RNA-Seq libraries from Liu et al. were prepared from the nucleus, it is likely that they may include more reads mapped to the introns than what would normally be expected. However, it is an arbitrary decision. None of the genes presented in fig 1C have intron retention because such variations could not be detected by LeafCutter.”

22. Methods: LeafCutter (commit 249fc26 on <https://github.com/davidaknowles/leafcutter>): Using the already aligned RNA-Seq reads as previously described, reads that span exon-exon junction and map with a minimum of 6 bp into each exon were extracted from the alignment (bam) files using filter_cs.py with the default settings. Intron clustering was performed using the default settings in leafcutter_cluster.py. Differential excision of the introns between the two conditions (TDP-43-positive neuronal nuclei vs. TDP-43-negative neuronal nuclei) were calculated using leafcutter_ds.R

- Are the settings for exon-exon junction reads used by LeafCutter and MAJIQ filtered the same way eg. overhangs into the exon, limits on the size of the implied intron etc.
- From this information and that in the main manuscript it appears to me that you are filtering MAJIQ output for the delta psi but not the LeafCutter output, why and what is the justification for this approach?
- How are you matching the output from MAJIQ and LeafCutter? The clustering will be different and direction of effects could be different so this is not trivial.

Thank you for the comments. We have now provided detailed comparisons of MAJIQ and LeafCutter in the Supplementary Note 1.

1. At the time of running the software (commit 249fc26 on <https://github.com/davidaknowles/leafcutter>), a script (filter_cs.py) provided by LeafCutter is used in “Step1 Converting bam to junct”. The script filters for reads that span introns longer than 50 bp and have at least 6 nt overhangs that are mapped into each spanning exon. MAJIQ does not explicitly filter for reads based on the length of the intron they span and the length of the overhangs. Moreover, during the alignment stage using STAR, such a problem has been dealt with by requiring reads that span unannotated junctions to have a minimum overhang of 8nt (**--alignSJoverhangMin 8**; this is consistent with the ENCODE standard options).

2. Since MAJIQ and LeafCutter use different units for transcript variations (LSVs vs. intron clusters), we also do not think the delta PSIs from the two tools are directly comparable. While MAJIQ focuses on the PSIs at a LSV, the PSIs in LeafCutter are normalized to the entire intron cluster. We also cannot compare the p-values from the two tools because MAJIQ does not produce a p-value but a Bayesian posterior for $P(\Delta\text{PSI} > C)$, which essentially means the Bayesian posterior probability of ΔPSI larger than a constant C . While ΔPSI gives us the information about the magnitude of changes, p-value, or in the case of MAJIQ, $P(\Delta\text{PSI} > C)$, shows the tool's belief in the splicing changes. In our case, we decide that the confidence in changes is more interesting than the magnitude of changes, since small changes in transcriptome could be sufficient to lead to functional changes. Specifically, when filtering for -LeafCutter outputs with an adjusted p-value smaller than 0.05, we get 139 unique genes. Therefore, to get an equivalent number of genes from the output of MAJIQ, we set the C as 0.1, which is

more permissive than the default setting 0.2. The filtered MAJIQ outputs contain 198 unique genes.

3. The outputs from the two tools are event matched and the coordinates of the corresponding regions are provided in the Supplementary Table. There are 66 overlapping genes, among which CASP contains 3 different regions, SETD5 contains 2 regions and SYNE1 contains 2 regions. Only 3 genes (APLP2, EPB41L1 and KIF3A) from the outputs of leafcutter did not reach a delta PSI larger than 10%. However, this can likely be attributed to the normalization in intron clusters.

23. Quantification of UNC13A splice : UNC13A contains more than 40 exons and RNA-Seq coverages of mRNA transcripts are often not uniformly distributed 43, we looked at reads spanning “Exon 19-Exon 20” junction, which is included in both the canonical isoform and the splice variant, and there is a strong correlation (Pearson’s $r = 0.99$) between the numbers of reads mapped to “Exon 19- Exon 20” junction and “Exon 20-Exon 21” junction. We observed that samples that have at least 2 reads spanning either “Exon 20-CE” junction or “CE-Exon 21” junction have at least either UNC13A TPM = 1.55 or 20 reads spanning “Exon 19- Exon 20” junction. Therefore, we selected the 1151 samples that had a TPM ≥ 1.55 , or at least 20 reads.

- This comes across as a slightly dirty analysis, why use two different ways of filtering read depth which are correlated but not equivalent? In particular, I don’t see the value of the TPM cut off. More generally, I think this analysis comes across as overly complicated and there is insufficient detail.

Thank you for your suggestions. We set the TPM cutoff because the coding sequence of the most common UNC13A isoform (GTEx) in the central nervous system, ENST00000519716.7, is 5109 bp, which is 1.5 times as long as the length of an average coding sequence (Piovesan et al, 2016), 3399 bp. The RNA-Seq coverage of mRNA transcripts are also often not uniformly distributed (Hansen et al., 2010). We observed that samples that have at least 2 reads spanning either “Exon 20-CE” junction or “CE-Exon 21” junction have at least UNC13A TPM = 1.55. This indicates that if we want to detect reads that span the cryptic exon junctions, there should be at least 1.55 UNC13A transcripts per million of sequenced full-length transcripts. A TPM lower than that 1.55 UNC13A may indicate that the tissue samples may contain too few neuronal cells for us to detect the splice variants, which would only be in neurons that are affected by TDP-43 pathology. We have provided detailed explanation for this analysis in Supplementary note 3.

Referee #2

In this study, Ma, Prudencio, Koike and colleagues report the inclusion of a novel cryptic exon (CE) in the gene UNC13A, an RNA target of TDP-43, in a subset of FTD and ALS postmortem patient samples, which results in a dramatic decrease of UNC13A mRNA and protein levels. Using a bioinformatic approach and experimental validation in cells, they show that CE inclusion is dependent on TDP-43 loss of function and is increased by two UNC13A small nucleotide polymorphisms (SNPs), previously shown to be associated with increased ALS and ALS-FTD risk. They also identify a novel UNC13A haplotype within the same intronic site that they propose is a potential risk for developing these diseases. Collectively, the work shows that the UNC13A CE inclusion is a marker of TDP-43 dysfunction in TDP-43 proteinopathies and indicates a direct link between TDP-43 loss of function and disease.

Following the recent discovery of the abnormal inclusion of STMN2 CE upon TDP-43 loss of function and the resulting loss of STMN2 mRNA and protein with detrimental effects for neurons (Melamed et al, 2019; Klim et al, 2019), the current work supports a mechanistic link between TDP-43 pathology and defects in key neuronal proteins. The findings are novel and exciting as they set the stage for several applications, including potentially UNC13A-based diagnostic or patient stratification tests, as well as therapeutic approaches for these diseases.

The manuscript is clearly written and the experiments are carefully designed and well-controlled. Due to the strong relevance of these findings for human disease and the high quality of the study, I enthusiastically recommend this work for publication in Nature, after revision to address the following concerns:

We thank the Referee for his/her enthusiasm for our findings, putting the work in the exciting context that has been recently emerging in the field, and for appreciating the significance of this work. We are grateful for the very helpful comments and suggestions.

1. My main concern is the lack of functional validation of the effect of UNC13A decrease in adult human neurons. While I appreciate that this is technically challenging, I think it is important to show that the direct effect of TDP-43 loss of function on UNC13A levels has significant consequences for neuronal maintenance and function.

We appreciate this comment and have considered it carefully. Referee 2 is asking us to perform experiments in adult human neurons to test if the effect of TDP-43 loss of function is due to decreased UNC13A levels. In other words, can we rescue TDP-43 loss of function phenotypes by either restoring UNC13A levels or by blocking cryptic splicing in the first place? This experiment is difficult to perform and even if it works, the results will be difficult to interpret.

There are several reasons for this. First, full function of *UNC13A* requires formation of synapses, which, in iPSC-derived neurons, requires full maturation (6-8wks). Second, as we demonstrate, TDP-43 regulates at least 65 cryptic splicing targets in the brain, including additional synaptic proteins (such as CAMKIIB). Knocking down TDP-43 leads to dysregulation of these, and it is unlikely that restoring *UNC13A* alone will be able to rescue all TDP-43-loss based neuronal defects. Importantly, knocking out *UNC13A* does not affect neuronal survival per se (CRISPR screen data from Kampann lab, Tian et al., 2021). Therefore, we cannot simply look at a neuron survival phenotype. Rather, experiments would require investigating synaptic function by electrophysiology on aged cultured iPSC-derived neurons – where lack of *UNC13A* is already known to cause deficits (loss of evoked and spontaneous release) (Fenske et al., 2019). Therefore, we suggest that these experiments will not be informative (because loss of TDP-43 function causes dysregulation of many other genes) and do have a timeline of at least 1 year.

Furthermore, the human genetics data in Fig. 4, showing the dose-dependent decrease in survival in individuals carrying the *UNC13A* risk alleles, along with our studies demonstrating that these risk alleles cause increased cryptic splicing of *UNC13A*, indicate that *UNC13A* is a key target of TDP-43. Finally, although we do not yet understand how lack of this synaptic protein (or it is formally possible that defects could be caused by production of truncated RNAs or aberrant peptides produced from the cryptic exon) contributes to degeneration, we do know that even mild *UNC13A* changes impact ALS risk in patients, from strong human genetics data. This is true so far only for *UNC13A*, hence very different from other genes with cryptic exons.

Referee 2 seems to be concerned that we have not proven that *UNC13A* is a therapeutic target. In other words, if we can block *UNC13A* cryptic splicing will that have any effect on patient survival? We don't have the answer to that question, although the human genetics described above (cryptic exon promoting risk alleles decrease survival) strongly argue that it would be beneficial to reduce cryptic splicing. In the revised manuscript we have added a discussion about future work that is needed in order to pursue *UNC13A* as a therapeutic target. We are very excited that cryptic exon targets of TDP-43 (like *STMN2* and now *UNC13A*) are emerging and it will be exciting for the field in the future to evaluate other ones and combinations of ones (e.g, *STMN2* + *UNC13A*).

2. The effect of the *UNC13A* SNPs on TDP-43 binding is not explored in this study, as the authors state that in most cases SNPs are just markers and not the actual genetic defect. Yet, the accompanying study by Brown et al showed that one of the previously identified SNPs (rs12973192) leads to a significantly lower binding affinity for TDP-43 than the wild type sequence (Fig. 4F-I, Brown et al), while, in contrast, there is enhanced TDP-43 binding on rs12608932 SNP (Fig. 4I, Brown, et al). In light of this somewhat puzzling result, I think it is very important that Ma and colleagues experimentally test the effect of the current SNPs and their novel haplotype rs56041637, on TDP-43 binding on this region.

To address this suggestion, we performed quantitative electrophoretic mobility shift assays (EMSA). We used purified recombinant TDP-43 and performed quantitative electrophoretic mobility shift assays (EMSA) with six different *UNC13A* probes (cryptic exon reference SNP vs risk SNP; intron reference SNP vs risk SNP; intron reference repeat vs risk repeat). The probes we used were slightly different than the ones used by Brown et al (ours are longer). These new data are in Fig. 4g and Extended Data Fig. 13. The quantified apparent binding affinity (KD,app) reveals that while TDP-43 binds the reference and risk versions of the cryptic exon with similar affinity, the intronic risk allele results in a minor reduction in affinity (Fig. 4g). TDP-43 had lower affinity for a probe containing the longer repeat version compared to the reference short repeat sequence (Fig. 4g). We note that the certainty of KD app value is limited by the number of replicates and the aggregate prone nature of TDP-43. Overall, we find in these *in vitro* binding assays that TDP-43 has a much higher affinity for the intron sequence compared to the exon or repeat sequence. The diminished binding affinity of TDP-43 to risk alleles of the intron and repeat sequence may contribute to the increased cryptic exon splicing found in ALS and FTD. Future studies will be required to explore how TDP-43 regulates the cryptic splicing of *UNC13A* and other splicing targets and the impact of different genetic variations on TDP-43 binding *in vivo*. These results are somewhat different than those reported in the accompanying manuscript but the experiments are slightly different (we use full-length TDP-43, which on one hand can be aggregation prone (though our MBP-tagged TDP-43 is soluble) and the other team does not (they use a construct only containing TDP-43's RRM). In any case, we do not want to over interpret results from *in vitro* binding studies, especially with SNPs that are common variants, which would necessarily be expected to have small effects, which could still be meaningful in terms of lifetime risk for FTD/ALS. We have added these new data to the revised manuscript and are cautious about interpretation.

3. As stated in the manuscript, rs12973192 and rs12608932 SNPs are mainly associated with increased ALS risk (van Es et al., 2009; Nicolas et al., 2018), but also shown to contribute to FTD in sporadic ALS (Placek et al., 2019). However, CE inclusion seems to be more frequent in FTLD-TDP (Figure 2) than in ALS/FTLD or ALS-TDP (Figure S5). How do the authors explain this result?

Thank you for pointing this out and we agree that it's indeed puzzling. We'd like to point out that in the human brain, at the single-cell level, *UNC13A* has the highest expression in neurons (FPKM = 3.3) and has much lower expression in mature astrocytes (FPKM = 0.5), oligodendrocytes (FPKM = 0.2), and microglia/macrophages (FPKM = 0.1) (Zhang et al., 2016). At the tissue level, *UNC13A* is more highly expressed in the frontal cortex (TPM = 530.2) than in the spinal cord (TPM = 35.54)(GTEx). It is also estimated that the non-neuron to neuron ratio (nNNR) of the human spinal cord, the cerebral cortex (gray and white matter combined), and the cerebellum are 6.5, 4.32 and 0.29, respectively (Bahney et al., 2017), suggesting that there are lower percentages of neurons in the spinal cord, compared to in the cerebral cortex. Therefore, it is possible that motor neurons may have lower *UNC13A* expression levels than the frontal cortex. Unfortunately, there is a lack of studies that characterize the transcriptomic differences across different neuronal subtypes in humans to verify this hypothesis.

In summary, while it is possible that motor neurons are more resistant to CE inclusion upon TDP-43 nuclear clearance, the lack of detection could also be explained by the lower expression of *UNC13A* in spinal cord compared the frontal and temporal cortices.

4. The authors hint at the possibility that *UNC13A*, *STMN2* and potentially other cryptic exon splicing events may “contribute to heterogeneity in clinical manifestation of TDP-43 dysfunction”, which is indeed an attractive hypothesis. In this context, I am wondering whether the apparent more frequent detection of *UNC13A* CE inclusion in cortical samples (compared to spinal cord samples) not only from FTL-D-TDP (Figure 2), where this is expected but also from ALS/FTLD and ALS-TDP patients (Figure S5) may suggest that *UNC13A* has a more important role in the function of cortical neurons compared to motor neurons. Providing any evidence for or against this hypothesis will significantly strengthen the study, in my opinion.

Thank you for proposing this very interesting hypothesis. While it is unclear from the literature whether *UNC13A* has a more important role in the function of cortical neurons compared to motor neurons, the available studies do indicate that *UNC13A* plays an important role at neuromuscular junctions. For instance, Engel et al. (2016) reported that a complete loss of *UNC13A* leads to a severe brain disease and premature death. Specifically, a truncating homozygous *UNC13A* mutation inhibits cholinergic transmission at the neuromuscular junctions. Additionally, Reddy-Alla et al. (2017) also showed that *Unc13A* was stably and precisely positioned at active zones of *Drosophila* glutamatergic neuromuscular junctions, which are chemical synapses between motor neurons and muscle fibers, and regulates synaptic vesicle activity, suggesting that *UNC13A* could play an important role in motor neurons. If mouse conditional knockouts of *Unc13a* (*Munc13-1*) become available, it will be interesting to test this hypothesis directly using neuronal subtype specific Cre driver lines. One picture that might emerge is that the cryptic target *STMN2* could play a key role in spinal cord whereas *UNC13A* could play a key role in cortical neurons; and perhaps some combination of effects could contribute to ALS/FTD. The human genetics (GWAS connection of *UNC13A* SNPs to ALS), however, does indicate that *UNC13A* is a risk factor for pure ALS, so future work will certainly be required to directly test *Unc13a* function in adult spinal cord (using conditional inactivation in mouse).

5. Linked to the previous point, FISH experiments show the presence of CE inclusion and reduction of *UNC13A* mRNA in cortical samples (Figure 3 and Figure S6). Are these also present in affected motor neurons of ALS patients? Or is it possible that motor neurons are more resistant to *UNC13A* CE inclusion, upon TDP-43 aggregation and nuclear clearance? If that were the case, it would explain the more frequent detection of *UNC13A* CE in FTL-D cases.

To address this suggestion, we have attempted Basescope® experiments on human spinal cord but *UNC13A* is expressed at much lower levels in spinal cord than in the brain. We note that this is in contrast to *STMN2*, which we robustly detect in spinal cord, consistent with

recently published studies. However, the mini-gene reporter experiment presented in Figure 4e also suggests that the *UNC13A* CE inclusion is not very sensitive to the cellular context as long as there is a depletion of TDP-43 expression. The experiment is performed in HEK293T cells, which normally do not express *UNC13A*. The Referee's suggestion that differences in expression of *UNC13A* between spinal cord and brain could explain more frequent detection of the cryptic exon in brain compared to spinal cord is interesting and we have added a discussion to the revised manuscript mentioning this point. We are very excited about exploring the relative contributions of *STMN2* and *UNC13A* cryptic exon inclusion (and perhaps combinations) and how they play out in different regions of the CNS in disease.

6. I find the detection of *UNC13A* CE foci very convincing and I applaud the authors for setting up this challenging assay. Given that the CE-containing transcript is expected to be unstable, I find it curious that their detection was possible at all and that it was confined to the nucleus. I am wondering if their presence and localization may be linked to errors in nucleocytoplasmic transport in human tissues that are now well documented in these diseases. One way to answer this question would be to test if the *UNC13A* CE foci are also present in cultured cells after knockdown of TDP-43.

This is an interesting point and we are not sure why the *UNC13A* CE containing transcripts are retained in the nucleus. The Referee's hypothesis about errors in nucleocytoplasmic transport is interesting but we note that when we perform Basescope analysis on the canonical transcript (e20/e21) we detect that transcript in the cytoplasm, even in neurons with TDP-43 pathology, suggesting that not all versions of *UNC13A* transcripts are prevented from nuclear export. We sent the authors of the accompanying manuscript (Brown et al) our Basescope® probes and they performed this suggested experiment and detected the cryptic exons in the nucleus of cultured iNs. They have included these data in their revised manuscript.

7. For the initial identification of *UNC13A* CE the authors using two different pipelines for splicing analyses (MAJIQ and LeafCutter) and found a rather small fraction (<50%) of overlapping alternatively spliced genes. What is the reason for this discrepancy? Are there specific advantages, or drawbacks to either of these pipelines? Despite the confirmed *STMN2*, *UNC13A* and *POLDIP3* the question is important in light of additional events that are potentially relevant, but maybe were only identified by one of the two pipelines.

Thank you for raising this question. To address this, in the revised manuscript we have discussed in detail the potential reasons for the discrepancy in outputs, advantages and drawbacks to either of these pipelines in Supplementary Note 1.

In summary, while both are capable of detecting *de novo* splicing junctions, MAJIQ and LeafCutter use different modeling strategies. MAJIQ identifies and compares LSVs (local splicing variations); LeafCutter, on the other hand, detects intron inclusions and collapses all overlapping intron to an "intron cluster", for which the inclusion levels are normalized and quantified. Additionally, only MAJIQ detects alternative first and last exons, or intron

retentions. Because of the different modeling strategies, the two tools also have different sets of parameters that users can adjust. Therefore, it is very possible with careful tweaking of the parameters used for the two algorithms (including the ones in the source codes), we could get more overlaps in the outputs. However, we did not attempt that because we only intended to use the two splicing analysis tools to help us narrow down the genes and intronic regions to focus on.

For the above mentioned reasons, we definitely think the events that were only identified by one of the two pipelines could provide valuable insights. However, it is challenging to present the outputs from the two pipelines in a spreadsheet due to the complexity of splicing events, and such a format may not be very informative. Therefore, we encourage readers to explore the outputs using the interactive interface provided by each of the pipelines. We provide instructions for using the interactive interfaces at https://github.com/rosaxma/TDP-43-UNC13A-2021/tree/main/RNA_alignment_and_splicing_analysis.

While we only displayed the overlapping genes in Fig. 1c, the outputs from the two tools are event matched and the coordinates of the corresponding regions are provided in the Supplementary table 1. There are 66 overlapping genes, among which CASP contains 3 different regions, SETD5 contains 2 regions and SYNE1 contains 2 regions. Such information has also been updated in the main text.

8. The authors use bulk RNA-seq data to find UNC13A CE inclusion and to determine the specificity of this event to FTL and ALS cases with TDP-43 pathology (Figure 2 and S5). To validate their pipeline, which is based on an arbitrary cutoff, it would be important to show that with this same method they can identify other alternatively spliced events, known to be linked to TDP-43 loss of function in the same datasets, such as STMN2 CE inclusion, as they did in Figure 1b.

Thank you for this good suggestion. In the revised manuscript, we show that with the same method, we can also identify the STMN2 CE inclusion (Extended Data Table 2).

9. It is not clear to me which SNP(s) are included in the two reporter minigenes that the authors developed. Is it all three at the same time? Different minigenes with either rs12973192, rs12608932, or rs56041637 alone would give information on the role of each SNP in increasing CE inclusion. These would also be useful in the context of point #2 above.

Thank you for this great suggestion. We did this as suggested and we also tested all permutations (single SNPs, doubles, and all 3). In total, we tested 8 constructs in the revised manuscript. We show these new data in Extended Data Fig. 10, which indicate that the risk variant at rs12973192 is the strongest contributor to cryptic exon inclusion but the other ones do contribute and are not simply additive (Extended Data Fig. 10). Because of linkage disequilibrium and the general principle of linked variants, we were and still are hesitant to pin all of the effects on one SNP and these new results support that notion. Our findings indicate that the *UNC13A* risk haplotype that is associated with increased risk for ALS and FTD/ALS causes increased cryptic splicing in the face of TDP-43 loss of function. The lead

GWAS SNP that is located right in the cryptic exon itself (rs12973192) can impact cryptic splicing by itself but there are likely additional variations contributing to cryptic splicing. Future efforts by us and others to fine map the locus will be informative.

10. The authors state that "our functional studies indicate that TDP-43 dysfunction is required for UNC13A cryptic exon inclusion [...] the UNC13A risk alleles exert a TDP-43 loss-of-function-dependent disease-modifying effect". Said functional studies consisted of downregulating TDP-43 levels or transiently overexpressing an RNA-binding TDP-43 mutant. Perhaps more relevant would be to test the levels of UNC13A CE inclusion in a model of TDP-43 misregulation at endogenous TDP-43 levels. For example, it was recently shown that inhibition of deacetylases leads to increased acetylation of the RRM of TDP-43 disrupting the RNA binding properties of the protein (Ye et al, Science 2020).

We tested the hypothesis by inducing histone acetylation using histone deacetylase inhibitor SAHA. We showed that SAHA significantly enhanced histone acetylation. However, we did not see an increase in the level of cryptic exon inclusion (Ct > 37).

(a) SH-SY5Y cells were incubated in the presence or absence of SAHA (20 μ M, 24 hours), with proteasome inhibitor bortezomib (BTZ; 2.5 μ M) added for the last 4 hours of treatment as suggested by Yu et al. Cells were subsequently stained with anti-acetyl-lysine (Abcam ab22550, 1:250) and Alexa Fluor-conjugated secondary antibodies. Subpanel iii shows there is a significant enhancement in acetylation (b) RT-qPCR of the RNA extracted post drug treatment shows that that treating the cell with histone deacetylase inhibitor does not induce *UNC13A* cryptic exon inclusion, nor does it lead to a decrease in total *UNC13A* level (One-way ANOVA, ***P < 0.001).

We think that future studies aimed at other ways to impair TDP-43 function and assess impact on cryptic exon inclusion will be informative.

11. I recommend the authors to show the precise location of the previously reported TDP-43 binding site on the UNC13A intron in Figure 1d. This will be very helpful for the readers. I note that the binding motifs that are currently highlighted are not necessarily identical with the binding site.

The binding sites have been highlighted in Figure 1d.

12. Page 3, line 11: reference to Figure 1f should also contain Figure 1g (its quantification), which is otherwise not referenced in the text.

Thank you for pointing this out. We have modified the text to incorporate the suggestion.

13. Extended Data Fig. 2a: the conservation of UNC13A among mammals is not clear from the image. Do black boxes represent conserved residues? Overall this figure is very difficult to read.

We have updated the figure to incorporate the suggestions and to make the figure easier to read.

14. I encourage the authors to make all raw data, code, and materials used in the study publicly available upon publication and not only upon request. I also encourage them to add more details on their analyses whenever possible for full transparency. For example, I could not find the filter_cs.py file. Is this available? Is it part of LeafCutter?

Thank you for your suggestion. The processing codes are available at <https://github.com/rosaxma/TDP-43-UNC13A-2021>. Filter_cs.py files is indeed a script provided by LeafCutter (commit 249fc26 on <https://github.com/davidaknowles/leafcutter>).

Referee #3

Amyotrophic lateral sclerosis (ALS) and frontotemporal dementia (FTD) are two neurodegenerative disorders with a common etiology; the RNA binding protein TDP-43 undergoes a shift from a nuclear- to a cytosolic localization, thereby losing its ability to function as a splicing repressor. The downstream effect involves mis-splicing events, for example the unwanted inclusion of so-called cryptic exons (CEs) in mature RNA. These CEs would normally be spliced out, and their inclusion results in downstream errors in protein homeostasis.

Munc13-1 is a central presynaptic protein. It is involved in a preparation step synaptic vesicles must undergo to make them release competent. Munc13-1 is the central isoform in the majority of synapses at the central nervous system (CNS), as well as in the neuromuscular junction. Its knock-out in mice is a cause for one of the most severe phenotypes in synaptic biology; in excitatory hippocampal neurons, it results in a 90% block of neurotransmission, and the concomitant removal of Munc13-2 leads to a complete block of neurotransmitter release, in both excitatory and inhibitory CNS neurons (Augustin et al., 1999; Varoqueaux et al., 2002) . Recently, Munc13-1 has been shown to be arranged in clusters that are thought to represent discrete SV release sites (Reddy-Alla et al., 2017; Sakamoto et al., 2018). The number of clusters is associated with the strength of the synapse; synapses with many clusters will release more glutamate, making them stronger, whereas synapses with less clusters will release less glutamate, making them weaker. Work in the *Drosophila* neuromuscular junction also showed that the amount of the Munc13-1 homolog UNC13A is scalable under plasticity (Bohme et al., 2019). Current interpretations of these data suggests that the levels of Munc13-1 in synapses may regulate the strength of neurotransmitter release. In addition to synaptic strength, Munc13-1 is a regulator of synaptic plasticity, translating elevations in Ca²⁺ and dynamic changes in membrane lipid concentrations into changes in the efficacy of neurotransmission during activity (Lipstein et al., 2013; Shin et al., 2010). Finally, two cases of human brain disease have been described, with the loss of Munc13-1 leading to a severe brain disease and premature death (Engel et al., 2016), and one variation described to result in a neurodevelopmental and a neuropsychiatric disorder (Lipstein et al., 2017).

A genetic association between non-coding SNPs in the UNC13A gene locus and ALS has emerged over a decade ago. Interestingly, the SNPs have been associated with increased disease risk and with increased disease progression, which makes the mechanism by which these contribute to disease significant and important to elucidate. To date, however, the molecular and cellular mechanisms linking these SNPs to the disease, and, more importantly, whether these indeed involve the Munc13 gene product, remained elusive.

In this exciting manuscript, Ma and colleagues establish these mechanisms and links. They demonstrate that the nuclear loss of TDP-43 leads to a CE inclusions in the UNC13A RNA, resulting in substantial reduction in protein expression. The supporting data includes:

- A splicing analysis of a previously published RNA-Seq dataset identified 65 mis-spliced transcripts, amongst which a highly significant hit is the UNC13A RNA. The major mis-

splicing event is the inclusion of an CE between exon 20 and 21 (Figure 1).

- TDP-43 directly binds to the intronic sequence that includes the CE (Figure 1).
- In vitro depletion of TDP-43 in three cell lines leads to inclusion of the CE, reduction of mRNA levels and protein abundance (Figure 1).
- RNA analysis in patient-derived material identifies the CE inclusion as a common event specific to TDP-43-mediated disease (Figure 2).
- The levels of phosphorylated TDP-43, which are correlated with its abundance in the cytoplasm, are positively correlated with the levels of CE inclusion (Figure 2).
- In situ hybridization identifies nuclear RNA transcripts of UNC13A including the CE in patient-derived material (Figure 3). Canonical UNC13A mRNA is reduced in TDP-43 affected cells from patient tissue.
- The presence of risk SNPs, that are located in close proximity to the CE, is strongly associated with CE inclusion in mRNA, also in patient material (Figure 4).
- Identification of yet another risk region (rs56041637), associated with the two SNPs.
- Minigene assay demonstrating TDP-43-dependent and risk SNPs extravagated mis-splicing of the CE.

The paper is clear, precise, very well-written, and exciting. The experimental array used and the conclusions drawn appear justified and robust to me, although I am not an expert in the full spectrum of methodologies used, and others may be better suited to determine whether all controls are included.

The major novelty is the univocal demonstration of a functional relationship between TDP-43 and Munc13-1. This exciting finding is highly significant, as it is associated with SNPs that exert a substantial effect on disease course. Because Munc13s are primarily synaptic proteins, the data presented here draws a strong connection between presynaptic dysfunction and the neurodegenerative disorders ALS and FTD, expanding the already identified link between presynaptic function and neurodegeneration in tauopathies and alpha-synucleinopathies. In fact, some data the authors decided not to follow up at this point (Fig 1b, see comment below) makes this link very strong. Notably, the clear demonstration that the RNA irregularities in the UNC13A transcript can be found in patient-induced material is extremely convincing of the relevance of the finding. This paper opens a new avenue for mechanistic studies of ALS/FTD in synaptic biology and its findings are likely to be followed up by many.

A major element that the current paper does not address is whether and how reducing the expression levels of the Munc13 proteins may lead to cellular pathology and/or to disease course modulation. It is important to note, that while a relationship between the reduction of Munc13-1/2 expression has been functionally established in hippocampal and striatal neurons (no Munc13s = no synaptic transmission (Varoqueaux et al., 2002)), and in the neuromuscular junction (no Munc13s = changes in synaptic transmission (Varoqueaux et al., 2005)) no such relationship has been established with regards to cell survival. In fact, while mice lacking the two isoforms die at birth due to a block of synaptic transmission, their CNS appears fully intact (Varoqueaux et al., 2002). In CNS-derived cultures or in organotypic slices in vitro, no degenerative effects have been observed (Broeke et al., 2010; Imig et al., 2014). In a single human patient with a Munc13-1 protein loss, neuromuscular architecture was fully preserved (Engel et al., 2016). Moreover, in mice, heterozygosity results in 50% reduction in protein abundance, but not in observable changes in synaptic function or in

animal behavior, although this has not been studied in great detail (Augustin et al., 1999; Varoqueaux et al., 2005). Because conditional mouse models to reduce the expression of Munc13 isoforms in the adult nervous system (to levels lower than 50%) have not been published yet, the effects of such reduction at the adult nervous system have not been studied and are difficult to address. As such, this goes beyond the scope of the current manuscript. I would therefore recommend the paper for publication, provided that the following major points will be addressed:

We thank the Referee for his/her remarkably detailed and informative description of Unc13a biology and for appreciating the importance of our findings functionally connecting UNC13A and TDP-43. We share the Referee's excitement for these findings and positive outlook that these data (along with the accompanying paper by Brown and colleagues) will open up new areas for the ALS/FTD field focused on synaptic biology and presynaptic proteins. We also are grateful for the Referee for cogently describing the phenotypes of Unc13a deficiency in rodents and humans, which has been immensely helpful to us in thinking about our findings.

Referee 3 correctly points out that the Unc13a (Munc13-1) knockout mice do not exhibit neurodegeneration. However, these mice, generated by Dr. Nils Brose, die a few hours after birth owing to severe deficits in synaptic transmission (Augustin et al., 1999). KO mouse phenotypes are consistent with homozygous nonsense mutations in UNC13A in humans, which cause severe neurodevelopmental phenotypes (Engel et al, 2016). Thus, it is not possible to study age-dependent neurodegenerative phenotypes in the homozygous knockout mice. The heterozygous mice do not exhibit a phenotype. This means that 50% reduction in UNC13A is not sufficient to elicit defects. But heterozygous mice do not mimic either the level or the timing of UNC13A reduction we observe in FTD/ALS cases – heterozygous KO mice have approximately 50% reduction of UNC13A from development, while our data (in our manuscript and the Brown et al manuscript) shows that UNC13A cryptic event can occur in up to 100% of UNC13A transcripts in TDP-43-affected neurons. Further, in the setting of FTD/ALS-associated TDP-43 nuclear clearing in human patients, we would expect that UNC13A loss would occur rapidly in adulthood, with likely very different compensatory mechanisms. To model the effects of Unc13a loss of function in adult mice would require the use of a conditional mouse model, which to our knowledge does not currently exist (or is at least not published), as indicated by Referee 3.

1. Presynaptic dysfunction as a disease-modifying element: In addition to Munc13-1, there is a surprisingly high number of central presynaptic proteins that appear to be mis-spliced (Fig. 1B: CADPS, ERC2, STXBP1, STBP5L, Syt7, several of the GEFs, are just those that I identify, see Taoufiq et al., PNAS 2020 or Koopmans et al., Neuron 2019). This is fascinating - all these proteins have been demonstrated to exert a major role in presynaptic function. A short description of the genetic nature of the mis-splicing events would be highly relevant to the synapse biology community and support the conclusion that presynaptic dysfunction is emerging a major mechanism in the pathogenesis of ALS/FTD.

Thank you for this great suggestion. We too are very excited by the enrichment of presynaptic genes as targets of TDP-43 cryptic exon splicing. We have now performed experiments to validate several of these cryptic splicing events in human iNs with TDP-43 knockdown and present this new data in the revised manuscript (Extended Data Fig. 6). We have highlighted the over representation of synaptic proteins whose mRNA appeared to be misspliced in the main text. In addition to UNC13A, we validated 3 additional pre-synaptic cryptic splicing targets (*KARLN*, *RAPGEF6*, and *SYT7*) in human iPSC-derived neurons, and include the specific details about these missplicing events. These are the ones that we are most interested in but we strongly suspect that others in the field will now want to follow up on these and others and that the focus on pre-synaptic function in ALS/FTD will open up new directions for the field.

2. Is the UNC13A CE introducing a stop codon? What is the mechanism to RNA decay? The statements in page 4 line 22 are not clear. Also, the supplemental note indicating three types of CEs is important and should be referred to in the main text. Yes, the UNC13A CE indeed introduces a stop codon. It is likely that this initiates nonsense mediated decay (NMD) of the transcript, as experimentally demonstrated by the accompanying manuscript (Brown et al). We have revised the main text to emphasize the different types of CEs.

3. In the minigene assay, how many CATC repetitions are included? Is there also a quantitative relationship there? Is this repeat affecting TDP-43 binding to that region? Or is it this just another marker?

In the minigene assay, 4 CATC repeats were included at rs56041637. We have updated the figure to clarify this point. Our new minigene reporter construct experiments in the revised manuscript (Extended Data Fig. 10) indicate that the increased repeat length, together with the cryptic exon and/or intronic risk SNPs promotes cryptic exon inclusion. The newly added TDP-43 binding experiments using electrophoretic mobility shift assays (EMSA) also suggested that TDP-43 had lower affinity for the radioactively labeled RNA probe containing the repeat sequence at rs56041637 compared to the probe containing the reference allele (Fig. 4g).

4. It is not clear whether and how the survival time analysis takes into account the third, newly-identified risk region. If not, the novelty of this analysis is not clear in view of the references mentioned by the authors. Please clarify.

The survival time analysis did not explicitly take into account the third risk region. As shown in Extended Data Fig. 9, the third region is in very high linkage disequilibrium with the other two known risk regions. In other words, the three risk regions tend to be inherited together. Therefore, the majority of the patients with the two known risk regions should also have the third risk region even though we did not try to genotype it. For the same reason, we also could not tease apart the effect of the third risk region on survival. The analysis itself is indeed not novel but it was just to show that, as it has demonstrated in the existing literature,

the risk haplotype is also a modifier of survival in the new patient population we have been analyzing and assessing *UNC13A* CE levels, etc.

5. Can the authors provide some evidence for reduced protein expression in patient-derived material?

Thank you for this suggestion. We have made several attempts to do this using both immunoblotting and immunohistochemistry on human patient brain sections. We have been unable to secure a suitable antibody that will allow TDP-43 + *UNC13A* double-immunofluorescence analyses (we have tried many – at least 5 different commercially available antibodies). Future work will be aimed at generating better reagents to analyze this comprehensively. Our studies of TDP-43 knockdown in human iPSC-derived motor neurons and iNs, indicates that *UNC13A* protein levels are decreased when TDP-43 is depleted.

6. The authors clearly demonstrate that TDP-43 pathology is necessary for detecting CE inclusion, and that in healthy individuals such inclusion cannot be seen. They also demonstrate that the fraction of CE-containing transcripts is increased as a function of increased pTDP-43 levels. Can the authors provide data for the time course of CE inclusion in patients, to test how this relates to disease progression? It would be fascinating to align such data to the time course of cell degeneration.

This is a very interesting suggestion and we would love to have this sort of data. We have thought about this suggestion carefully but it really seems quite difficult to be able to perform a convincing analysis. We would need to analyze a large number of patients and healthy subjects and obtain biofluids (e.g., CSF) and then be able to detect the CE in that material. So far we have not been able to develop a suitable assay for *UNC13A* CE detection in CSF, though we are continuing to work on it. Then, we would have to collect samples over the course of a patient's disease (perhaps several years) and then be fortunate to obtain autopsy material to confirm pathology and then correlate with levels of CE and other TDP-43 pathological features. We are really intrigued by doing this type of analysis but with currently available assays and the time it would take (many years), it is not currently possible to perform.

Referee #4

In the manuscript “TDP-43 represses cryptic exon (CE) inclusion in FTD/ALS gene *UNC13A*”, Ma et al. reveal that *UNC13A* gene is a target of the splicing regulator TDP-43. Loss of TDP-43 from the nucleus resulted in the inclusion of a cryptic exon splicing event in *UNC13A* transcript and reduced *UNC13A* protein expression. In addition, the authors showed that *UNC13A* genetic variants strongly associated with FTD/ALS risk are within the CE region, and these variants increase CE splicing when TDP-43 dysfunctions.

General Comments:

Overall, this study demonstrates a strong connection between *UNC13A* genetic variants and loss of TDP-43 function. The results are of immediate interest to people in neurodegeneration field. In general, the manuscript is clearly written and the findings are novel. The study provides a new explanation of how *UNC13A* genetic variants increase ALS/FTD risk and change survival. My main remaining question is to what extent this *UNC13A* CE event contributes to ALS/FTD pathology, which may be beyond the scope of this work.

We thank the Referee for his/her enthusiasm for the work and for appreciating that the results will be of immediate interest to the field. We are grateful for the very helpful comments and suggestions. We are very excited about the new directions that these findings will open up and are especially eager to figure out how *UNC13A* CE (and other CE events, like *STMN2* and others) contribute to FTD/ALS. The human genetics (*UNC13A* SNPs are among the strongest GWAS hits for FTD/ALS) tells us that *UNC13A* is important for disease risk and we have now functionally connected this to TDP-43 loss of function and provide evidence that the SNP that increases risk for FTD/ALS does so by making the *UNC13A* CE more prone to inclusion upon TDP-43 dysfunction.

1. Figure 1F, I wonder whether a double band was detected for *UNC13A* protein after shTDP-43 KD? What is the estimated protein size for *UNC13A* CE transcript?

We did not detect a double band for *UNC13A*. The antibody we used detects the N-terminal of *UNC13A*. Therefore, if there is a truncated protein and the protein is abundant enough for the antibody, we should be able to detect it. The estimated protein size for *UNC13A* CE transcript is 840 amino acid-long, whereas a canonical *UNC13A* protein is 1703 amino acid-long.

2. Figure 4g, Ext Figure 8b May the authors switch to two-way ANOVA to estimate the means of qPCR readout based on two categorical variables (siRNA and GFP constructs)?

We agree with this point and have incorporated your suggestions. The updated figure is in Extended Data Fig. 12.

3. Ext Figure 6c Please add a plot with quantitative measurement (e.g., Figure 3c).

Thank you for your suggestion. We have added the plot to Extended Data Fig. 8d in the revised manuscript. There is variability of *UNC13A* mRNA puncta detected across samples but we see a trend towards reduced levels of *UNC13A* mRNA in patient samples compared to controls, consistent with our RT-qPCR data in Fig. 1.

4. “To discover cryptic splicing targets regulated by TDP-43 that may also play a role in disease pathogenesis, we utilized a recently generated RNA sequencing (RNA-Seq) dataset... Here, we re-analyzed the data to identify novel 30 alternative splicing events impacted by the loss of nuclear TDP-43. We performed splicing analyses using two pipelines, MAJIQ 12 and LeafCutter 13, designed to detect novel splicing events (Fig. 1a).” Comprehensive data mining of publicly available NGS datasets using the latest bioinformatics approaches is critical to building this paper's hypothesis. The authors stated that “All data used in this study are available upon request. All codes used in this study are available upon request.” Can the authors upload their data processing codes to the GitHub repository so that the readers can repeat the published results (recommended by Nature Portfolio journals)? Also, I wonder whether the authors can release their own RNAseq data (e.g., a series of 117 frontal cortex brain samples from the Mayo Clinic Brain Bank) to a data portal?

Thank you for your suggestion. We are absolutely committed to making all of our data easily accessible so that others in the field can repeat and extend our analyses in any way they would like. We have uploaded the codes to <https://github.com/rosaxma/TDP-43-UNC13A-2021>. The 117 brain samples were not RNA sequenced. The cryptic exon inclusion level in these samples were measured using RT-qPCR.

5. “we scored the *UNC13A* splice variant as present if there were more than two reads spanning at least one of the exon-exon junctions.” Are those reads uniquely mapped?

Thank you for pointing this out. We had indeed forgotten to filter for uniquely mapped reads. We have now added this step and as a result, one of the samples did not meet the criteria that we had set. We have updated the statistics in Fig 2b and in the Methods (“Quantification of *UNC13A* splice variants in bulk RNA sequencing”; p. 58).

6. Ext Figure4 Can the authors use one-way ANOVA test? Looks like there is no significant difference between sgTDP-43-guide1 and sgTDP-43-guide2, is that correct?

It is correct, there is no significant difference between the two guides. We agree with the comment and have incorporated your suggestion (p. 30).

Referee #5

A. In this study, Ma et al mine the Liu et al RNAseq dataset from TDP-43 mutant brain and identified incorporation of an out-of-frame CE unexpectedly incorporated in UNC13, a gene previously implicated in FTD/ALS through GWAS. The GWAS SNPs occur proximal to the CE, and Ma demonstrates in human brain, neuronal cell lines, and iPSC derived motor neurons that there is incorporation of this unusual CE. Most convincing is that patient FTLTDP but not other genetic forms of FTLTDP samples show CE inclusion using two methods. Their data suggest that UNC13A risk SNPs, or a 4bp STR in linkage disequilibrium, increase the rate of CE incorporation following TDP-43 depletion.

B. This study links several previous studies in meaningful ways, and helps explain why the UNC13A SNPs are disease-associated. This may be one of the first examples of a GWAS peak that is later discovered to mediate to cryptic exon incorporation.

C. In general the approaches appropriately support the conclusions. Quality of data is excellent. The presentation could be further improved, as suggested below.

We thank the Referee for his/her careful reading of our manuscript and for appreciating the importance of our discovery explaining why *UNC13A* SNPs are disease-associated (because they promote cryptic exon inclusion when TDP-43 is depleted from the nucleus or dysfunctional). Thank you for the helpful comments and suggestions.

1. Fig 1b, all the genes are $P(\Delta\Psi > 0.1) > 0.95$. Is this what authors define as highly significant? Or does *UNC13A* have a higher p-value indicated by the color?

Thank you for pointing these out.

1. We have clarified the rationale for picking this threshold in Supplementary Note 1. Briefly, instead of providing p-values, MAJIQ provides a Bayesian posterior for $P(\Delta\Psi > C)$, which essentially means the Bayesian posterior probabilities of $\Delta\Psi$ (Ψ represents changes of local splicing variations between two conditions) larger than a constant C. Therefore, $P(\Delta\Psi > 0.1) > 0.95$ indicates that based on the data from Liu et al, MAJIQ is more than 95% confident that the genes in the list have $\Delta\Psi$ larger than 0.1.

Filtering LeafCutter outputs for genes that contain splicing events with p-values ≤ 0.05 gave us 139 unique genes. Therefore, to get an equivalent number of genes from the output of MAJIQ, we set the C as 0.1, which is more permissive than the default setting 0.2. We decided to decrease the threshold for $\Delta\Psi$ because for our study the confidence in changes is more interesting than the magnitude of changes, and small changes in transcriptome could be sufficient to lead to functional changes.

2. We highlighted *UNC13A* in the table because it is a known GWAS hit and a gene we decided to follow up in this paper. To emphasize this point, we have highlighted it again in the figure legend.

2. The minigene reporter assay would be more informative if the specific risk variants could be further functionally interrogated to determine which one(s) bias CE incorporation. Does the critical variant influence TDP-43 binding in the author's hands?

We did this as suggested and we also tested all permutations (single SNPs, doubles, and all 3). In total, we tested 8 constructs in the revised manuscript. We show these new data in Extended Data Fig. 10, which indicate that the risk variant at rs12973192 is the strongest contributor to cryptic exon inclusion but the other ones do contribute and are not simply additive (Extended Data Fig. 10). Because of linkage disequilibrium and the general principle of linked variants, we were and still are hesitant to pin all of the effects on one SNP and these new results support that notion. Our findings indicate that the *UNC13A* risk haplotype that is associated with increased risk for ALS and FTD/ALS causes increased cryptic splicing in the face of TDP-43 loss of function. The lead GWAS SNP that is located right in the cryptic exon itself (rs12973192) can impact cryptic splicing by itself but there are likely additional variations contributing to cryptic splicing. Future efforts by us and others to fine map the locus will be informative.

3. The finding that the SNPs affect survival in FTLD-TDP patients is positive, but it would be interesting to understand if they have no effect on survival in other forms of disease. If they still have an impact would the authors consider that there are genes other than TDP-43 influencing CE incorporation?

Thank you for your suggestion. This is a very interesting idea. The Referee seems to be asking if the SNPs in *UNC13A* that increase risk for FTD/ALS do so in other forms of FTD or ALS not associated with TDP-43 pathology. It would have been interesting to explore this aspect. However, this would require information from forms of disease that normally do not display TDP-43 pathology, for example, FTLD-FUS, FTLD-TAU and ALS-SOD1 patients. The few cases of these that we have looked at and for which we have *UNC13A* SNP genotype data, we do not see any large differences in survival either way, but the numbers are really too small to make any firm conclusions at this point. Because such patients are rare, we do not have enough samples to conduct well-powered survival analyses. It is possible that TDP-43 suppresses splicing by recruiting other splicing factors. Therefore, the downstream splicing factors could also influence CE incorporation as well. That said, we do show in Extended Data Fig. 11 that by upregulating TDP-43 lacking IDR region, we see a partial rescue of the cryptic splicing, suggesting the binding of TDP-43 to the intron is important. It will certainly be a future goal to discover the complete mechanism by which TDP-43 represses cryptic exon inclusion (including identifying potential cofactors).

4. Does population VAF for UNC14B risk (maybe you can find them in 1000Genome/gnomAD) correlate with the prevalence rate of ALS/FTD in different populations (PMID 22231873, 30879893, and 27185810)?

Thank you for the questions. It would have been interesting to explore this aspect, however, with the data that are currently available, we are not able to provide a definite answer.

The minor allele frequencies at rs12973192 and rs12608932 and linkage disequilibrium (LD) statistics between them are shown in the table below. Notably, the LD is the strongest in the EUR population. The data was from the 1000 Genomes Project

Since ALS or motor neuron disease is highly prevalent in EUR and less prevalent in AFR, AMR, EAS and SAS) (GBD 2016 Motor Neuron Disease Collaborators, 2018), the frequencies of the minor allele at rs12973192 are roughly in congruence with the prevalence of motor neuron disease in different populations. However, such correlation does not apply to rs12608932, possibly due to the weaker LD between the two SNPs in some populations.

As discussed in the paper (GBD 2016 Motor Neuron Disease Collaborators, 2018), in addition to ethnicities and ancestries, such differences could also be attributed to better diagnosis and treatment strategies in regions that are commonly resided by EUR populations.

Here in our paper, we focused on the genetics of the EUR populations and owing to the various confounders and insufficient data, we are hesitant to say that the VAF for UNC13A risk alleles are correlated with the prevalence rate of ALS/FTD in different populations.

	rs12973192 (G)	rs12608932 (C)	D'	R ²
AFR (African)	0.234	0.330	0.9904	0.6106
AMR (American)	0.216	0.304	0.9808	0.6072
EAS (East	0.196	0.696	1.0	0.1066

Asian)				
EUR (European)	0.342	0.349	0.9911	0.9525
SAS (South Asian)	0.399	0.484	1.0	0.7081

5. Page3 Line 15, Data Fig. 3a and 3b, the authors mention direct sequencing, but the figure shows only RT-PCR results.

Thank you for pointing this out. The mapping of the direct sequencing has been added to Extended Data Fig 5.

6. Extended Data Fig. 4. Both bars with significant changes appear to refer to guide2. Was guide1 non-functional?

Thank you for pointing this out. It was indeed a mistake on our side. We have changed the colors of the bars to match the figure legend.

7. Recommend indicating in Fig. 1a the data is generated in Liu et al. PMID 31042469.

Thank you for your suggestion. We have added the element to the figure.

8. Extended Fig 2a the authors might want to highlight where the CE is?

Thank you for your suggestion. We have added the element to the figure (Extended Data Fig. 2).

9. Page3 Line 45, are these calculated with the same criteria?

Yes, they are calculated with the same criteria.

10. Fig. 2c and Extended Data Fig. 5c are probably redundant as log-transformation does not change the rank so Spearman correlation should have exactly the same rho and p-value.

We agree with this. We have removed the log-transformation and Fig 2c is now showing the raw data.

11. Why SNP genotyping instead of direct Sanger sequencing for genotyping the SNPs? There are only 600 bp between rs12608932 and rs12973192, so why is not possible to genotype the entire haplotype with a single Sanger reaction?

Thank you for this suggestion. The genotyping was performed at the early stage of the project and we decided to focus on the two SNPs because they are known GWAS hits. We were also able to make the experiment high throughput by using Taqman SNP Genotyping Assay. While it could be interesting to perform direct Sanger sequencing for the entire haplotype, since rs12608932 and rs12973192 are in very high LD (linkage disequilibrium) in the population we are looking at (as indicated by the existing genotyping data shown in the response to comment 4) and they are also in high LD with rs56041637 as indicated by the LD analysis presented in Extended Data Fig. 9. We think performing additional Sanger sequencing may only add limited information.

12. Ref 38, 42 has a different ref format.

Thank you for your suggestion. We have corrected it.

Reviewer Reports on the First Revision:

Referee #1

The authors have addressed my comments in a satisfactory manner.

Referee #2

The revised manuscript by Ma et al is significantly improved and addresses most of the important points raised by all five referees. I congratulate the authors for the thorough rebuttal and the additional data that they present in their revised manuscript. They clarify all the points that have been ambiguous in the earlier version. In particular, they added new experimental data that validate additional synaptic targets that they identified beyond UNC13A. They also extended their work on the binding of TDP-43 on the risk alleles and coordinated with the authors of the accompanying manuscript of Brown et al regarding the detection of the CE-containing transcript. Overall, I think the work is truly exciting and thorough and should be published in Nature.

Without attempting to significantly delay publication, I would like to insist on the only important point that they have not addressed, which is the functional consequence of UNC13A loss for adult neurons. This has been my main concern and I see that some of the other referees raised the same point. The authors argue that since there are several other synaptic targets of TDP-43 with possibly similar effects, it might be difficult to detect an UNC13A-specific effect. I would counterargue that exactly because TDP-43 has so many important targets in neurons, it is critical to test this. In my view, depending on the impact of UNC13A loss in the context of TDP-43 pathology/misregulation, this target might be anywhere between "the" key effector of neurotoxicity to an inert bystander (in reality it is probably somewhere in-between). I understand that the full mechanistic elucidation is complex and may be beyond the scope of the current study and I do not propose to significantly delay the publication of this exciting finding. However, I think that an initial test of the functional consequences of UNC13A loss for adult neurons and/or rescue experiments by combining TDP-43 loss with co-expression of a TDP-43-insensitive UNC13A construct may be more revealing than the authors predict. I agree that the genetic implications of this gene as an ALS risk argue for an important role of UNC13A in neurodegeneration. I am therefore more optimistic that this direct link would be demonstratable experimentally.

Referee #3

The manuscript reviewed here by Ma et al., together with the accompanying manuscript by Brown et al., present together very compelling evidence for the involvement of TDP-43-dependent mRNA processing events in Munc13-1 and other synaptic protein in the pathophysiology of FTD/ALS. Reading this manuscript and rebuttal, I consider the manuscript strong and recommend it for publication in Nature. Albeit not all issues have addressed or reconciled, I predict that the findings presented here will be followed-up by the authors and by additional labs, and that the mechanisms underlying these very important findings will be clarified in the coming years.

One note to the authors – FTD/ALS are degenerative disorders. The obvious effect of neuronal loss is the concomitant loss of synaptic transmission and information transfer within neuronal networks. The loss of the Munc13-1 protein is expected to cause the same effect – inhibition of neurotransmission, albeit without neuronal loss. A plausible mechanism for Munc13-1 variations being a risk factor is that in parallel to neuronal loss, the remaining, ‘still surviving’ neurons will exhibit reduced synaptic transmission due to reduced Munc13-1 levels, resulting in an apparent neurodegeneration phenotype.

Referee #4

All my concerns related to statistics and RNA-seq bioinformatics have been addressed. I don't have further comments.

Referee #5

The authors have addressed our concerns. We also checked the responses from the authors to the other reviewers, and we feel that the authors responses were also satisfactory. The authors have demonstrated convincing evidence that TDP-43 represses cryptic exon inclusion in UNC13A. We think the paper will change the way we think about ALS.

Author Rebuttals to First Revision:

We have reduced the size of the manuscript to fit within 8 pages, based on the macro that you provided. We cut some text and moved some parts of main figures into extended data figures. We have also consolidated the extended data figures down to 10, per your guidance.

Here are responses to the specific Editorial requests (responses in **bold**):

1. Data presentation: Please ensure that data presented in a plot, chart or other visual representation format shows data distribution clearly (e.g. dot plots, box-and-whisker plots). When using bar charts, please overlay the corresponding data points (as dot plots) whenever possible and always for $n \leq 10$.

Done

2. Statistics: Wherever statistics have been derived (e.g. error bars, box plots, statistical significance) the legend needs to provide and define the n number (i.e. the sample size used to derive statistics) as a precise value (not a range), using the wording “n=X biologically independent

samples/animals/cells/independent experiments/n= X cells examined over Y independent experiments” etc. as applicable.

Legends requiring revision:

Please note that this information is missing in the legends of figures 3c; 4b ,4f and in the legends of extended data figures 4; 7a; 10b-10c; 11b; 12b-12c.

The information has been included in 3c (now Extended data Fig. 6), 4b, 4f (now Extended data Fig. 9), Extended data Fig. 4 (now Extended Data Fig. 3c), 7a (now Extended data Fig. 5a), 10b - 10c (now Extended Data Fig. 8d-8e, 11b (now Extended Data Fig. 9b), 12b-12c (now Extended Data Fig. 12e-12f).

3. Statistics such as error bars, significance and p values cannot be derived from $n < 3$ and must be removed from all such cases. Please note that this should be rectified for figure 4g (if $n=2$).

Error bars and p-values have been removed from figure 4g and Extended Figure. We also removed such elements from Extended Data Fig.8d and e.

All error bars need to be defined in the legends (e.g. SD, SEM) together with a measure of centre (e.g. mean, median). For example, the legends should state something along the lines of “Data are presented as mean values \pm SEM” as appropriate. All box plots need to be defined in the legends in terms of minima, maxima, centre, bounds of box and whiskers and percentile.

Legends requiring revision:

1. Please note that the measure of centre for the error bars needs to be defined in the legend of figure 2a and in the legend of extended data figure 7a.

The center for the error bars have been defined in both figure 2a and figure 7a (Now Extended Data Fig. 5a).

4. The figure legends must indicate the statistical test used. Where appropriate, please indicate in the figure legends whether the statistical tests were one-sided or two-sided and whether adjustments were made for multiple comparisons. For null hypothesis testing, please indicate the test statistic (e.g. F, t, r) with confidence intervals, effect sizes, degrees of freedom and P values noted. Please provide the test results (e.g. P values) as exact values whenever possible and with confidence intervals noted.

Legends requiring revision:

1. Please indicate the statistical test used for data analysis and where appropriate, please specify whether it was one-sided or two-sided and whether adjustments were made for multiple comparisons,

in the legend of figure 2c and in the legends of extended data figures 8c; 14a, 14c, 14e; in the legend of extended data table 1.

Statistical tests are specified in Figure 2c (now Extended Data Fig. 5c), Extended Data Fig. 8c (now Extended Data Fig. 6e), Extended Data Figs. 14a, 14c and 14e (now Extended Data Figs. 10b, 10d, 10f), and Extended Data Table 1 (now Supplementary Table 1).

2. Please note that the exact p value should be provided, when possible, in the legends of figures 1e, 1g-1h, 1j; 2a, 2c; 3c; 4f and in the legends of extended data figures 4; 6g-6i; 7a; 11b; 12b; in the extended data table 4.

The exact p values have been included when applicable.

5. Reproducibility: Please state in the legends how many times each experiment was repeated independently with similar results. This is needed for all experiments, but is particularly important wherever results from representative experiments (such as micrographs) are shown. If space in the legends is limiting, this information can be included in a section titled “Statistics and Reproducibility” in the methods section.

Legends requiring revision:

Please note that this information is missing in the legend of figure 3b and in the legends of extended data figures 3b; 8a; 13.

Updated legends of Figure 3b, Extended Data 3b (now Extended Data Fig. 3a and b), Extended Data Fig. 8a (now Extended Data Fig. 6a), and Extended Data Fig. 13 (now Extended Data Fig. 10a) with this information.

6. Data availability: Please ensure that datasets deposited in public repositories are now publicly accessible, and that accession codes or DOI are provided in the "Data Availability" section. As long as these datasets are not public, we cannot proceed with the acceptance of your paper. For data that have been obtained from publicly available sources, please provide a URL and the specific data product name in the data availability statement. Data with a DOI should be further cited in the methods reference section.

Data deposited in GEO is now publicly available.

7. Gels and Blots: Quantitative comparisons between samples on different gels/blots are discouraged; if this is unavoidable, the figure legend must state that the samples derive from the same experiment and that gels/blots were processed in parallel.

Panels requiring revision:

1. Unit for molecular weight markers are missing for figures 1f, 1i and for extended data figure 3b.

Figure 1f, 1i (Now Fig. 1d and g), and Extended data 3b (now Extended Data Fig. 3a and b) have been updated.

2. Full scans are missing for figure 4e.

Full scan has been included in Supplementary Figure 1.

8. Micrographs: Please ensure that all micrographs include a scale bar and this scale bar is defined on the panels or in the figure legends.

Panels requiring revision:

1. Please note that the scale bar needs to be defined for extended data figure 8a.

The scale bar has been defined in the figure legends of Extended Data Fig 6a.

Please let me know if you have any questions or require additional information.

I wish to participate in transparent peer review.

Thank you very much for your interest in this work.